# Discrete Diffusion Schrödinger Bridge Matching for Graph Transformation

**Jun Hyeong Kim**[1][*], **Seonghwan Kim**[1][*], **Seokhyun Moon**[1][*], **Hyeongwoo Kim**[1][*],
**Jeheon Woo**[1][*], **Woo Youn Kim**[1][†]
[1]KAIST

## Abstract

Transporting between arbitrary distributions is a fundamental goal in generative modeling. Recently proposed diffusion bridge models provide a potential solution, but they rely on a joint distribution that is difficult to obtain in practice. Furthermore, formulations based on continuous domains limit their applicability to discrete domains such as graphs. To overcome these limitations, we propose Discrete Diffusion Schrödinger Bridge Matching (DDSBM), a novel framework that utilizes continuous-time Markov chains to solve the SB problem in a high-dimensional discrete state space. Our approach extends Iterative Markovian Fitting to discrete domains, and we have proved its convergence to the SB. Furthermore, we adapt our framework for the graph transformation and show that our design choice of underlying dynamics characterized by independent modifications of nodes and edges can be interpreted as the entropy-regularized version of optimal transport with a cost function described by the graph edit distance. To demonstrate the effectiveness of our framework, we have applied DDSBM to molecular optimization in the field of chemistry. Experimental results demonstrate that DDSBM effectively optimizes molecules' property-of-interest with minimal graph transformation, successfully retaining other features. Source code is available at here.

## 1 Introduction

Transporting an initial distribution to a target distribution is a foundational concept in modern generative modeling. Denoising diffusion models (DDMs) have been highly influential in this area, with a primary focus on generating data distributions from simple prior (Sohl-Dickstein et al., 2015; Song & Ermon, 2019; Ho et al., 2020; Song et al., 2020; Kim et al., 2024b). Despite their promising results, setting the initial distribution as a simple prior makes DDMs hard to work in tasks where the initial distribution becomes a data distribution, such as image-to-image translation. To tackle this, diffusion bridge models (DBMs) extend DDMs to transport data between arbitrary distributions (Liu & Wu, 2023; Liu et al., 2023; Zhou et al., 2023). However, training DBMs requires a coupling between the initial and target distributions, which is often difficult to obtain in practice. The Schrödinger Bridge (SB) provides an attractive framework for constructing a joint distribution of two data distributions while aligning with the underlying stochastic dynamics (Pariset et al., 2023; Kim et al., 2023; Dong et al., 2024).

Formally, the SB problem seeks the stochastic process that connects two distributions and is closest to a reference process, measured by the Kullback-Leibler (KL) divergence (Schrödinger, 1932). The SB problem can be described as an entropy-regularized optimal transport (EOT) problem, which introduces an entropy term to the optimal transport (OT) objective, resulting in randomness in the transport process (Léonard, 2013). Here, the transportation cost is determined by the system's natural dynamics; for example, in the case of Brownian motion, the transportation cost becomes $L^2$ (De Bortoli et al., 2021). The SB/EOT can be computed efficiently using the Sinkhorn algorithm, though high-dimensional or large-scale data applications remain challenging (Sinkhorn, 1967; Cuturi, 2013). In recent, many methods have been proposed to approximate SB via distribution learn-

---

[*]Equal contribution
[†]Correspondence to wooyoun@kaist.ac.kr

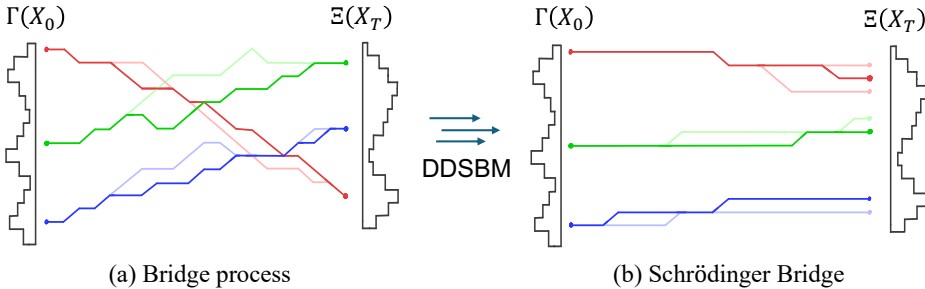

(a) Bridge process                                          (b) Schrödinger Bridge

Figure 1: A schematic illustration of DDSBM transforming (a) bridge process to (b) Schrödinger Bridge in discrete spaces.

ing, utilizing techniques developed in DBM and DDM (De Bortoli et al., 2021; Liu et al., 2022; Somnath et al., 2023; Liu et al., 2023; Peluchetti, 2023a; Shi et al., 2024; Lee et al., 2024).

Despite the progress, most of the methods focus on the continuous spaces, where diffusion processes are represented by Brownian motion, and SB problems in discrete domains are less explored. It is particularly significant in fields that handle discrete state data, such as graphs or natural language (Austin et al., 2021; Vignac et al., 2022). Directly applying frameworks for approximating SB formulated in continuous spaces to these domains limits its potential since it does not reflect the intrinsic properties of discrete data space. To bridge the gap, we propose a novel framework called Discrete Diffusion Schrödinger Bridge Matching (DDSBM) utilizing the continuous-time Markov chains (CTMCs) to solve the SB problem in a high-dimensional discrete setting. Our approach leverages Iterative Markovian Fitting (IMF), which was originally proposed for the SB problem in a continuous domain (Peluchetti, 2023a; Shi et al., 2024).

We then extend our formulation to the graph domain, where the underlying process can be interpreted as independent modifications to both the nodes and edges (Vignac et al., 2022). In this case, the cost function of the corresponding EOT can then be regarded as the graph edit distance (GED), which is especially suited for systems where preserving graph similarity is crucial (Sanfeliu & Fu, 1983). The molecular optimization in drug/material discovery is such a case in that molecules are represented as graphs. In addition, the goal is to obtain the molecules with desired molecular properties while retaining favorable properties in acclaimed molecules. Since molecular structures are closely related to their properties, it is highly advantageous to minimize structural changes (or graph editing) during the optimization.

To validate our framework, we evaluated the performance of DDSBM on molecular optimization tasks, with criteria of demonstrating optimal structural modifications to achieve desired property. DDSBM shows promising results in molecular distribution shift with minimum structural change compared to the previous graph-to-graph translation models. As a direct result of this, DDSBM retains multiple properties of the initial molecules, along with desired property. Lastly, we applied DDSBM for a more challenging task, where proper joint pairing between two molecular spaces can not be defined properly.

Our contributions are as follows:

- We propose a novel framework, DDSBM, for the SB problem in discrete state spaces by exploiting the IMF procedure and prove its convergence to the SB solution.

- We extend our framework to the graph domain, demonstrating a connection between the objective function and the GED.

- By reformulating molecular optimization as the SB problem, we show that our approach successfully addresses molecular optimization tasks.

## 2 RELATED WORKS

**Diffusion Bridge Models.** Diffusion bridge models (DBMs) have recently shown state-of-the-art results in a variety of continuous domains, such as images, biology, and chemistry (Liu & Wu, 2023; Liu et al., 2023; Somnath et al., 2023; Zhou et al., 2023; Lee et al., 2024). While Igashov et al. (2023); Yang et al. (2023) have extended these models to discrete domains, they focused on settings where well-defined data pairs exist. To the best of our knowledge, we are the first to study DBM in discrete domains where proper joint distributions between data points are not provided or well-defined.

**Schrödinger Bridge Problem.** The Schrödinger Bridge (SB) problem is an important concept in recent generative modeling (Liu et al., 2022; Somnath et al., 2023; Peluchetti, 2023a; Shi et al., 2024). In particular, incorporating the SB problem into DBM can address scenarios where no appropriate joint distribution is available, as demonstrated by recent works (Pariset et al., 2023; Kim et al., 2023; Dong et al., 2024). For example, Somnath et al. (2023) proposed learning an SB based on an assumed (partial) true coupling, while Shi et al. (2024) showed that it is possible to generate high-quality samples from arbitrary couplings that are well-aligned with the initial data. However, most existing approaches focus on continuous spaces. To the best of our knowledge, we are the first to propose a framework for solving the SB problem in high-dimensional discrete spaces.

**Molecular Optimization.** Molecular optimization is a promising strategy in drug/material discovery that aims to improve acclaimed molecules to satisfy multiple domain-specific properties. One major approach to the molecular optimization problem is to formulate it as a graph transformation problem, which can be categorized into latent-based and graph-editing approach. The latent based approaches such as JT-VAE by Jin et al. (2018) and HierG2G by Jin et al. (2020) encode an input graph into a single latent vector and then decode it into a whole graph that follows a certain data distribution. The graph-editing approaches such as Modof by Chen et al. (2021) and DeepBioisostere by Kim et al. (2024a) learn a graph editing procedure to transport a given molecule to another. Despite their promising results in optimizing molecular graphs, their training schemes rely on paired data created by predefined rules, which limits not only the general applicability but also the optimality of the transformations. In this work, we propose a more flexible framework for molecular optimization by formulating it as a graph SB problem, leading to more optimal graph transformation accompanying less structural changes.

## 3 THEORETICAL BACKGROUND

### 3.1 SCHRÖDINGER BRIDGE PROBLEM

Consider the standard Brownian motion $(X_t)_{0 \le t \le \tau}$ defined in Euclidean space, taken as a reference process with a given transition density. Given an initial distribution $\Gamma$ for $X_0$, the distribution for $X_\tau$ is subsequently determined by $\Gamma$ and the transition density of the reference process. Suppose we not only prescribe the initial distribution $\Gamma$ but also specify a terminal distribution $\Xi$ at $t = \tau$. In this scenario, the reference process $X_t$ generally fails to hold the boundary conditions. Schrödinger Bridge (SB) problem is finding the process that closely resembles the reference process while satisfying the boundary conditions on initial and terminal distributions. For an intuitive introduction to the SB problem, please refer to Appendices A.1 and A.2.

Specifically, the reference process with the path measure $\mathbb{Q}$ is given, the SB problem is finding a process with path measure $\mathbb{P}$ by minimizing Kullback-Leibler (KL) divergence to the reference process, $D_{\text{KL}}(\mathbb{P}\|\mathbb{Q})$. The SB solution is characterized as below:

$$\mathbb{P}^{\text{SB}} = \arg \min_{\mathbb{P}} \{ D_{\text{KL}}(\mathbb{P}\|\mathbb{Q}) : \mathbb{P}_0 = \Gamma, \mathbb{P}_\tau = \Xi \}. \tag{1}$$

If we additionally fix the initial and terminal coupling (joint distribution) $\mathbb{P}_{0,\tau}$, the optimality can be found easily as a mixture of bridges $\mathbb{P}_{0,\tau}\mathbb{Q}_{\cdot|0,\tau}$ (see formal definition at Definition 3.1), which implies that finding the SB solution is equivalent to finding optimal coupling $\mathbb{P}_{0,\tau}^{\text{SB}}$ (Léonard, 2013). In particular, such optimal coupling is called static SB solution, which could be defined as follows:

$$\mathbb{P}_{0,\tau}^{\text{SB}} = \arg \min_{\mathbb{P}_{0,\tau}} \{ D_{\text{KL}}(\mathbb{P}_{0,\tau}\|\mathbb{Q}_{0,\tau}) : \mathbb{P}_0 = \Gamma, \mathbb{P}_\tau = \Xi \}. \tag{2}$$

Note that the KL-divergence is decomposed into the entropy term $H(\mathbb{P}_{0,\tau})$ and the cross-entropy term $\mathbb{E}_{\mathbb{P}_{0,\tau}} [-\log q(x_\tau|x_0)]$, where $q$ denotes the transition density of $\mathbb{Q}_{\tau|0}$. The transition density in cross-entropy term is $L^2$ distance when the reference process is the standard Brownian motion. In general, the static SB problem is equivalent to the entropy-regularized optimal transport (EOT) problem with the cost function $c(x, y) = -\log q_{\tau|0}(y|x)$ (Léonard, 2013).

## 3.2 STOCHASTIC PROCESS OVER DISCRETE SPACE

To extend the SB problem to discrete state spaces, a suitable stochastic process is necessary. Unlike continuous spaces, where Brownian motion serves as the canonical reference process, discrete spaces require a distinct approach that aligns with their structure.

Let the state space $(\mathcal{X}, d_\mathcal{X})$ be a finite metric space, where $\mathcal{X}$ is the finite set of states, and $d_\mathcal{X}$ represents the distance between states. The corresponding path space $\Omega = D([0, \tau], \mathcal{X})$ consists of all left-limited and right-continuous (cádlág) paths over $\mathcal{X}$ within the time interval $[0, \tau]$. A path $\omega \in \Omega$ represents a sequence of states indexed by time, where $\omega_t \in \mathcal{X}$ denotes the state at time $t$. The space of path measures is denoted $\mathcal{P}(\Omega)$ and the set of Markov path measures is denoted as $\mathcal{M} \subset \mathcal{P}(\Omega)$.

In this work, we model the reference path measure for the SB problem, $\mathbb{Q} \in \mathcal{M}$, as continuous-time Markov chains (CTMCs). A CTMC describes the stochastic evolution over $\mathcal{X}$ in continuous time, which is characterized by transition rates. The transition rate $A_t(x, y)$ defines the instantaneous rate of transition from $x$ to $y$ at time $t$. This rate satisfies $A_t(x, y) \geq 0$ for $x \neq y$, and $\sum_{y \in \mathcal{X}} A_t(x, y) = 0$. The time evolution of the stochastic process is governed by the Kolmogorov forward equation:

$$\frac{\partial P_{s:t}(x, y)}{\partial t} = \sum_{z \in \mathcal{X}} P_{s:t}(x, z) A_t(z, y), \tag{3}$$

where $P_{s:t}(x, y)$ is the probability of transitioning from $x$ at time $s$ to $y$ at time $t$. To ensure that the reference process $\mathbb{Q}$ is suitable for constructing the SB problem, we assume that the reference process is an irreducible CTMC, meaning that every state in $\mathcal{X}$ can be reached from any other state. More specifically, for any pair of states $x, y \in \mathcal{X}$, $\mathbb{Q}_{\tau|0}(y|x) := P_{0:\tau}(x, y) > 0$.

## 3.3 EXTENSION OF ITERATIVE MARKOVIAN FITTING METHOD TOWARD DISCRETE SPACE

Recently, there have been many studies to solve the SB problem with diffusion processes on smooth manifolds using denoising score matching (Peluchetti, 2023a; Shi et al., 2024). We refer to Appendix A.3 for a brief description of the method. While these approaches primarily focus on continuous spaces, we here aim to address the SB problem in discrete spaces, specifically dealing with left limited and right continuous (càdlàg) paths over finite state spaces. While it is known that the SB problem can be solved in finite state spaces (Sinkhorn, 1967; Cuturi, 2013; Chow et al., 2021), existing methods face limitations when handling high-dimensional or large-scale data and are not suitable for generative modeling applications. For an intuitive explanation and our insights into the discrete SB problem, refer to Appendix A.4.

Overcoming these challenges, we extend the Iterative Markovian Fitting (IMF) method, originally formulated on continuous diffusion processes, to the discrete setting of Markov chains over finite state spaces. The IMF method and projection operations were introduced in previous work (Peluchetti, 2023a; Shi et al., 2024); we adopt these concepts without conceptual changes. Our contribution lies in providing the theoretical extension toward the discrete setting, which to the best of our knowledge, has not been previously established.

In this subsection, we introduce the previously developed IMF method and prove a convergence theorem specific to the discrete-state problem setting, with detailed proofs provided in Appendix B. Despite extending the IMF method to discrete spaces, our approach is still fundamentally rooted in the intrinsic properties of the SB solution. Specifically, we rely on the representation of the unique Markov measure as a mixture of bridges $\mathbb{P}_{0,\tau}\mathbb{Q}_{\cdot|0,\tau}$ (see Theorem B.1).

According to Theorem B.1, the SB solution is a mixture of pinned-down measures of $\mathbb{Q}(\cdot|X_0 = x_0, X_\tau = x_\tau)$, where the pair $(x_0, X_\tau = x_\tau)$ is drawn from the coupling $\mathbb{P}_{0,\tau}^{SB}$. Based on this, the projection method first constructs a reciprocal measure, which is a mixture of pinned-down processes from a

given initial coupling. Although each pinned-down process is Markov, the mixture generally loses the Markov property in general as a collection of Markov processes is non-convex (Léonard et al., 2014). Thus, it then identifies the Markov measure that is closest to the mixture. This yields an improved coupling, and the process is iterated to obtain a measure that is both the mixture of pinned-down measures and Markov—the desired SB solution.

In this context, we define the reciprocal projection to describe the construction of a reciprocal mixture from a given coupling (see Definition 3.1). Similarly, the term Markov projection is used to describe the approximation of a reciprocal process with a Markov process (see Definition 3.2).

**Definition 3.1.** *(Reciprocal Projection)*
*$\Lambda \in \mathcal{P}(\Omega)$ is in the reciprocal class $\mathcal{R}(\mathbb{Q})$ with respect to a Markov measure $\mathbb{Q}$ if $\Lambda = \Lambda_{0,\tau} \mathbb{Q}_{|0,\tau}$.*
*For a measure $\Lambda \in \mathcal{P}(\Omega)$, its reciprocal projection with respect to $\mathbb{Q}$ is*

$$\Pi_{\mathcal{R}(\mathbb{Q})}(\Lambda)(\cdot) := \iint_{(\cdot)} \Lambda(dx_0, dx_\tau) \mathbb{Q}(dx_t | x_0, x_\tau).$$

Among the measures with the coupling $\Lambda_{0,\tau} \neq \mathbb{P}_{0,\tau}^{\text{SB}}$, the minimizer of the KL-divergence to the reference process is the (non-Markov) reciprocal projection $\Pi_{\mathcal{R}}(\Lambda)$. The reciprocal projection consists of a mixture of bridges, where each bridge is derived from Doob's h-transform with the realization of an end-point pair $(x, y) \sim \Lambda_{0,\tau}$ (Levin & Peres, 2017). Obviously, it preserves the initial-terminal coupling. Although each pinned-down bridge is Markov (see Lemma B.4), the mixture is generally not Markov.

**Definition 3.2.** *(Markov Projection)*
*Given a path measure $\Lambda \in \mathcal{R}(\mathbb{Q})$, a Markov path measure that minimizes the reverse KL-divergence to $\Lambda$ is called as Markov projection of $\Lambda$,*

$$\Pi_{\mathcal{M}}(\Lambda) = \arg\min_M \{ D_{KL}(\Lambda \| M) : M \in \mathcal{M} \}.$$

The Markov projection preserves the marginal distribution at all times $t$, but does not preserve the coupling. Furthermore, the generator of the projected Markov measure and the reverse KL-divergence $D_{\text{KL}}(\Lambda \| \Pi_{\mathcal{M}}(\Lambda))$ are explicitly derived in the Proposition B.2.

For a given reciprocal process $\Lambda^{(0)} \in \mathcal{R}(\mathbb{Q})$, we consider a sequence $(\Lambda^{(n)})_{n \in \mathbb{N}}$ which is defined by the recurrence relation:

$$\Lambda^{(2n+1)} = \Pi_{\mathcal{M}}(\Lambda^{(2n)}), \tag{4}$$
$$\Lambda^{(2n+2)} = \Pi_{\mathcal{R}(\mathbb{Q})}(\Lambda^{(2n+1)}),$$

where $\Lambda_0^{(0)} = \Gamma$ and $\Lambda_\tau^{(0)} = \Xi$. Under mild assumptions, the resulting sequence of measures of iterative projection converges to $\mathbb{P}^{\text{SB}}$ in law (see Theorem 3.3 and Appendix B.5).

**Theorem 3.3.** *(Convergence of Iteration)*
*Assume that $D_{KL}(\Lambda_{0,\tau}^{(0)} \| \mathbb{P}_{0,\tau}^{SB}) < \infty$, $\Lambda^{(n)} \ll \mathbb{P}^{SB}$ for all $n \in \mathbb{N}$. Then the sequence of KL-divergence to $\mathbb{P}^{SB}$ is non-increasing,*

$$D_{KL}(\Lambda^{(2n)} \| \mathbb{P}^{SB}) \geq D_{KL}(\Lambda^{(2n+1)} \| \mathbb{P}^{SB}) \geq D_{KL}(\Lambda^{(2n+2)} \| \mathbb{P}^{SB}).$$

*$D_{KL}(\Lambda^{(2n)} \| \mathbb{P}^{SB}) = D_{KL}(\Lambda^{(2n+1)} \| \mathbb{P}^{SB})$ if and only if $\Lambda^{(2n)} = \Lambda^{(2n+1)} = \mathbb{P}^{SB}$. Moreover, $\Lambda^{(n)}$ converges to $\mathbb{P}^{SB}$ in law as $n \to \infty$.*

## 4 METHODS

Here, we propose the Discrete Diffusion Schrödinger Bridge Matching (DDSBM) framework, which solves the SB problem in discrete state spaces with a diffusion generative modeling, supported by our theoretical background in Section 3.3. Approaches to the SB problem in continuous spaces are based on stochastic differential equations, while our method uses continuous-time Markov chains (CTMCs) in discrete state spaces. We first propose the DDSBM framework that adjusts the Iterative Markovian Fitting (IMF) to càdlàg paths in discrete spaces, whose convergence is ensured by Theorem 3.3 (Section 4.1). We then discuss how the DDSBM framework can be implemented for a graph transformation problem (Section 4.2) and introduce a graph permutation matching algorithm to reflect the permutation-invariance nature of graphs (Section 4.3).

## 4.1 ALGORITHM FOR SOLVING SCHRÖDINGER BRIDGE PROBLEM ON DISCRETE STATES

The IMF algorithm begins with a random initial coupling $\pi$ such that $\pi_0 = \Gamma$ and $\pi_\tau = \Xi$. Following the definition of the reciprocal class in Definition 3.1, we construct the initial reciprocal bridge to obtain the measure $\Lambda^{(0)}$. Given a reciprocal measure $\Lambda^{(2n)} \in \mathcal{R}(\mathbb{Q})$, the next step is to compute its Markov projection $M^{(2n+1)} := \Pi_{\mathcal{M}}(\Lambda^{(2n)})$. The exact form of the transition rate for $M^{(2n+1)}$ is provided in Proposition B.2. In practice, the transition rate is approximated by a neural network.

To achieve this, it first samples pairs $(x_0, x_\tau)$ from $\Lambda_{0,\tau}^{(2n)}$ and samples intermediate states $x_t$ by constructing the bridge $\mathbb{Q}(\cdot|X_0, X_\tau)$. The sampled pairs $(x_t, x_\tau)$ are distributed according to $\Lambda_{t,\tau}^{(2n)}$. Using these realizations, the rate matrix of $M^{(2n+1)}$ is approximated by parameterized Markov measure $M^\theta$, by minimizing the following loss function:

$$\mathcal{L}(\theta) = \int_0^\tau \mathbb{E}_{\Lambda_{t,\tau}^{(2n)}} \left[ (A_t^{\mathbb{Q}\cdot|\tau} - A_t^{M^\theta})(X_t, X_t) + \sum_{y \neq X_t} A_t^{\mathbb{Q}\cdot|\tau} \log \frac{A_t^{\mathbb{Q}\cdot|\tau}}{A_t^{M^\theta}}(X_t, y) \right] dt, \quad (5)$$

where $A_t^{\mathbb{Q}\cdot|\tau}$ denotes the generator of pinned down process of $\mathbb{Q}(\cdot|X_\tau)$, explicitly defined in Lemma B.4. From the approximated generator $A_t^{M^\theta}$, we can sample $x_\tau$ given $x_0$, where $(x_0, x_\tau) \sim M_{0,\tau}^\theta \approx M_{0,\tau}^{(2n+1)}$. Note that, until the sequence of measures converges, the new joint coupling $M_{0,\tau}^\theta \approx M_{0,\tau}^{(2n+1)}$ will differ from the previous one $\Lambda_{0,\tau}^{(2n)}$.

Once the Markov measure $M^{(2n+1)}$ is obtained, we proceed to compute the corresponding reciprocal measure $\Lambda^{(2n+1)}$ through reciprocal projection, $\Lambda^{(2n+1)} := \Pi_{\mathcal{R}(\mathbb{Q})}(M^{(2n+1)})$. In theory, as shown in Proposition B.2, the time marginal distributions are preserved under Markov projection, meaning that $\Lambda_t^{(2n)} = M_t^{(2n+1)}$ for all $t \in [0, \tau]$. However, in practice, since the Markov projection is approximated using a neural network, repeatedly applying this approximation can lead to an accumulation of errors in the time marginals. Such accumulated errors may violate the marginal condition of the SB problem, particularly leading to a potential failure in satisfying the terminal condition $\mathbb{P}_\tau = \Xi$.

To compensate these errors, the next Markov measure $M^{(2n+2)} := \Pi_{\mathcal{M}}(\Lambda^{(2n+1)})$ is approximated in a time-reversal way (see Proposition B.8). Based on the time-symmetric nature of Markov measures, we can leverage the time-reversed generator $\tilde{A}_t^{M^{(2n+2)}}$, which enables the sampling of $x_0$ conditioned on $x_\tau$, where $(x_0, x_\tau) \sim M_{0,\tau}^{(2n+2)}$. The approximation of $\tilde{A}_t^{M^{(2n+2)}}$ is achieved by minimizing the following loss function:

$$\mathcal{L}(\phi) = \int_0^\tau \mathbb{E}_{\Lambda_{0,t}} \left[ (\tilde{A}_t^{\mathbb{Q}\cdot|0} - \tilde{A}_t^{M^\phi})(X_t, X_t) + \sum_{y \neq X_t} \tilde{A}_t^{\mathbb{Q}\cdot|0} \log \frac{\tilde{A}_t^{\mathbb{Q}\cdot|0}}{\tilde{A}_t^{M^\phi}}(X_t, y) \right] dt, \quad (6)$$

where $\tilde{A}_t^{\mathbb{Q}\cdot|0}$ denotes the time-reversal generator of the pinned down process of $\mathbb{Q}(\cdot|X_0)$, and $\phi$ represents the parameters of the neural network approximating the time-reversed generator $\tilde{A}_t^{M^\phi}$.

In this manner, the iterative Markov projection following the reciprocal projection is performed alternately in a forward and backward (time-reversal) fashion (see Algorithm 1). Finally, this process yields a sequence of measures $(\Lambda^{(n)})_{n \in \mathbb{N}}$ and $(M^{(n)})_{n \in \mathbb{N}^+}$, which converge to $\mathbb{P}^{\mathrm{SB}}$ in theory.

## 4.2 APPLY DDSBM ON GRAPHS

We present a method for applying the previously described solution to graph transformation. Here, we represent a graph as $\mathbf{G} = (\mathbf{V}, \mathbf{E})$, where $\mathbf{V} = (v^{(i)})_i$ and $\mathbf{E} = (e^{(ij)})_{ij}$ denote node and edge features, respectively. In a molecular graph, for example, the node and edge features correspond to atomic types and edge features, respectively. Here, $\mathbf{V}$ and $\mathbf{E}$ are modeled as products of categorical random variables.

As the reference process, we define a jump process in which the nodes and edges of the graph vector change discretely, assuming that all nodes and edges are independent. Therefore, the transition

probability $P$ and the rate $A$ of the reference process is described as follows:

$$P_{s:t}^{\mathbf{G}}(\mathbf{G}_1, \mathbf{G}_2) = \prod_i P_{s:t}^V(v_1^{(i)}, v_2^{(i)}) \prod_{i,j} P_{s:t}^E(e_1^{(ij)}, e_2^{(ij)}),$$

$$\partial_t P_{s:t}^{(\cdot)}(x,y) = \sum_{z \in \mathcal{X}^{(\cdot)}} P_{s:t}^{(\cdot)}(x,z) A_t^{(\cdot)}(z,y), \tag{7}$$

$$A_s^{(\cdot)}(x,y) = \partial_t P_{s:t}^{(\cdot)}(x,y)\big|_{t=s},$$

where $\cdot$ denotes $V$ or $E$, $\mathbf{G}_1$ denotes $(\mathbf{V}_1, \mathbf{E}_1) = \left((v_1^{(i)}), (e_1^{(ij)})\right)$, $\mathbf{G}_2$ denotes $(\mathbf{V}_2, \mathbf{E}_2) = \left((v_2^{(i)}), (e_2^{(ij)})\right)$, and $\mathcal{X}^V$ and $\mathcal{X}^E$ denote the state space of the nodes and edges, respectively. More specifically, we use a monotonically decreasing function for the signal to noise ratio, $\bar{\alpha} : [0, \tau] \to (0, 1]$, in which the transition rate is defined as:

$$A_t^{(\cdot)}(x,y) = \partial_t(\ln \bar{\alpha}(t))\left(\delta_{xy} - \mathbf{m}^{(\cdot)}(y)\right), \tag{8}$$

where $\delta_{xy}$ denotes the Kronecker delta, and $\mathbf{m}^{(V)}$ and $\mathbf{m}^{(E)}$ denote the prior distribution of nodes and edges as proposed in the previous discrete diffusion work (Vignac et al., 2022). According to the Kolmogorov equation, we get the transition probability as,

$$P_{s:t}^{(\cdot)}(x,y) = \frac{\bar{\alpha}(t)}{\bar{\alpha}(s)}\delta_{xy} + \left(1 - \frac{\bar{\alpha}(t)}{\bar{\alpha}(s)}\right)\mathbf{m}^{(\cdot)}(y). \tag{9}$$

Note that the choice of $\mathbf{m}(\cdot)$ as uniform is associated to the diffusion on the state space $\mathcal{X}$, where the $\mathcal{X}$ is considered fully-connected graph. Moreover, the stationary distribution of the associated generator always becomes $\mathbf{m}(\cdot)$. Although non-uniform choice of $\mathbf{m}(\cdot)$ breaks the diffusion formulation on $\mathcal{X}$, it does not harm SB formulation.

The reference process defined above, $\mathbb{Q}$, is a permutation-equivariant process since each node and edge changes independently. Formally,

$$\mathbb{Q}_{s:t}(\mathbf{G}_1, \mathbf{G}_2) = \mathbb{Q}_{s:t}(\sigma\mathbf{G}_1, \sigma\mathbf{G}_2), \quad \forall \sigma \in S_n, \ 0 \le s \le t \le \tau, \tag{10}$$

where $S_n$ is the permutation group for $n$ elements. Although we represented a graph as a vectorized form, $\mathbf{G}$, the graph itself must be permutation-invariant. To reflect this, we define a graph with $n$ nodes as a set of graph vectors,

$$\mathcal{G} := \{\sigma(\mathbf{G}) | \sigma \in S_n\}, \tag{11}$$

where $\mathbf{G}$ is an arbitrary vectorization of $\mathcal{G}$. The nature choice of a joint probability between $\mathcal{G}$ and $\mathbf{G}$ is to define it with an indicator function,

$$p(\mathcal{G}_i, \mathbf{G}_j) = p(\mathbf{G}_j|\mathcal{G}_i)p(\mathcal{G}_i), \ p(\mathbf{G}_j|\mathcal{G}_i) = p_{ij}\mathbf{1}_{\mathcal{G}_i}(\mathbf{G}_j) = \begin{cases} p_{ij} & \text{if } \mathbf{G}_j \in \mathcal{G}_i, \\ 0 & \text{otherwise}, \end{cases} \ p_{ij} = \frac{1}{|\mathcal{G}_i|}, \tag{12}$$

where $|\mathcal{G}_i|$ is the number of distinct graph vectors in a graph $\mathcal{G}_i$. Now, we can define the transition kernel between graphs with an associated stochastic process of graph vectors as follows:

$$\tilde{Q}(\mathcal{G}'|\mathcal{G}) := \sum_{\sigma \in S_{\mathcal{G}'}, \mu \in S_{\mathcal{G}}} p(\mathcal{G}'|\sigma(\mathbf{G}'))\mathbb{Q}_{0:\tau}(\sigma(\mathbf{G}'), \mu(\mathbf{G}))\, p(\mu(\mathbf{G})|\mathcal{G}) \tag{13}$$

$$= \sum_{\sigma \in S_{\mathcal{G}'}, \mu \in S_{\mathcal{G}}} \mathbf{1}_{\mathcal{G}'}(\sigma(\mathbf{G}'))\, \mathbb{Q}_{0:\tau}(\sigma(\mathbf{G}'), \mu(\mathbf{G}))\frac{1}{|\mathcal{G}|} \tag{14}$$

$$= \sum_{\omega \in S_{\mathcal{G}'}} \mathbb{Q}_{0:\tau}(\omega(\mathbf{G}'), \mathbf{G}), \tag{15}$$

where $S_{\mathcal{G}}$ and $S_{\mathcal{G}'}$ are permutation groups of $\mathcal{G}$ and $\mathcal{G}'$, and $\mathbf{G}$ and $\mathbf{G}'$ are arbitrary graph vectors of $\mathcal{G}$ and $\mathcal{G}'$, respectively. Equation (15) connects the transition kernels on graphs and their vectorized forms.

### 4.3 GRAPH PERMUTATION MATCHING

The transition probability of the reference process depends on graph permutations (see Equation (7)), so graph permutation matching must be considered beforehand. Although this issue does not affect the sampling phase, it becomes problematic when computing the likelihood of the reference process for two given graphs $\mathcal{G}$ and $\mathcal{G}'$, or when constructing a reciprocal process, which is a mixture of Markov bridges between the two graphs (see Appendix D).

One way to handle this is selecting a graph permutation that maximizes the likelihood under the reference process. Finding the optimal permutation can be formulated as a quadratic assignment problem (QAP), where the objective is to minimize the negative log-likelihood (NLL), consisting of the sum of the NLLs for both the nodes and edges. While exact solutions are possible through mixed integer programming, the problem is NP-hard, so alternative methods are preferred. Specifically, we employ a max-pooling algorithm by Cho et al. (2014), which is an approximation method categorized by continuous relaxation. After obtaining an approximate solution, we use the Hungarian algorithm (Kuhn, 1955), implemented in the Pygmtools (Wang et al., 2024), to discretize the assignment vector to the final solution. We observed that the graph matching is highly accurate in molecular graph matching (see Appendix D.5). The details of the algorithm are described in Appendix D.4. We utilized the graph matching algorithm for every reciprocal bridge construction and likelihood computation.

Recall that the SB problem could be interpreted to the EOT problem, where the cost function corresponds to the NLL. Thus, defining the reference process can be interpreted as defining a distance (cost) on the set of graphs. Interestingly, we found that the likelihood of optimal permutation is interpreted as the graph edit distance (see Appendix D.6). This leads to the conclusion that the SB problem, with the $\mathbb{Q}$ as Equation (9), is analogous to an OT problem over the metric space of graphs equipped with graph edit distance (GED) as metric, where the GED computation is well known to be NP-hard problem.

## 5 RESULTS AND DISCUSSIONS

Here, we demonstrate the effectiveness of the Discrete Diffusion Schrödinger Bridge Matching (DDSBM) framework on graph transformation tasks. Specifically, we apply DDSBM to a chemical domain, where the graph transformation task is nontrivial due to additional constraints imposed by molecular graphs and their associated properties. We conducted experiments on two different molecule datasets: ZINC250K (Kusner et al., 2017) and Polymer (St. John et al., 2019). Throughout this section, we first elaborate on the common experimental setups and metrics for evaluation. The second and third sections provide a detailed analysis of ZINC250K and Polymer experiments. Also, we discuss ablation studies for graph matching algorithms and the initial data coupling method to analyze their effects on the overall performance of DDSBM and the convergence of the Iterative Markovian Fitting (IMF) iteration (see Appendix F.2). Furthermore, for readers who might be interested in the performance of DDSBM on unconditional graph generation tasks, i.e. synthetic graph data such as Community-20, Planar, or the small molecule graph dataset, QM9, we present the DDSBM results for them in Appendix G.

### 5.1 EXPERIMENTAL SETUP AND METRICS

**Experimental Setup.** To train the models on graph transformation problems, an initial coupling between two distributions is necessary. We randomly coupled the data of initial and terminal distributions to obtain paired data. The molecular pairs are divided into training and test datasets in a ratio of 8:2. All the following reported values are average of three independent training runs with different random seeds. For standard deviation, please refer to Appendix F.1. Also, detailed explanations about model architectures and hyperparameters can be found in Appendix E.

**Metrics.** By definition of SB shown in Equation (1), both joint and marginal distributions at each side must be examined to assess the degree to which the SB has been successfully achieved. We evaluate these distributions from two perspectives: graph structural properties and molecular properties. Analysis on graph structural properties examines whether our model could capture the data-intrinsic features and the optimality of transporting between given data distributions. Since the graph structural features might exhibit weak correlation with the molecular properties, we also analyze the

Table 1: **Distribution shift performance on ZINC.** Reference refers to metrics from the initial coupling, used as a standard to evaluate each model's graph translation. For both AtomG2G and HierG2G, we've excluded the generated molecules that are too large with more atoms than the maximum number of atoms in our dataset for computing metrics other than validity. ↑ and ↓ denote higher and lower values are better, respectively. The best performance is highlighted in bold.

| Model | Type | Val.(↑) | Uniq.(↑) | Nov.(↑) | NLL(↓) | NSPDK(↓) | LogP $W_1$(↓) | QED MAD(↓) | SAscore MAD(↓) | FCD(↓) |
|---|---|---|---|---|---|---|---|---|---|---|
| Reference[1] | - | - | - | - | 360.862 | 1.47e-4 | 2.007 | 0.153 | 0.595 | 4.807 / 0.279 |
| AtomG2G | Latent | 99.9 | 64.4 | 99.3 | 355.025 | 9.70e-3 | 0.162 | 0.143 | 0.697 | 5.019 |
| HierG2G | Latent | **100.0** | 73.7 | 99.5 | 344.458 | 2.10e-2 | **0.113** | 0.146 | 0.687 | 5.742 |
| DBM | Bridge | 87.6 | **100.0** | **100.0** | 288.572 | 8.04e-4 | 0.150 | 0.141 | 0.608 | 1.046 |
| DDSBM | Schrödinger Bridge | 94.8 | **100.0** | 99.9 | **160.461** | **7.30e-4** | 0.139 | **0.120** | **0.402** | **0.833** |

[1] NLL, $W_1$, and MADs were calculated using random pairs from the test set. Two FCD values are provided: the first compares the initial molecules in the test set with the terminal molecules in the training set, and the second compares the terminal molecules in both sets. Also, the reference NSPDK is computed with the terminal molecules from training and test sets.

molecular properties to validate the effectiveness of the DDSBM framework on the molecular optimization tasks. Besides the basic metrics for molecular generative models—validity (**Val.**), uniqueness (**Uniq.**), and novelty (**Nov.**)— we evaluate molecular properties with Wasserstein-1 distance ($W_1$) and Fréchet ChemNet Distance (**FCD**). On ZINC250K, mean absolute differences (MAD) of drug-likeness (**QED**), and synthetic accessibility (**SAscore**) are also evaluated. For graph structural properties, we analyze negative log-likelihood (**NLL**) and neighborhood subgraph pairwise distance kernel (**NSPDK**). More details about the metrics are explained in Appendix E.4.

## 5.2 SMALL MOLECULE TRANSFORMATION

First, we validate our proposed methods on the SB problem between two molecule distributions constructed from the standard ZINC250K dataset. We constructed two sets of molecules whose $\log P$ values are largely different. Molecules from the ZINC250K dataset were randomly selected and divided into two sets whose $\log P$ values follow the Gaussian distributions centered at 2 and 4 with variance of 0.5, respectively (see Appendix E.2 for more details). We compared our methods with three baseline models that perform graph-to-graph translation. AtomG2G and HierG2G are latent-based models that encode an input graph into a latent vector and decode it into another graph (Jin et al., 2020). Diffusion Bridge Model (DBM) is a bridge model trained with the same reference process as DDSBM, which is equivalent to the first Markov projection in DDSBM. We refer readers to Appendix E.3 for more details about the baselines and implementation of our models. We note that, for these three baseline models, the initial coupling is utilized during the whole training procedure without change, while DDSBM dynamically alters the training data pair by IMF.

Table 1 shows overall results of our method compared to the three baseline models. DDSBM consistently outperforms the other models in terms of both NSPDK, NLL, and FCD. The lower NSPDK and FCD suggest that DDSBM-generated molecules are more similar to those in the target dataset, while the minimal NLL indicates that DDSBM applies minimal structural change on initial graphs. This result demonstrates that DDSBM achieves more *optimal* graph transformation between fixed initial and terminal distributions. When it comes to the model type, bridge models shows superior in FCD, NLL, and NSPDK, compared to the latent-based models. The bridge models, DBM and DDSBM, dynamically transform a graph based on the reference dynamics, favoring the retention of the given structure, whereas whole-graph reconstruction using a latent vector does not. This leads to lower NLL values for the bridge models, which is ensured by the Definition 3.2. Additionally, the lower NSPDK and FCD values of the bridge models highlight that constructing a dynamic bridge within the graph domain enhances the performance of distribution learning for the target distribution. The latent models have a higher validity, but given that their uniqueness is significantly lower compared to the bridge models, which achieve 100% uniqueness, the bridge models are better suited for tasks that require distinct and diverse molecule generation. Furthermore, we attribute the best performance of DDSBM to the gradual updating of training pairs, which become more similar than the random initial pairs, simplifying the graph transportation process.

Next, we analyze the molecular properties of the source and generated molecules. DDSBM resulted in the lowest MAD in QED and SAscore, meaning the deviation of molecular properties other than $\log P$ is the smallest for DDSBM. It is noteworthy that DDSBM achieves minimal changes in various molecular properties despite being trained solely to optimize graph transformations with minimal

structural alterations, without explicit knowledge on molecular properties. Meanwhile, the latent-based models exhibited larger MAD values in QED and SAscore, indicating that they are vulnerable to losing other properties of the initial molecules. This can be inferred from the result of HierG2G; although HierG2G achieves the lowest $W_1$ value in $\log P$ so that it modulates $\log P$ the best to the desired degree, its much larger NLL value suggests that it could generate a graph with less consideration of the initial graph constraint, as illustrated in Figure 14. For a better understanding of the overall results, we provide the corresponding distributions of properties of the models in Figure 10.

Despite the promising results, we observe that all models except DDSBM have a high reliance on a predefined initial coupling. Thus, we conducted additional experiments using pseudo-optimal initial coupling based on Tanimoto similarity, which is detailed in Appendix F.2.2. Interestingly, we found that introducing the pseudo-optimal initial coupling not only accelerated the training of DDSBM in practice but also allowed it to achieve performance on par with the previous results.

## 5.3 POLYMER GRAPH TRANSFORMATION

The Polymer dataset (St. John et al., 2019) consists of 91,000 monomer molecules with their optical excitation energies (GAPs) obtained by time-dependent density functional theory calculations. We reconstructed the Polymer dataset for a transport problem between two sets of molecules with distinct GAPs, corresponding to green and blue optical colors, respectively. The reconstructed dataset contains 7,603 molecular pairs. This task is considered as a more challenging application because the relationship between the graph structure and the target GAP property is highly non-linear, making it hard to predict the effect of specific structural changes on the GAP. In this context, we apply our DDSBM model to find the optimal transformation between the two sets of molecules.

The performance of DDSBM is compared to DBM, which serves as a baseline. The GAPs of the molecules generated by the models were obtained using the pre-trained MolCLR model (Wang et al., 2022), which has a mean absolute error (MAE) of 0.027 eV for the GAP prediction. Table 2 shows the overall results of our method on the Polymer dataset. DDSBM outperforms DBM on most of the metrics evaluated, which is consistent with the results of the experiments on ZINC. In terms of validity, DBM shows significantly lower scores, which contrasts with the results observed on the ZINC250K dataset. This can be attributed to the characteristics of the molecules in the Polymer dataset, which have relatively large sizes and multiple ring structures. In this context, minimal transformations are advantageous for achieving high validity, and DBM may have struggled to learn these changes from the randomly paired data. Examples of the generated molecular graphs are visualized in Figure 15.

Table 2: **Distribution shift performance on Polymer.** Reference refers to metrics from the initial coupling, used as a standard to evaluate each model's graph translation. ↑ and ↓ denote higher and lower values are better, respectively. The best performance is highlighted in bold.

| Model | Type | Val.(↑) | Uniq.(↑) | Nov.(↑) | FCD(↓) | NLL(↓) | NSPDK(↓) | GAP $W_1$(↓) |
|---|---|---|---|---|---|---|---|---|
| Reference[1] | - | - | - | - | 1.469 / 0.384 | 749.800 | 5.64e-4 | 0.312 |
| DBM | Bridge | 43.4 | **99.8** | **97.4** | 2.230 | 580.415 | 5.82e-3 | 0.249 |
| DDSBM | Schrödinger Bridge | **97.4** | 94.5 | 71.3 | **1.074** | **212.047** | **4.18e-3** | **0.127** |

[1] NLL and $W_1$ were calculated using random pairs from the test set. Two FCD values are provided: the first compares the initial molecules in the test set with the terminal molecules in the training set, and the second compares the terminal molecules in both sets. Also, the reference NSPDK is computed with the terminal molecules from training and test sets.

## 6 DISCUSSION

In this paper, we propose Discrete Diffusion Schrödinger Bridge Matching (DDSBM), a novel framework utilizing continuous-time Markov chains to solve the SB problem in a high-dimensional discrete state space. To this end, we extend Iterative Markovian Fitting (IMF), proving its convergence to SB. We successfully apply our framework to graph transformation, specifically for molecular optimization. Experimental results demonstrate that DDSBM effectively transforms molecules to achieve the desired property with minimal graph transformation, while retaining other features. However, the IMF requires iterative sampling from the learned Markov process, which can be more computationally intensive than simple bridge matching.

ACKNOWLEDGMENTS

This work was supported by the Korea Environmental Industry and Technology Institute (Grant No. RS202300219144), the Technology Innovation Program funded by the Ministry of Trade, Industry & Energy, MOTIE, Korea (Grant No. 20016007), and Basic Science Research Programs through the National Research Foundation of Korea funded by the Ministry of Science and ICT (Grant No. RS-2023-00257479, Grant No. 2018R1A5A1025208 and Grant No. NRF-2022M3J6A1063021).

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

## A    INTROUDCTION TO SCHRÖDINGER BRIDGE

In this section, we will begin with a thought experiment to briefly understand the Schrödinger Bridge problem (SBP), followed by a simple toy example in continuous space. Specifically, we will focus on intuitively understanding how the Iterative Markovian Fitting (IMF) method works through a toy example. Building on this, we aim to gain insight into the Schrödinger bridge problem and explore its application in discrete spaces.

### A.1    THE LAZY GAS EXPERIMENT AND THE SCHRÖDINGER BRIDGE PROBLEM

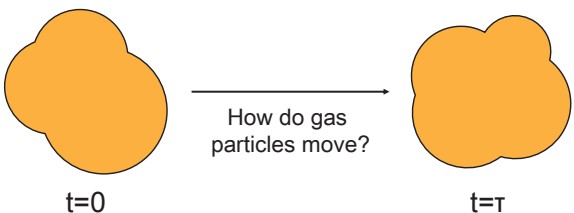

Figure 2: A schematic illustration of the lazy gas experiment

Imagine a *gas*, a collection of non-interacting particles confined to a specific region of space. At time $t = 0$, these particles are distributed according to a given distribution. By time $t = \tau$, the gas must reorganize itself to match a different distribution.

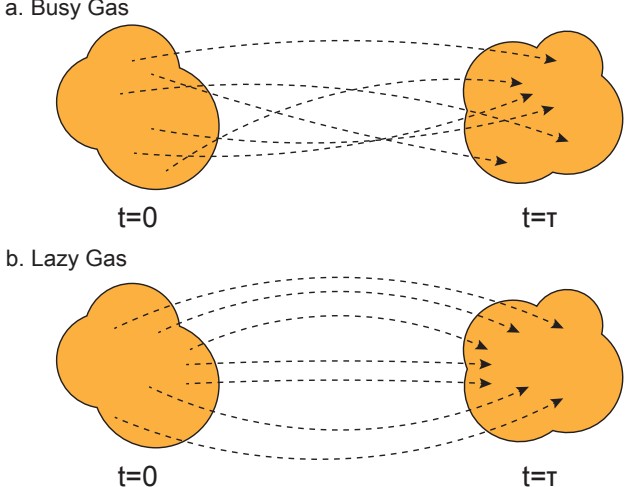

Figure 3: A schematic comparison between lazy gas and busy gas

*Busy gas* will find a way to perform the given transition regardless of the amount of work required. In contrast, *lazy gas* will seek to make the transition while minimizing the total work, following the principle of least action. This simple yet thought-provoking scenario, known as the *lazy gas experiment*, provides an intuitive starting point for understanding the SBP. Proposed by Erwin Schrödinger, the SBP seeks to identify the most *natural* probabilistic process that connects two probability distributions under certain constraints. At its core, the problem is to find a probabilistic interpolation that minimizes the cost.

In essence, the SBP addresses the following question: *Given the initial and final distributions of a stochastic process, what is the optimal way to probabilistically interpolate between these two distributions?* This process must satisfy two important principles. First, the process must exactly match the given initial and final distributions (boundary condition). Second, it must minimize some

properly defined *cost*. This cost is typically defined as the Kullback-Leibler (KL) divergence with respect to a reference process representing the underlying dynamics.

In the example of the *lazy gas experiment*, if the gas particles are moving according to the reversible Brownian motion (a Wiener process with zero drift), the cost can be defined as the total distance traveled by the particles. This principle of cost minimization ensures that the particles follow the optimal paths that most efficiently (in terms of $L^2$ norm) connect the initial and final distributions. This analogy intuitively captures the essence of the SBP's goal: *to achieve a natural and efficient transition between given marginal distributions*[1].

## A.2 SCHRÖDINGER BRIDGE WITH TOY EXAMPLE IN CONTINUOUS SPACE

To better understand the SBP in a real-world context, consider a simple toy example involving probabilistic transitions between two probability distributions in a one-dimensional continuous space. Here, we use reversible Brownian motion as the reference process. This example allows us to contrast the optimal transition process (Schrödinger bridge) between the initial and final distributions with an inefficient, suboptimal one. By visualizing these differences, we gain an intuitive understanding of the SBP. In particular, this example illustrates how the central goal of the problem - minimizing costs - is achieved in practice by reducing the Kullback-Leibler (KL) divergence relative to the reference process.

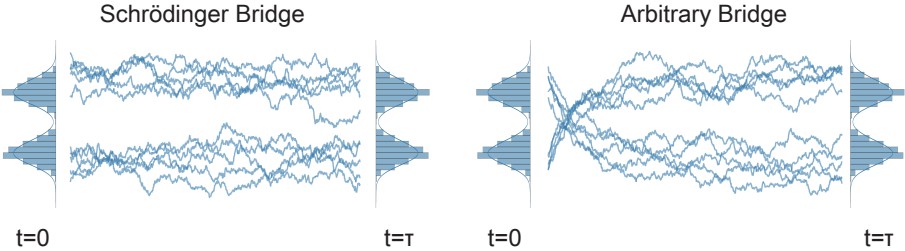

Figure 4: A schematic comparison between Schrödinger bridge and aribtrary bridge

In Figure 4, the trajectory labeled *Schrödinger Bridge* on the left represents the optimal probabilistic path connecting the two distributions. These paths are carefully structured not only to satisfy the given boundary condition, but also to adhere to optimal cost, achieving a balance between efficiency and regularity. On the right, the trajectories labeled *Arbitrary Bridge* illustrate a suboptimal transport process. While the boundary condition is still satisfied, the intermediate paths are not optimal for the given cost. In this context, the cost, defined by the $L^2$ norm, highlights the differences in the distances traveled by each point. As shown in the figure, the movements in the optimal paths are significantly more efficient compared to those in the arbitrary paths.

Considering these differences, the next question is: *how can the Schrödinger Bridge be found in practice?*

## A.3 SOLVING SCHRÖDINGER BRIDGE PROBLEM IN CONTINUOUS SPACE

Since the SBP does not admit a closed-form solution, iterative algorithms such as Iterative Proportional Fitting (IPF) and Iterative Markovian Fitting (IMF) are used to approximate it. Both methods adopt the Markovian projection for a given stochastic process to minimize the KL divergence.

IPF algorithm iteratively fits their marginal distribution to the given distribution, while preserving the reciprocal class and Markovian properties of the given process. In contrast, IMF method alternates between reciprocal projection and Markovian projection to refine the process while keeping the marginal distributions fixed.

---

[1]The experiments presented in this section are based on Villani (2009), with the associated explanations informed by Léonard (2013).

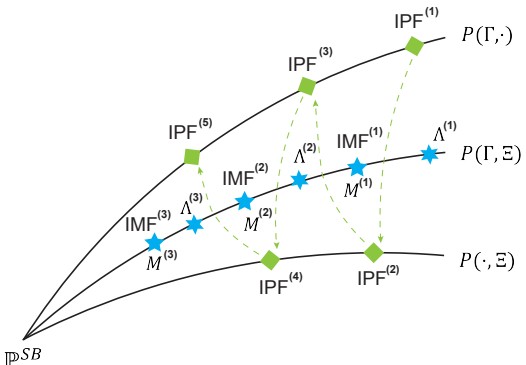

Figure 5: A schematic comparison between Iterative Proportional Fitting (IPF) and Iterative Markovian Fitting (IMF) method. The image are reproduced based on Peluchetti (2023a). In the notation, $P(\Gamma, \cdot)$ represents the collection of path measures where only the initial distribution is fixed at $\Gamma$, $P(\cdot, \Xi)$ represents the collection of path measures where only the terminal distribution is fixed at $\Xi$, and $P(\Gamma, \Xi)$ represents the collection of path measures where both the initial and terminal distributions are fixed at $\Gamma$ and $\Xi$, respectively.

At this point, reciprocal projection is the process of constructing a mixture bridge using the reference process and Doob's h-transform based on the given joint distribution. Markovian projection, on the other hand, is performed by generative modeling with neural networks to produce a Markov process similar to the given one in terms of the KL divergence. In practice, training occurs during the Markovian projection phase, which plays the same role as bridge matching in refining the probabilistic process.

In this section, we will focus on IMF since it is the main reference method for our DDSBM framework. We will explore the interchange between the two projections and the principles by which they work together to refine the given process.

First, when reversible Brownian motion is given as the reference process in one dimensional continuous space, our goal is to find the Schrödinger bridge between the marginal distributions. We start by obtaining the initial coupling between the marginal distributions. For visualization purposes, we assume an initial joint distribution in which points opposite in space are coupled.

Next, we use reciprocal projection to construct a reciprocal mixture bridge connecting the marginal distributions (Figure 6a). While this bridge has the reciprocal property, the Markov property is generally lost. Moreover, it shows significant differences from the joint distribution of the SB, which resembles the left side of Figure 6c.

We then apply Markov projection to this mixture bridge to derive the Markov process that is closest in terms of KL divergence (Figure 6b). The obtained process reduces the KL divergence compared to the previous process, bringing it closer to the Schrödinger bridge (see Figure 5). In addition, the joint distribution becomes more similar to the SB joint distribution than the initial coupling, as reflected by the reduced cost.

By iteratively repeating this process, the obtained Markov process converges to the Schrödinger bridge, as guaranteed by the convergence theorem (Shi et al., 2024; Peluchetti, 2023a). Through IMF iterations, we progressively refine the process, eventually arriving at the Schrödinger bridge (Figure 6c).

## A.4 SCHRÖDINGER BRIDGE PROBLEM IN DISCRETE SPACE

So far, we have described the SBP in the context of continuous space. Since the Wiener process used in continuous space cannot reflect the characteristics of discrete space well, we need to introduce a new diffusion process to solve SBP in discrete space.

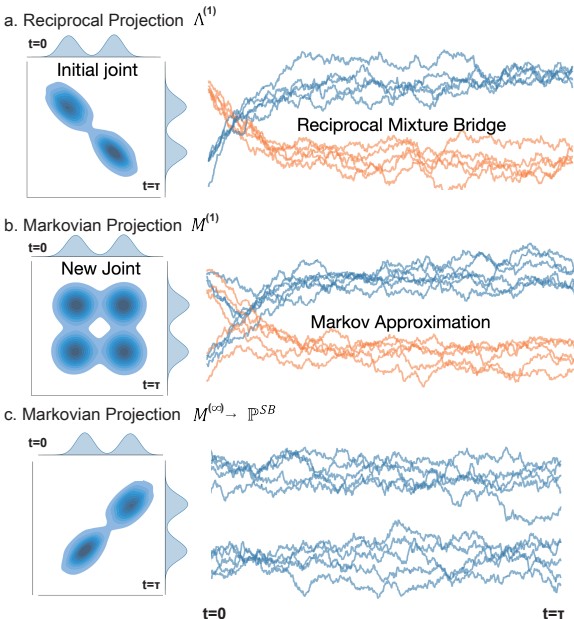

Figure 6: A schematic illustration of Iterative Markovian Fitting method.

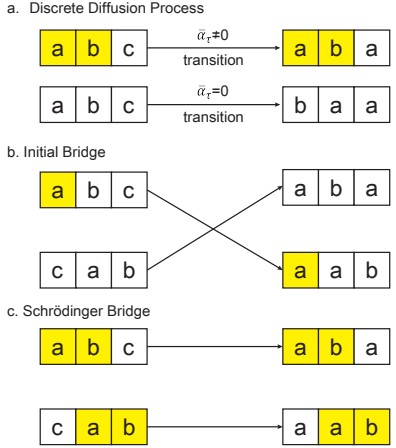

Figure 7: A schematic illustration of the discrete diffusion process and a comparison between the initial bridge and the Schrödinger bridge

Let us consider discrete random variables with three categories ($a$, $b$, $c$). In this context, we can adopt the discrete diffusion process proposed in Austin et al. (2021), such as the uniform transition process. Unlike processes where all information about a state is lost ($\bar{\alpha}_\tau = 0$), we define a discrete diffusion process that retains partial information about its initial state ($\bar{\alpha}_\tau \neq 0$). This approach is visualized in Figure 7a, highlighting the retained structure.

In the case of continuous space, the cost is typically represented by the $L^2$ norm. In discrete space, however, the cost can be thought of as the dissimilarity between discrete states. Thus, minimizing the $L^2$ norm in the continuous space is equivalent to minimizing the negative log-likelihood (NLL) of the joint distribution in the discrete state space, given by the reference diffusion process we used. The $\bar{\alpha}_\tau$ can be interpreted as a parameter that indicates how much of the original structure is retained

during the diffusion process. If the original discrete diffusion process ($\bar{\alpha}_\tau = 0$) is used, the NLL assigns the same value to all pairs, making it impossible to define the SBP.

By solving the SBP using our DDSBM framework, it is possible to refine from an initial non-optimal coupling to the optimal coupling, i.e. the Schrödinger bridge, from the perspective of NLL. As shown in Figure 7b and Figure 7c, this procedure can be understood as finding the coupling that best preserves the current state. This insight is a major reason for introducing DDSBM into graph domains, especially for molecular optimization.

Figure 8 illustrates the initial bridge and the Schrödinger bridge in the graph domain. The initial bridge shows a non-optimal coupling between two graphs, where the structural consistency is not well preserved. In contrast, the Schrödinger bridge shows an optimal coupling that preserves the core structure of the graph while transforming to the target graph, reflecting the principle of minimal change.

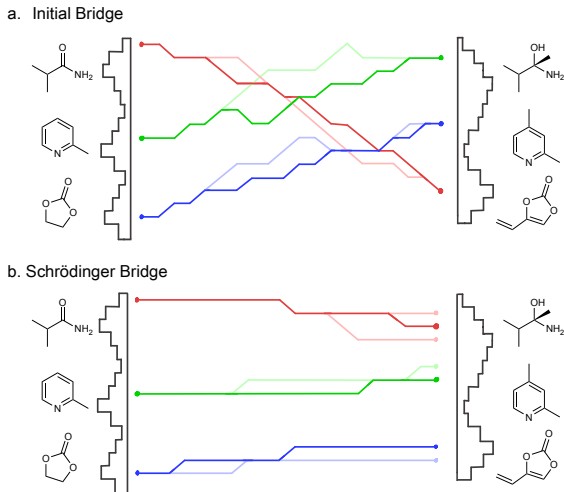

Figure 8: A schematic comparison between the initial bridge and the Schrödinger Bridge in graph domain

# B  PROPOSITIONS AND PROOF

## B.1  NOTATIONS

In this section, we introduce the notations that will be used throughout the proofs of the propositions. $\Omega = D([0,\tau], \mathcal{X})$ denotes the space of all left-limited and right-continuous (càdlàg) paths over a finite space $\mathcal{X}$. We assume the state space $\mathcal{X}$ has connected finite graph structure (fully-connected graph), which imply it becomes metric space with graph distance metric $d_{\mathcal{X}}(\cdot, \cdot)$. Accordingly, the sample space $\Omega$ is equipped with Skorokhod topology with Skorokhod metric $d_{\Omega}(\cdot, \cdot)$, and associated Borel $\sigma$-algebra. $X = (X_t)_{t \in [0,\tau]}$ denotes the canonical process given by:

$$X_t(\omega) = \omega_t, \quad t \in [0,\tau], \quad \omega = (\omega_s)_{s \in [0,\tau]}.$$

The reference measure $\mathbb{Q}$ is an irreducible Markov measure on $\Omega$ with its associated canonical filtration. Assume that the transition probability of $\mathbb{Q}$, denoted by $P_{s:t}(x, y)$, from $(s, x) \in [0, \tau] \times \mathcal{X}$ to $(t, y) \in [0, \tau] \times \mathcal{X}$, is continuous and differentiable over time. The measure is generated by the transition rate function $A_s(x, y)$, which gives the rate of transition from $x \in \mathcal{X}$ to $y \in \mathcal{X}$ at time

$s \in (0, \tau)$, and satisfies the Kolmogorov forward equation:

$$\frac{\partial P_{s:t}(x, y)}{\partial t} = \sum_{z \in \mathcal{X}} P_{s:t}(x, z) A_t(z, y), \tag{16}$$

$$A_s(x, y) = \left[ \frac{\partial P_{s:t}(x, y)}{\partial t} \right]_{t=s}.$$

We also assume that $\mathbb{Q}$ can construct a bridge $\mathbb{Q}(\cdot | X_0 = x, X_\tau = y)$ for all $x, y \in \mathcal{X}$. For any Markov measure $M$, the corresponding generator is denoted as $A^{(M)}$.

## B.2 Theorem B.1

**Theorem B.1.** *(Uniqueness of the Schrödinger Bridge Solution)*
*If the reference process $\mathbb{Q}$ is Markov, then under mild conditions the Schrödinger Bridge solution $\mathbb{P}^{SB}$ exists and is unique. Furthermore, the solution is mixture with static Schrödinger Bridge solution represented as $\mathbb{P}^{SB} = \mathbb{P}^{SB}_{0,\tau} \mathbb{Q}_{\cdot | 0,\tau} \in \mathcal{R}(\mathbb{Q})$, and also it is in $\mathcal{M}$. Conversely, a process $\mathbb{P} = \mathbb{P}_{0,\tau} \mathbb{Q}_{\cdot | 0,\tau}$ is Markov if and only if $\mathbb{P} = \mathbb{P}^{SB}$.*

*Proof.* This is direct consequence of Theorem 2.12 of (Léonard, 2013). $\qquad\square$

## B.3 Markov projection

**Proposition B.2.** *(solution of Markov projection)*
*Let $M^* = \Pi_{\mathcal{M}}(\Lambda)$, $\Lambda \in \mathcal{R}(\mathbb{Q})$ and the generator of $\mathbb{Q}$ be $A_t(x, y)$ with transition probability $P_{s:t}(x, y)$. Under mild assumptions, the generator of the Markov measure $M^*$ becomes*

$$A_t^{(M^*)}(X_t, y) = \mathbb{E}_{\Lambda_{\tau|t}} \left[ A_t(X_t, y; X_\tau) \Big| X_t \right],$$

$$A_t(x, y; z) = A_t(x, y) \frac{P_{s:\tau}(y, z)}{P_{s:\tau}(x, z)} - \delta_{xy} \sum_u A_t(y, u) \frac{P_{t:\tau}(u, z)}{P_{t:\tau}(x, z)},$$

*, where $z \in \mathcal{X}$.*

*The reverse KL-divergence is*

$$D_{KL}(\Lambda \| M^*) = \int_0^\tau \mathbb{E}_{\Lambda_{0,t}} \left[ (A_t^{(\Lambda_{|0})} - A_t^{(M^*)})(X_t, X_t) + \sum_{y \neq X_t} A_t^{(\Lambda_{|0})} \log \frac{A_t^{(\Lambda_{|0})}}{A_t^{(M^*)}}(X_t, y) \right] dt,$$

*where the $A^{\Lambda_{|0}}$ is the generator for the conditioned measure $\Lambda_{|0}$ which is defined as*

$$A_t^{\Lambda_{|0}}(X_t, y) = \mathbb{E}_{\Lambda_{\tau|0,t}} \left[ A_t(X_t, y; X_\tau) \Big| X_t, X_0 \right].$$

*Moreover, for any $t \in [0, \tau], \Lambda_t = M_t^*$.*

### B.3.1 KL-divergence of Markov measure

Consider two Markov path measure $\tilde{M} \ll M$. Based on the Girsanov's formula, we can express the Radon-Nikodym derivative as follow:

$$\frac{dM}{d\tilde{M}}(\omega) = \frac{dM_0}{d\tilde{M}_0}(\omega_0) \exp \left( \int_0^\tau \log \frac{A_t^M}{A_t^{\tilde{M}}}(\omega_{t-}, \omega_t) dN_t(\omega) + \int_0^\tau (A_t^M - A_t^{\tilde{M}})(\omega_t, \omega_t) dt \right),$$

where $N_t(\omega)$ is the number of jumps of the path $\omega$ up to time $t$ and $\omega_{t-}$ is the left limit of the path at time $t$[2]. Note that, due to the compactness of time interval, the number of jumps of each path in

---

[2]See Chazottes et al. (2006) or Appendix 1 of Kipnis & Landim (2013)

$\Omega$ is at most finite, and thus $N_t(\omega)$ is bounded. Also, the escape rate of the state $x$ associated to $M$ is $-A_t^M(x,x)$. Thus, we can construct a natural martingale[3]

$$N_t + \int_0^t A_s(\omega_s, \omega_s)ds,$$

which is zero-mean process.

For any continuous and bounded function $\phi : \mathcal{X} \to \mathbb{R}$, we can change the integrator $dN_t$ as follow:

$$\mathbb{E}_M\left[\int_0^t \phi(\omega_s)dN_s\right] = \mathbb{E}_M\left[\int_0^t -\phi(\omega_s)A_s^M(\omega_s, \omega_s)ds\right] = \int_0^t \mathbb{E}_{x\sim M_s}\left[-\phi(x)A_s^M(x,x)\right]ds.$$

The KL-divergence is expectation of logarithm of Radon-Nikodym derivative, which leads to:

$$D_{\mathrm{KL}}(M\|\tilde{M}) = D_{\mathrm{KL}}(M_0\|\tilde{M}_0) + \mathbb{E}_M\left[\int_0^\tau \log\frac{A_t^M}{A_t^{\tilde{M}}}(\omega_{t-},\omega_t)dN_t(\omega) + \int_0^\tau (A_t^M - A_t^{\tilde{M}})(\omega_t, \omega_t)dt\right],$$

$$= D_{\mathrm{KL}}(M_0\|\tilde{M}_0) + \int_0^\tau \mathbb{E}_{x\sim M_s}\left[-A_s^M(x,x)\sum_{y\neq x}p_s(x,y)\log\frac{A_s^M}{A_s^{\tilde{M}}}(x,y)\right]ds$$

$$+ \int_0^\tau \mathbb{E}_{x\sim M_s}\left[(A_s^M - A_s^{\tilde{M}})(x,x)\right]ds,$$

where $p_s(x,y)$ is the probability of jump from $x$ to $y$ given that a jump occurs, which is $\frac{A_s^M(x,y)}{-A_s^M(x,x)}$. By applying this, we obtain KL-divergence represented solely in terms of transition rate $A^M$ and $A^{\tilde{M}}$:

$$D_{\mathrm{KL}}(M\|\tilde{M}) = D_{\mathrm{KL}}(M_0\|\tilde{M}_0) + \int_0^\tau \mathbb{E}_{x\sim M_s}\left[\sum_{y\neq x}A_s^M(x,y)\log\frac{A_s^M}{A_s^{\tilde{M}}}(x,y) + (A_s^M - A_s^{\tilde{M}})(x,x)\right]ds. \tag{17}$$

### B.3.2 KL-DIVERGENCE OF RECIPROCAL MEASURE TO MARKOV MEASURE

**Lemma B.3.** *If a reciprocal measure $\Lambda \in \mathcal{R}(\mathbb{Q})$ is conditioned on $X_0$ being a.s. constant, then the corresponding measure $\Lambda_{\cdot|0}$ is Markov. For any $M \in \mathcal{M}$ such that $M_0 = \Lambda_0$ and $\Lambda \ll M$, the KL-divergence $D_{KL}(\Lambda\|M)$ disintegrates as follow:*

$$D_{KL}(\Lambda\|M) = \mathbb{E}_{\Lambda_0}\left[D_{KL}(\Lambda_{\cdot|0}\|M_{\cdot|0})\right].$$

*Proof.* According to proposition 2.5 of Léonard et al. (2014) and lemma 1.4 of Jamison (1974), conditioning $\Lambda$ on $X_0$ not only preserves its reciprocal property, but also transforms it into a Markov process. Due to the absolute continuity together with $|\mathcal{X}| < \infty$, the KL-divergence is finite. Then, the KL-divergence can be reformulated as follows:

$$D_{\mathrm{KL}}(\Lambda\|M) = \mathbb{E}_\Lambda\left[\frac{d\Lambda}{dM}\right],$$

$$= \mathbb{E}_\Lambda\left[\frac{d\Lambda_{\cdot|0}}{dM_{\cdot|0}}\right],$$

$$= \mathbb{E}_{\Lambda_0}\mathbb{E}_{\Lambda_{\cdot|0}}\left[\frac{d\Lambda_{\cdot|0}}{dM_{\cdot|0}}\right],$$

$$= \mathbb{E}_{\Lambda_0}\left[D_{\mathrm{KL}}(\Lambda_{\cdot|0}\|M_{\cdot|0})\right].$$

$\square$

According to Lemma B.3, while a reciprocal measure $\Lambda \in \mathcal{R}(\mathbb{Q})$ is in general non-Markov, the conditional measure $\Lambda_{\cdot|0}$ is Markov. Note that we can compute the KL-divergence between two Markov measure based on Equation (17).

---

[3]See Lemma 5.1 of Kipnis & Landim (2013)

### B.3.3 GENERATOR OF CONDITIONED PROCESS

To compute $D_{\text{KL}}(\Lambda\|M)$ based on Lemma B.3, Equation (17), we need the generator of the conditioned process $\Lambda_{\cdot|0}$. Before deriving the generator of $\Lambda_{\cdot|0}$, we first consider the pinned process of $\mathbb{Q}$ conditioned on $X_\tau = z$ in prior.

**Lemma B.4.** *Let* $(X_t)_{0 \le t \le \tau}$ *be a Markov process under the reference measure* $\mathbb{Q}$ *with transition probability* $P_{s:t}(\cdot, \cdot), (s \le t)$, *and generator* $A_s(\cdot, \cdot)$. *Consider the process conditioned on* $X_\tau = z$ *with corresponding measure denoted by* $\mathbb{Q}^{(z)}$. *Then, the conditioned process is also Markov, and its generator is given by:*

$$A_s(x, y; z) = A_s(x, y)\frac{P_{s:\tau}(y, z)}{P_{s:\tau}(x, z)} - \delta_{xy}\left[\sum_u A_s(x, u)\frac{P_{s:\tau}(u, z)}{P_{s:\tau}(x, z)}\right].$$

*Proof.* The conditional probability of $X_t$ given the natural filtration $\mathcal{F}_u$ and $X_s$ with $u \le s \le t \le \tau$ under the measure $\mathbb{Q}^{(z)}$ is as follows:

$$\begin{aligned} \mathbb{Q}^{(z)}(X_t = y | X_s = x, \mathcal{F}_u) &= \mathbb{Q}(X_t = y | X_s = x, X_\tau = z, \mathcal{F}_u), \\ &= \mathbb{Q}(X_t = y | X_s = x, X_\tau = z), && (\because \mathbb{Q} \in \mathcal{M}) \\ &= \mathbb{Q}^{(z)}(X_t = y | X_s = x), \end{aligned}$$

which confirms $\mathbb{Q}^{(z)}$ is Markov.

Next, the transition probability of $\mathbb{Q}^{(z)}$, denoted by $P_{s:t}(x, y; z)$, is derived as:

$$\begin{aligned} P_{s:t}(x, y; z) &= \mathbb{Q}^{(z)}(X_t = y | X_s = x), \\ &= \mathbb{Q}(X_t = y | X_s = x)\frac{\mathbb{Q}(X_\tau = z | X_t = y)}{\mathbb{Q}(X_\tau = z | X_s = x)}, \\ &= P_{s:t}(x, y)\frac{P_{t:\tau}(y, z)}{P_{s:\tau}(x, z)}. \end{aligned}$$

Note that, we assumed the measure $\mathbb{Q}$ construct bridge everywhere, the probability ratio has finite value.

Finally, the corresponding generator $A_s(x, y; z)$ is obtained using the Kolmogorov forward equation:

$$\begin{aligned} A_s(x, y; z) &= \frac{\partial}{\partial t}P_{s:t}(x, y; z)\bigg|_{t=s}, \\ &= A_s(x, y)\frac{P_{s:\tau}(y, z)}{P_{s:\tau}(x, z)} + \delta_{xy}\left[\frac{\partial_s P_{s:\tau}(y, z)}{P_{s:\tau}(x, z)}\right], \\ &= A_s(x, y)\frac{P_{s:\tau}(y, z)}{P_{s:\tau}(x, z)} - \delta_{xy}\left[\sum_u A_s(x, u)\frac{P_{s:\tau}(u, z)}{P_{s:\tau}(x, z)}\right]. \end{aligned}$$

$\square$

We now consider the transition probability and generator of conditioned measure $\Lambda_{\cdot|0}$ of $\Lambda \in \mathcal{R}(\mathbb{Q})$.

**Lemma B.5.** *For a reciprocal measure* $\Lambda \in \mathcal{R}(\mathbb{Q})$, *the conditioned process with* $X_0 = x_0$ *is denoted by* $\Lambda_{\cdot|0=x_0}$. *The generator of* $\Lambda_{\cdot|0=x_0}$ *is given by the conditional expectation:*

$$A_s^{\Lambda_{\cdot|0=x_0}}(x, y) = \mathbb{E}_{z \sim \Lambda_{\tau|0,s}}\left[A_s(x, y; z)|X_0 = x_0, X_s = x\right],$$

*where* $A_s(x, y; z)$ *is the generator of the conditioned process of* $\mathbb{Q}$ *with* $X_\tau = z$.

*Proof.* We denote the transition probability conditioned on $X_0 = x_0$ by $P_{s:t}^{\Lambda_{\cdot|0=x_0}}$, which is computed as follows:

$$P_{s:t}^{\Lambda_{\cdot|0=x_0}}(x, y) = \int_{\mathcal{X}} \Lambda_{\cdot|0=x_0}(X_\tau = z | X_s = x)\Lambda_{\cdot|0=x_0}(X_t = y | X_s = x, X_\tau = z)dz.$$

The first term in the integrand is

$$
\begin{aligned}
\Lambda_{\cdot|0=x_0}(X_\tau = z | X_s = x) &= \Lambda(X_\tau = z | X_0 = x_0, X_s = x), \\
&= \frac{\Lambda(X_\tau = z | X_0 = x_0)\Lambda(X_s = x | X_0 = x_0, X_\tau = z)}{\Lambda(X_s = x | X_0 = x_0)}, \\
&= \frac{\nu_\tau(z; x_0)}{\nu_s(x; x_0)} P_{0:s}(x_0, x; z), \\
&= \frac{\nu_\tau(z; x_0)}{\nu_s(x; x_0)} P_{0:s}(x_0, x) \frac{P_{s:\tau}(x, z)}{P_{0:\tau}(x_0, z)}, \\
&= \frac{\nu_\tau(z; x_0)}{\nu_s(x; x_0)} \frac{\mu_s(x; x_0)}{\mu_\tau(z; x_0)} P_{s:\tau}(x, z),
\end{aligned}
$$

where $\nu_s(\cdot; x_0), \mu_s(\cdot; x_0)$ are the probability mass functions of $\Lambda_{s|0}, \mathbb{Q}_{s|0}$, respectively, with $X_0 = x_0$. Given the initial-terminal condition, $\Lambda$ is equivalent to the reference $\mathbb{Q}$ based on the definition of reciprocal class Definition 3.1. Similarly, the second term is same as $\mathbb{Q}(X_t = y | X_0 = x_0, X_s = x, X_\tau = z)$, which can be expressed as $P_{s:t}(x, y; z)$ due to the Markov property of $\mathbb{Q}$. Therefore, the transition probability is:

$$
P_{s:t}^{\Lambda_{\cdot|0=x_0}}(x, y) = \frac{\mu_s(x; x_0)}{\nu_s(x; x_0)} P_{s:t}(x, y) \int_{\mathcal{X}} P_{t:\tau}(y, z) \frac{\nu_\tau(z; x_0)}{\mu_\tau(z; x_0)} dz.
$$

Accordingly, the generator is derived as:

$$
\begin{aligned}
A_s^{\Lambda_{\cdot|0=x_0}}(x, y) &= \partial_t P_{s:t}^{\Lambda_{\cdot|0=x_0}}(x, y) \Big|_{t=s}, \\
&= \frac{\mu_s(x; x_0)}{\nu_s(x; x_0)} \partial_t P_{s:t}(x, y) \Big|_{t=s} \int_{\mathcal{X}} P_{t:\tau}(y, z) \frac{\nu_\tau(z; x_0)}{\mu_\tau(z; x_0)} dz \\
&\quad + \frac{\mu_s(x; x_0)}{\nu_s(x; x_0)} P_{s:t}(x, y) \int_{\mathcal{X}} \partial_t P_{t:\tau}(y, z) \Big|_{t=s} \frac{\nu_\tau(z; x_0)}{\mu_\tau(z; x_0)} dz, \\
&= \frac{\mu_s(x; x_0)}{\nu_s(x; x_0)} A_s(x, y) \int_{\mathcal{X}} P_{s:\tau}(y, z) \frac{\nu_\tau(z; x_0)}{\mu_\tau(z; x_0)} dz \\
&\quad - \delta_{xy} \frac{\mu_s(x; x_0)}{\nu_s(x; x_0)} \int_{\mathcal{X}} \sum_u [A_s(y, u) P_{s:\tau}(u, z)] \frac{\nu_\tau(z; x_0)}{\mu_\tau(z; x_0)} dz, \\
&= \int_{\mathcal{X}} \frac{\mu_s(x; x_0)}{\nu_s(x; x_0)} \left[ A_s(x, y) P_{s:\tau}(y, z) - \delta_{xy} \sum_u A_s(y, u) P_{s:\tau}(u, z) \right] \frac{\nu_\tau(z; x_0)}{\mu_\tau(z; x_0)} dz, \\
&= \int_{\mathcal{X}} \frac{\nu_\tau(z; x_0)}{\mu_\tau(z; x_0)} \frac{\mu_s(x; x_0)}{\nu_s(x; x_0)} P_{s:\tau}(x, z) A_s(x, y; z) dz, \\
&= \int_{\mathcal{X}} \Lambda_{|0}(X_\tau = z | X_s = x) A_s(x, y; z) dz, \\
&= \mathbb{E}_{z \sim \Lambda_{\tau|0,s}}[A_s(x, y; z) | X_0 = x_0, X_s = x].
\end{aligned}
$$

In conclusion, the generator of $\Lambda_{\cdot|0}$ is given as the conditional expectation,

$$
A_s^{\Lambda_{\cdot|0}}(x, y) = \mathbb{E}_{\Lambda_{\tau|0,s}}[A_s(x, y; X_\tau) | X_0, X_s = x].
$$

$\square$

### B.3.4 MARGINAL DISTRIBUTION OF MIXTURE OF MARKOV CHAINS

A reciprocal process $\Lambda \in \mathcal{R}(\mathbb{Q})$ can be represented as a mixture of Markov chains as:

$$
\Lambda(\cdot) = \sum_{x,y} \mathbb{Q}(\cdot | X_0 = x, X_\tau = y) \Lambda_{0,\tau}(x, y).
$$

We here propose the mixture representation of Markov chain similar to theorem 2 of Peluchetti (2023b) which describing mixture of diffusion process over Euclidean space.

**Proposition B.6.** *Consider a family of Markov measures on $\Omega$ indexed by $u \in \mathcal{I}$*

$$\partial_t P^u_{s:t}(x, y) = \sum_z P^u_{s:t}(x, z) A^u_t(z, y),$$

$$A^u_s(x, y) = \partial_t P^u_{s,t}(x, y)\Big|_{t=s},$$

*where $P^u$ and $A^u$ is the transition probability and generator for each measure associated to the process $X^u = (X^u_t)_{t \in [0,\tau]}$. We here assume that $A^u_s$ is finite for every $u$. Let the corresponding Markov measure be $M^u$, and $\mu^u_t$ be the density of $M^u$, the marginal distribution of $X^u_t$. Let a mixture of $M^u$ with the index distribution $\mathcal{U}$ over $\mathcal{I}$ be $\Lambda$,*

$$\Lambda(\cdot) = \int_{\mathcal{I}} M^u(\cdot) \mathcal{U}(du).$$

*We denote mixture marginal density $\mu_t$ and mixture initial distribution $\Lambda_0$ as:*

$$\mu_t(\cdot) = \int_{\mathcal{I}} \mu^u_t(\cdot) \mathcal{U}(du),$$

$$\Lambda_0(\cdot) = \int_{\mathcal{I}} M^u_0(\cdot) \mathcal{U}(du).$$

*Let $X = (X_t)_{t \in [0,\tau]}$ be another Markov chain generated by:*

$$\partial_t P_{s:t}(x, y) = \sum_z P_{s:t}(x, z) A_t(z, y),$$

$$A_t(x, y) = \frac{1}{\mu_t(x)} \int_{\mathcal{I}} A^u_t(x, y) \mu^u_t(x) \mathcal{U}(du),$$

$$X_0 \sim \Lambda_0.$$

*Then, the marginal density of $X_t$ is $\mu_t$. It is assumed that exchange of $\partial_t$ and $\int_{\mathcal{I}} \mathcal{U}(du)$ is justified.*

*Proof.* We start from verifying $A_t(x, y)$ admits conditions of transition rate function. For $x \neq y$ it is trivial that $A_t(x, y)$ is finite and non-negative. Also, $\sum_{y \in \mathcal{X}} A_t(x, y)$ becomes zeros for all $x$ and $t$ as:

$$\sum_{y \in \mathcal{X}} A_t(x, y) = \sum_{y \in \mathcal{X}} \frac{1}{\mu_t(x)} \int_{\mathcal{I}} A^u_t(x, y) \mu^u_t(x) \mathcal{U}(du),$$

$$= \frac{1}{\mu_t(x)} \int_{\mathcal{I}} \sum_{y \in \mathcal{X}} A^u_t(x, y) \mu^u_t(x) \mathcal{U}(du),$$

$$= 0.$$

Thus, mixture of generators $A_t(x, y)$ holds the condition for a generator of Markov measures.

Next, for $t \in (0, \tau)$,

$$\partial_t \mu_t(x) = \partial_t \int_{\mathcal{I}} \mu^u_t(x) \mathcal{U}(du),$$

$$= \int_{\mathcal{I}} \partial_t \mu^u_t(x) \mathcal{U}(du), \qquad \text{(assumption)}$$

$$= \int_{\mathcal{I}} \sum_{y \in \mathcal{X}} A^u_t(y, x) \mu^u_t(y) \mathcal{U}(du), \qquad (\because \text{Kolmogorov equation})$$

$$= \sum_{y \in \mathcal{X}} \left( \int_{\mathcal{I}} A^u_t(y, x) \mu^u_t(y) \mathcal{U}(du) \right),$$

$$= \sum_{y \in \mathcal{X}} A_t(y, x) \mu_t(y).$$

The equality of the left-hand side and the final line corresponds to the Kolmogorov equation for the process $X$ generated by $A_t$. This shows that $\mu_t$ is the marginal distribution of $X_t$. $\qquad \square$

### B.3.5 Minimizer of the KL-divergence

The Markov projection of a reciprocal process $\Lambda$, denoted $\Pi_{\mathcal{M}}(\Lambda)$, is a Markov process $M^*$ which minimizes the reverse KL-divergence $D_{\mathrm{KL}}(\Lambda, M)$. We here characterize $M^*$ by specifying its generator. While the generator $A_t^{\Lambda_{\cdot|0}}$ of $\Lambda_{\cdot|0}$ represented as a conditional expectation given $X_0, X_t$ as noted in Lemma B.5, that of the $M^*$ is supposed to be represented as the conditional expectation without $X_0$ condition. The following lemma says that $\mathbb{E}_{\Lambda_{0|t}} \left[ A_t^{\Lambda_{\cdot|0}}(X_t, y) \right]$ is the generator.

**Lemma B.7.** *Let the Markov projection of a reciprocal process $\Lambda \in \mathcal{R}(\mathbb{Q})$ be $M^* = \Pi_{\mathcal{M}}(\Lambda)$. Then the generator of $M^*$ is $A_t^{M^*}(X_t, y) = \mathbb{E}_{\Lambda_{\tau|t}} \left[ A_t(X_t, y; X_\tau) \middle| X_t \right]$.*

*Proof.* Because $\mathbb{Q}$ is assumed to be able to construct bridge everywhere, $A_t(x, y; z) = 0 \iff A_t(x, y) = 0 \quad \forall x \neq y, z \in \mathcal{X}, t \in [0, \tau)$.

We claim that the Markov measure $M$ generated by $A_t^M(x, y) = \mathbb{E}_{\Lambda_{\tau|t}} \left[ A_t(X_t, y; X_\tau) | X_t = x \right]$ with $M_0 = \Lambda_0$ is a.s. $M^*$. We can re-formulate the generator as:

$$
\begin{aligned}
A_t^M(X_t, y) &= \mathbb{E}_{\Lambda_{\tau|t}} \left[ A_t(X_t, y; X_\tau) \middle| X_t \right], \\
&= \sum_{x_\tau} \Lambda(X_\tau = x_\tau | X_t) A_t(X_t, y; x_\tau), \\
&= \sum_{x_\tau} \sum_{x_0} \Lambda(X_0 = x_0, X_\tau = x_\tau | X_t) A_t(X_t, y; x_\tau), \\
&= \sum_{x_0} \Lambda(X_0 = x_0 | X_t) \sum_{x_\tau} \Lambda(X_\tau = x_\tau | X_0 = x_0, X_t) A_t(X_t, y; x_\tau), \\
&= \sum_{x_0} \Lambda(X_0 = x_0 | X_t) A_t^{\Lambda_{\cdot|0}}(X_t, y), \\
&= \mathbb{E}_{\Lambda_{0|t}} \left[ A_t^{\Lambda_{\cdot|0}}(X_t, y) \middle| X_t \right],
\end{aligned}
\tag{18}
$$

which conclude that it becomes conditional expectation of the generator of $\Lambda_{\cdot|0}$. Also, based on the Proposition B.6, the $M_t = \Lambda_t$ for all $t$.

Note that $A_t^{\Lambda_{\cdot|0}}(x, y) = \mathbb{E}_{\Lambda_{\tau|0,t}} \left[ A_t(x, y; X_\tau) \middle| X_0, X_t = x \right]$, which implies $A_t^M(x, y) = 0 \implies A_t^{\Lambda_{\cdot|0}}(x, y) = 0 \quad \forall x \neq y, z \in \mathcal{X}, t \in [0, \tau)$. Thus, $\Lambda \ll M$. Because $M^*$ is the minimizer of KL-divergence $D_{\mathrm{KL}}(\Lambda \| \cdot)$, we can assume $M_0^* = \Lambda_0$. Based on this, we want to show that $D_{\mathrm{KL}}(\Lambda \| M^*) - D_{\mathrm{KL}}(\Lambda \| M) = 0$. Recall that Equation (17), where the KL-divergence is formulated as

$$
\begin{aligned}
D_{\mathrm{KL}}(\Lambda \| M) &= \int_0^\tau \mathbb{E}_{\Lambda_{0,t}} \left[ \left( A_t^{\Lambda_{\cdot|0}} - A_t^M \right) (X_t, X_t) + \sum_{y \neq X_t} A_t^{\Lambda_{\cdot|0}} \log \frac{A_t^{\Lambda_{\cdot|0}}}{A_t^M} (X_t, y) \right] dt, \\
&= \int_0^\tau \mathbb{E}_{\Lambda_{0,t}} \left[ \underbrace{\sum_{y \neq X_t} \left( A_t^{\Lambda_{\cdot|0}} \log \frac{A_t^{\Lambda_{\cdot|0}}}{A_t^M} - A_t^{\Lambda_{\cdot|0}} + A_t^M \right) (X_t, y)}_{f(t, \Lambda, M)} \right] dt.
\end{aligned}
$$

Then, $D_{\mathrm{KL}}(\Lambda \| M^*) - D_{\mathrm{KL}}(\Lambda \| M) = \int_0^\tau \Delta dt \leq 0$ by definition, where $\Delta := f(t, \Lambda, M^*) - f(t, \Lambda, M)$.

$$\Delta = \mathbb{E}_{\Lambda_{0,t}} \left[ \sum_{y \neq X_t} \left( A_t^{M^*} - A_t^M + A_t^{\Lambda_{\cdot|0}} \log \frac{A_t^M}{A_t^{M^*}} \right) (X_t, y) \right],$$

$$= \mathbb{E}_{\Lambda_t} \mathbb{E}_{\Lambda_{0|t}} \left[ \sum_{y \neq X_t} \left( A_t^{M^*} - A_t^M + A_t^{\Lambda_{\cdot|0}} \log \frac{A_t^M}{A_t^{M^*}} \right) (X_t, y) \right],$$

$$= \mathbb{E}_{\Lambda_t} \sum_{y \neq X_t} \mathbb{E}_{\Lambda_{0|t}} \left[ \left( A_t^{M^*} - A_t^M + A_t^{\Lambda_{\cdot|0}} \log \frac{A_t^M}{A_t^{M^*}} \right) (X_t, y) \right],$$

$$= \mathbb{E}_{\Lambda_t} \sum_{y \neq X_t} \left( A_t^{M^*} - A_t^M + \mathbb{E}_{\Lambda_{0|t}} \left[ A_t^{\Lambda_{\cdot|0}} | X_t \right] \log \frac{A_t^M}{A_t^{M^*}} \right) (X_t, y), \qquad (\because M, M^* \in \mathcal{M})$$

$$= \mathbb{E}_{\Lambda_t} \sum_{y \neq X_t} \left( A_t^{M^*} - A_t^M + A_t^M \log \frac{A_t^M}{A_t^{M^*}} \right) (X_t, y), \qquad (\because Equation\ (18))$$

$$= \mathbb{E}_{M_t} \sum_{y \neq X_t} \left( A_t^{M^*} - A_t^M + A_t^M \log \frac{A_t^M}{A_t^{M^*}} \right) (X_t, y). \qquad (\because M_t = \Lambda_t)$$

We can deduce that $\int_0^\tau \Delta dt = D_{\mathrm{KL}}(M \| M^*) \geq 0$, which conclude that $M = M^*$.

$\square$

### B.3.6    PROOF OF PROPOSITION B.2

Now we prove the proposition Proposition B.2 using above lemmas.

*Proof.* Proposition B.2

By the Lemma B.3 and Equation (17), the KL-divergence of $\Lambda$ to any Markov measure $M \in \mathcal{M}$ disintegrates as:

$$D_{\mathrm{KL}}(\Lambda \| M) = \int_0^\tau \mathbb{E}_{\Lambda_{0,t}} \left[ \left( A_t^{\Lambda_{\cdot|0}} - A_t^M \right) (X_t, X_t) + \sum_{y \neq X_t} A_t^{\Lambda_{\cdot|0}} \log \frac{A_t^{\Lambda_{\cdot|0}}}{A_t^M} (X_t, y) \right] dt,$$

where the $A_t^{\Lambda_{\cdot|0}}$ is the generator of pinned process of $\Lambda_{\cdot|0}$ stated by Lemma B.5. According to Lemma B.7, the Markov measure $M^*$ generated by $A_t(x, y) = \mathbb{E}_{\Lambda_{\tau|t}} [A_t(x, y; X_\tau) | X_t = x]$ is the minimizer of $D_{\mathrm{KL}}(\Lambda \| M)$ for $M \in \mathcal{M}$. In last, Proposition B.6 ensure the time marginals of $\Lambda, M^*$ for all $t \in [0, \tau]$ are equivalent. $\square$

### B.4    TIME-REVERSAL MARKOV PROJECTION

**Proposition B.8.** *Let $M^* = \Pi_{\mathcal{M}}(\Lambda), \Lambda \in \mathcal{R}(\mathbb{Q})$ and the forward generator of $\mathbb{Q}$ be $A_t(x, y)$ with the transition probability $P_{s:t}(x, y)$. Let $X = (X_t)_{t \in [0, \tau]}$ be the canonical process and $Y = (Y_t)_{t \in [0, \tau]}$ be the time reversal of $X$, where $Y_t = X_{\tau-t}$. Under mild assumptions, the time-reversal generator of $M^*$ becomes*

$$\tilde{A}_t^{M^*}(Y_t, x) = \mathbb{E}_{0|t} \left[ \tilde{A}_t(Y_t, x; X_0) \Big| Y_t \right]$$

$$\tilde{A}_s(y, x; z) = \partial_t \tilde{P}_{s:t}(y, x; z) \Big|_{t=s}$$

$$= A_{\tau-s}(x, y) \frac{P_{0:\tau-s}(z, x)}{P_{0:\tau-s}(z, y)} - \delta_{xy} \sum_u A_{\tau-s}(u, x) \frac{P_{0:\tau-s}(z, u)}{P_{0:\tau-s}(z, x)},$$

where $z \in \mathcal{X}$, and $\tilde{A}_s(\cdot, \cdot; z)$ is the generator for the conditioned process of $Y$ with $Y_\tau = z$. The reverse KL-divergence is

$$D_{KL}(\Lambda \| M^*) = \int_0^\tau \mathbb{E}_{\Lambda_{\tau,t}} \left[ (\tilde{A}^{\Lambda \cdot|\tau} - \tilde{A}^{M^*})(Y_t, Y_t) + \sum_{x \neq Y_t} \tilde{A}^{\Lambda \cdot|\tau} \log \frac{\tilde{A}^{\Lambda \cdot|\tau}}{\tilde{A}^{M^*}}(Y_t, x) \right] dt,$$

where the $\tilde{A}^{\Lambda \cdot|\tau}$ is the time-reversal generator for the conditioned measure $\Lambda_{\cdot|\tau}$ which is defined as

$$\tilde{A}_t^{\Lambda \cdot|\tau}(Y_t, x) = \mathbb{E}_{\Lambda_{0|t,\tau}} \left[ \tilde{A}(Y_t, x; Y_\tau) \middle| Y_t, Y_0 \right].$$

*Proof.* Follow the proof of Proposition B.2. Note that the Markov measure is time-symmetric.  □

### B.5   PROOF OF THEOREM 3.3

*Proof.* With iterative projection, we have a sequence of measure $\Lambda^{(n)}$ such that:

$$\Lambda^{(2n+1)} = \Pi_{\mathcal{M}}(\Lambda^{(2n)}),$$
$$\Lambda^{(2n+2)} = \Pi_{\mathcal{R}(\mathbb{Q})}(\Lambda^{(2n+1)}),$$

where $\Lambda^{(0)} \in \mathcal{R}(\mathbb{Q})$. Let $\Pi^{(n)} = \Lambda^{(2n)}$ and $M^{(n)} = \Lambda^{(2n+1)}$. We will omit the superscription $\cdot^{(n)}$ if there is no confusion.

Let $\mathbb{P} \in \mathcal{M}$ be a Markov process generated by a generator $A_t^{\mathbb{P}}(x, y) < \infty$ with initial distribution $M_0$ with the Kolmogorov forward equation. Assuming that $D_{\text{KL}}(\Pi \| \mathbb{P}) < \infty$, then

$$D_{\text{KL}}(\Pi \| \mathbb{P}) - D_{\text{KL}}(\Pi \| M) = \mathbb{E}_\Pi \left[ \log \frac{dM}{d\mathbb{P}} \right] < \infty,$$

which implying the Radon-Nikodym derivative $\frac{dM}{d\mathbb{P}} < \infty$ over the support of $\Pi$.

Then we will first show the equality:

$$\mathbb{E}_\Pi \left[ \log \frac{dM}{d\mathbb{P}} \right] = \mathbb{E}_M \left[ \log \frac{dM}{d\mathbb{P}} \right].$$

As $\Pi_0 = M_0$, it is sufficient to show

$$\mathbb{E}_\Pi \left[ \log \frac{dM_{\cdot|0}}{d\mathbb{P}_{\cdot|0}} \right] = \mathbb{E}_M \left[ \log \frac{dM_{\cdot|0}}{d\mathbb{P}_{\cdot|0}} \right],$$

where

$$\log \frac{dM_{\cdot|0}}{d\mathbb{P}_{\cdot|0}}(\omega) = \int_0^\tau \log \frac{A_t^M}{A_t^{\mathbb{P}}}(\omega_{t-}, \omega_t) dN_t(\omega) + \int_0^\tau (A_t^M - A_t^{\mathbb{P}})(\omega_t, \omega_t) dt$$

is given from Appendix B.3.1. Since $\Pi_{|0}$ is Markov from Lemma B.3 with generator

$$A_t^{\Pi_{|0}}(x, y) = \mathbb{E}_{\Pi_{\tau|0,t}} \left[ A_t(x, y; X_\tau) | X_0, X_t = x \right],$$

which is derived at the Lemma B.5. Then,

$$\mathbb{E}_\Pi \left[ \int_0^\tau \log \frac{A_t^M}{A_t^{\mathbb{P}}}(X_{t-}, X_t) dN_t \right]$$

$$= \mathbb{E}_\Pi \left[ \int_0^\tau \sum_{y \neq X_t} A_t^{\Pi \cdot|0} \log \frac{A_t^M}{A_t^{\mathbb{P}}}(X_t, y) dt \right],$$

$$= \mathbb{E}_\Pi \left[ \int_0^\tau \sum_{y \neq X_t} \mathbb{E}_\Pi [A_t(X_t, y; X_\tau) | X_0, X_t] \log \frac{A_t^M}{A_t^{\mathbb{P}}}(X_t, y) dt \right],$$

$$= \mathbb{E}_\Pi \left[ \int_0^\tau \sum_{y \neq X_t} \mathbb{E}_\Pi [A_t(X_t, y; X_\tau) | X_t] \log \frac{A_t^M}{A_t^{\mathbb{P}}}(X_t, y) dt \right],$$

by the tower property of conditional expectation. From the Lemma B.7 and Proposition B.6, we can replace the $\mathbb{E}_\Pi[A_t(X_t, y; X_\tau)|X_t]$ as $A_t^M(X_t, y)$,

$$\mathbb{E}_\Pi\left[\int_0^\tau \sum_{y \neq X_t} \mathbb{E}_\Pi[A_t(X_t, y; X_\tau)|X_t] \log \frac{A_t^M}{A_t^\mathbb{P}}(X_t, y)dt\right],$$

$$= \mathbb{E}_\Pi\left[\int_0^\tau \sum_{y \neq X_t} A_t^M \log \frac{A_t^M}{A_t^\mathbb{P}}(X_t, y)dt\right],$$

$$= \mathbb{E}_M\left[\int_0^\tau \sum_{y \neq X_t} A_t^M \log \frac{A_t^M}{A_t^\mathbb{P}}(X_t, y)dt\right],$$

where the last equation justified since $\Pi_t = M_t$ for all $t \in [0, \tau]$. The other term, $\int_0^\tau (A_t^M - A_t^\mathbb{P})(X_t, X_t)dt$, can be treated as the same way, which establishes the equality. From the result, we get $D_{\text{KL}}(\Pi\|\mathbb{P}) = D_{\text{KL}}(\Pi\|M) + D_{\text{KL}}(M\|\mathbb{P})$. The equality is derived for other diffusion processes (Peluchetti, 2023a; Shi et al., 2024; Liu & Wu, 2023), but in this paper we extended it to continuous Markov chain with discrete state space case.

By letting $\mathbb{P} = \mathbb{P}^{\text{SB}}$, we get

$$D_{\text{KL}}(\Pi\|\mathbb{P}^{\text{SB}}) \geq D_{\text{KL}}(M\|\mathbb{P}^{\text{SB}}),$$

the equality holds if and only if $\Pi = M$. By the assumption $D_{\text{KL}}(\Lambda^{(0)}\|\mathbb{P}^{\text{SB}}) < \infty$, the sequence of KL-divergence $\{D_{\text{KL}}(\Lambda^{(n)}\|\mathbb{P}^{\text{SB}})\}_{n \in \mathbb{N}}$ is non-increasing and bounded, which implies the KL-divergence converges. Thus,

$$\lim_{n \to \infty} D_{\text{KL}}(\Pi^{(n)}\|\mathbb{P}^{\text{SB}}) - D_{\text{KL}}(M^{(n)}\|\mathbb{P}^{\text{SB}}) = \lim_{n \to \infty} D_{\text{KL}}(\Pi^{(n)}\|M^{(n)}) = 0.$$

We here utilize Aldous' tightness criteria for showing the tightness of $\mathbb{P}^{\text{SB}}$. Since the state space is finite it is sufficient to show the following: For any $\varepsilon > 0$, there exists $\delta > 0$ such that for all stopping time $t$,

$$\mathbb{P}^{\text{SB}}(d_\mathcal{X}(X_{t+\theta}, X_t) \geq \eta) \leq \varepsilon,$$

for arbitrary small $\eta > 0$ and $0 < \theta < \delta^4$. To check the criteria, we will show that the leaving rate of $\mathbb{P}^{\text{SB}}$ is bounded. From the Lemma B.5 and Lemma B.7, we know the analytic form of the transition rate $A_t$. Since Theorem B.1 ensure that the $\mathbb{P}^{\text{SB}}$ is Markov as well as is in reciprocal class,

$$A_t^{\mathbb{P}^{\text{SB}}}(x, x) = \mathbb{E}_{\mathbb{P}^{\text{SB}}}[A_t(x, x; X_\tau)|X_0 = x_0, X_t = x],$$

$$= \sum_{X_\tau \in \mathcal{X}}\left[\mathbb{P}^{\text{SB}}(X_\tau|X_t = x, X_0 = x_0)A_t(x, x)\right.$$

$$\left. - \frac{\mathbb{P}^{\text{SB}}(X_\tau|X_t = x, X_0)}{P_{t:\tau}(x, X_\tau)} \sum_{u \in \mathcal{X}} A(x, u)P_{t:\tau}(u, X_\tau)\right],$$

$$= \sum_{X_\tau \in \mathcal{X}}\left[\mathbb{P}^{\text{SB}}(X_\tau|X_t = x)A_t(x, x)\right.$$

$$\left. - \frac{\mathbb{P}^{\text{SB}}(X_\tau|X_t = x, X_0)}{P_{t:\tau}(x, X_\tau)} \sum_{u \in \mathcal{X}} A(x, u)P_{t:\tau}(u, X_\tau)\right],$$

$$= A_t(x, x) - \sum_{X_\tau \in \mathcal{X}}\left[\frac{\mathbb{P}^{\text{SB}}(X_\tau|X_t = x, X_0)}{P_{t:\tau}(x, X_\tau)} \sum_{u \in \mathcal{X}} A(x, u)P_{t:\tau}(u, X_\tau)\right],$$

$$= A_t(x, x) - \sum_{X_\tau \in \mathcal{X}}\left[\frac{d\mathbb{P}_{t,\tau}^{\text{SB}}}{d\mathbb{Q}_{t,\tau}}(x, X_\tau) \sum_{u \in \mathcal{X}} A(x, u)P_{t:\tau}(u, X_\tau)\right].$$

---

[4] See section 16 of Billingsley (2013) and section 3 of Ethier & Kurtz (2009)

The Radon-Nikodym derivative is finite and positive for $x$, such that $\mathbb{P}^{\text{SB}}(X_t = x) > 0$. Because the state space $\mathcal{X}$ is finite, there is finite upper bound $u_t$ for every possible pair $(x, x_\tau)$. Thus,

$$
A_t^{\mathbb{P}^{\text{SB}}}(x, x) \geq A_t(x, x) - \sum_{X_\tau \in \mathcal{X}} \left[ u_t \sum_{u \in \mathcal{X}} A(x, u) P_{t:\tau}(u, X_\tau) \right],
$$
$$
= A_t(x, x) - u_t \sum_{u \in \mathcal{X}} A(x, u) \sum_{X_\tau \in \mathcal{X}} P_{t:\tau}(u, X_\tau),
$$
$$
= A_t(x, x) - u_t \sum_{u \in \mathcal{X}} A(x, u),
$$
$$
= A_t(x, x).
$$

The leaving rate $c_t^{\text{SB}}(x)$ of $\mathbb{P}^{\text{SB}}$ at $(t, x)$ is bounded by that of $Q$, $c_t(x)$. Assuming the leaving rate of $Q$ is uniformly bounded by $c$, $c_t^{\text{SB}}$ is uniformly bounded by $c$. Back to the Aldous' tightness criteria, by choosing $\delta = \varepsilon/c$ we can see that $\mathbb{P}^{\text{SB}}$ is tight.

The sequence $\{\Pi^{(n)}\}_{n \in \mathbb{N}}$ is tight. Since $\mathbb{P}^{\text{SB}}$ is tight, for any $\varepsilon > 0$ we can choose a compact and measurable $K$ (under Skorokhod topology and associated Borel $\sigma$ algebra). For any measurable $K$,

$$
D_{\text{KL}}(\Pi^{(n)} \| \mathbb{P}^{\text{SB}}) = \mathbb{E}_{\Pi^{(n)}} \left[ -\log \frac{d\mathbb{P}^{\text{SB}}}{d\Pi^{(n)}} | K^c \right] \Pi^{(n)}(K^c) + \mathbb{E}_{\Pi^{(n)}} \left[ -\log \frac{d\mathbb{P}^{\text{SB}}}{d\Pi^{(n)}} | K \right] \Pi^{(n)}(K),
$$
$$
\geq -\log \frac{\mathbb{P}^{\text{SB}}(K^c)}{\Pi^{(n)}(K^c)} \Pi^{(n)}(K^c) - \log \frac{\mathbb{P}^{\text{SB}}(K)}{\Pi^{(n)}(K)} \Pi^{(n)}(K),
$$

by the Jensen inequality. If $\{\Pi^n\}_{n \in \mathbb{N}}$ is not tight, for each compact $K$ and $\lambda > 0$, there is at least one $n \in \mathbb{N}$ where $\Pi^{(n)}(K^c) \geq \lambda, \Pi^{(n)}(K) < 1 - \lambda$. Thus, for $\varepsilon > 0$ there exists $n \in \mathbb{N}$, such that

$$
-\log \frac{\mathbb{P}^{\text{SB}}(K^c)}{\Pi^{(n)}(K^c)} \Pi^{(n)}(K^c) \geq -\log(\varepsilon/\lambda)\lambda,
$$

which implies that the lower bound of $D_{\text{KL}}(\Pi^{(n)} \| \mathbb{P}^{\text{SB}})$ can be arbitrary large. However, the KL-divergence is non-increasing and upper bounded by $D_{\text{KL}}(\Pi^{(0)} \| \mathbb{P}^{\text{SB}}) < \infty$, contradiction occurs. Thus, the sequence of measure should be tight. Similarly, we can show the $\{M^{(n)}\}_{n \in \mathbb{N}}$ is tight.

The space $(\Omega, d_\Omega)$ is a Polish space[5], and therefore, by Prokhorov's theorem, the collections of measures $\{\Pi^{(n)}\}_{n \in \mathbb{N}}$ and $\{M^{(n)}\}_{n \in \mathbb{N}}$ are relatively compact. Each subsequence of $\{M^{(n)}\}_{n \in \mathbb{N}}$ has a sub-subsequence $\{\Pi^{(i)}\}_{i \geq 0}$ weakly converges to $\Pi^{(\infty)}$ as $i \to \infty$. Similarly, there is a sub-subsequence $\{M^{(i)}\}_{i \geq 0}$ that converges in law to $M^{(\infty)}$. By the lower semi-continuity of the KL-divergence, $D_{\text{KL}}(\Pi^{(\infty)} \| M^{(\infty)}) \leq \liminf_{i \to \infty} (\Pi^{(i)} \| M^{(i)}) = 0$, which implies the two convergence point is equal to $\mathbb{P}^{(\infty)}$. The resulting measure is Markov and is in $\mathcal{R}(\mathbb{Q})$, because the state space is finite, which deduce $\mathbb{P}^{\text{SB}} = \mathbb{P}^{(\infty)}$. We choose an arbitrary convergent point of sub-subsequence resulting in $\mathbb{P}^{\text{SB}}$, the convergence is ensured[6]. $\qquad\square$

## C  Algorithm

---

**Algorithm 1** Iterative Discrete Markovian Fitting

---

**Require:** Joint distribution $\pi$, tractable bridge $\mathbb{Q}_{\cdot|0,\tau}$, number of outer iterations $N \in \mathbb{N}$

1: Let $\Lambda^{(0)} = \pi \mathbb{Q}_{\cdot|0,\tau}$
2: **for** $n = 0, \ldots, N$ **do**
3:     Learn $\phi$ using Equation (6) with $\Lambda^{(2n)}$
4:     Let $\Lambda^{(2n+1)} = M_b^{(2n+1)} \mathbb{Q}_{\cdot|0,\tau}$
5:     Learn $\theta$ using Equation (5) with $\Lambda^{(2n+1)}$
6:     Let $\Lambda^{(2n+2)} = M_f^{(2n+2)} \mathbb{Q}_{\cdot|0,\tau}$
7: **end for**
8: **return** $\theta^*, \phi^*$

---

[5]See section 12 of Billingsley (2013) or section 3.5 of Ethier & Kurtz (2009)
[6]See Theorem 2.6 of Billingsley (2013)

# D  GRAPH PERMUTATION MATCHING

## D.1  INTRODUCTION

A graph $\mathcal{G} = (\mathcal{V}, \mathcal{E})$ is defined as the set of nodes $\mathcal{V} = \{v_i\}$ and the set of edges $\{e_{ij}\}$, where $i$ and $j$ denote the node indices. Under a permutation $\sigma \in S_n$, the structure of the graph $\mathcal{G}$ remains unchanged, as the node and edge sets are invariant to indexing:

$$\sigma(\mathcal{V}) = \{v_{\sigma(i)}\} = \mathcal{V},$$
$$\sigma(\mathcal{E}) = \{e_{\sigma(i)\sigma(j)}\} = \mathcal{E},$$

However, the vectorized representation of a graph, $\mathbf{G} = (\mathbf{V}, \mathbf{E})$ is affected by the permutation because $\mathbf{V}$ and $\mathbf{E}$ are treated as ordered sets. Thus, a graph $\mathcal{G}$ can be considered as a set of all permuted version of its vectorized representation:

$$\mathcal{G} = \{\sigma(\mathbf{G}) : \sigma \in S_n\},$$

where the $\mathbf{G}$ is an arbitrarily indexed vectorization of $\mathcal{G}$.

The likelihood of transitioning from an initial graph $\mathbf{G} = (\mathbf{V}, \mathbf{E})$ to a terminal graph $\mathbf{G}' = (\mathbf{V}', \mathbf{E}')$, denoted $p(\mathbf{G}'|\mathbf{G}) = \mathbb{Q}(X_\tau = \mathbf{G}'|X_0 = \mathbf{G})$, depends on both node and edge correspondences, whereas the likelihood $p(\mathcal{G}'|\mathbf{G})$, which is permutation-invariant, does not. More specifically, for any permutation $\sigma$, the likelihood $p(\sigma(\mathbf{G}')|\mathbf{G})$ changes, even though the underlying graph $\mathcal{G}'$ remains unchanged. Calculating the likelihood of the graph itself (as opposed to the graph vector) would require summing the likelihoods of all possible permutations of $\mathbf{G}'$, but this approach is computationally prohibitive due to the factorial number of permutations.

This dependency on graph permutation introduces challenges not only in computing likelihoods but also in constructing reciprocal measures. Ideally, a reciprocal measure over graph domain would account for all permutations of graph vectors. However, as with likelihood computations, the construction of reciprocal measures is highly sensitive on the alignment of graph vectors, making proper handling of these permutations essential for the iterative Markovian fitting (IMF) algorithm.

## D.2  SUB-OPTIMAL VS. OPTIMAL PERMUTATION

In practice, the key insight is that the likelihood difference of graph vectors between the optimal permutation $\sigma^*$ and sub-optimal permutations $\sigma$ is substantial, allowing us to neglect sub-optimal permutations. The design choice of the reference process $\mathbb{Q}$, particularly its signal to noise ratio $\frac{\bar{\alpha}(\tau)}{\bar{\alpha}(0)} \gg 0$ (see Equation (9)), ensures the transitions preserving the initial states are far more probable. As a result, the likelihood for sub-optimal permutations, where node and edge correspondences are mismatched, is expected to be significantly lower than for the optimal permutation. The reason is that even a small mismatch in node alignment between $\sigma^*$ and $\sigma$ can cause a large number of edge state mismatches, leading to a dramatic decrease in the total likelihood.

Thus, the likelihood $p(\sigma^*\mathbf{G}'|\mathbf{G}) \gg p(\sigma\mathbf{G}'|\mathbf{G})$ and the contribution of sub-optimal permutation in calculating $p(\mathcal{G}'|\mathcal{G})$ is negligible. It is sufficient that focus solely on the optimal permutation $\sigma^*$ for the likelihood computation, as the probability of sub-optimal arrangements is effectively zero in practical terms.

The prioritization of optimal permutation could be also utilized in the construction of reciprocal measures. Suppose we have only one graph for initial and terminal, where the one is $\mathcal{G}$ and $\mathcal{G}'$, respectively. Without loss of generality, we assume an even distribution over graph vectors $\mathbf{G} \in \mathcal{G}$, and then consider permutations over $\sigma\mathbf{G}' \in \mathcal{G}'$. Note that $p(\sigma\mathbf{G}'|\mathbf{G}) = p(\sigma'\sigma\mathbf{G}'|\sigma'\mathbf{G})$. According to the static SB solution, the graph vectors $\sigma\mathbf{G}' \in \mathcal{G}'$ are distributed according to the likelihood $p(\sigma\mathbf{G}'|\mathbf{G})$ for each graph vector $\mathbf{G} \in \mathcal{G}$, where the optimal permutation is selected most frequently. In this context, neglecting sub-optimal permutation can be viewed as the static OT solution.

As the original SB problem is formulated with the dynamics over graph *vector* domain, arranging the graph vectors is a part of the SB problem. However, by relying on the optimal permutation, we can reduce complexity of the SB problem aroused by the graph vectors alignment. This strategy ensures the transition from a graph vector to the optimally permuted graph vector, resulting in more reliable convergence of the iterative algorithm.

### D.3 STANDARD FORMULATION OF GRAPH MATCHING PROBLEM

In this section we will discuss about the standard formulation of the graph matching problem and how the task of finding the optimal permutation $\sigma^*$, which minimizes the negative log-likelihood (NLL) $-\log p(\sigma \mathbf{G}'|\mathbf{G})$, can be framed as a graph matching problem.

Consider two graph vectors $\mathbf{G} = (\mathbf{V}, \mathbf{E})$ and $\mathbf{G}' = (\mathbf{V}', \mathbf{E}')$ along with the cost function $\mathbf{c}^V$ and $\mathbf{c}^E$, which define the cost of mapping nodes and edges, respectively. The binary assignment matrix $\mathbf{X} \in \{0, 1\}^{n \times n'}$ represents the matching between nodes, where $n$ and $n'$ denote the number of nodes of $\mathbf{G}$ and $\mathbf{G}'$, respectively. If $v_i \in \mathbf{V}$ matches $v'_a \in \mathbf{V}'$, then $\mathbf{X}_{i,a} = 1$, while all other entries for node $v_i$ are zero.

The total node matching cost is given by $\sum_{\mathbf{X}_{i,a}=1} \mathbf{c}^V(v_i, v'_a)$, where $\mathbf{c}^V(v_i, v'_a)$ denotes the cost of matching from $v_i \in \mathbf{V}$ to $v'_a \in \mathbf{V}'$. Similarly, the total edge matching cost is $\sum_{\mathbf{X}_{i,a}=1, \mathbf{X}_{j,b}=1} \mathbf{c}^E(e_{ij}, e'_{ab})$, where $\mathbf{c}^E(e_{ij}, e'_{ab})$ is the cost of matching from $e_{ij} \in \mathbf{E}$ to $e'_{ab} \in \mathbf{E}'$.

We define the cost matrix $\mathbf{A} \in \mathbb{R}^{nn' \times nn'}$, where the diagonal components $\mathbf{A}_{ia,ia} = \mathbf{c}^V(v_i, v'_a)$ represent node costs and the off-diagonal components $\mathbf{A}_{ia,jb} = \mathbf{c}^E(e_{ij}, e'_{ab})$ represent edge costs. By flattening the assignment matrix to be $\mathbf{x} \in \{0, 1\}^{nn'}$, we can write the total cost function $f$ as

$$f(\mathbf{x}) = \sum_{\mathbf{x}_{ia}=1, \mathbf{x}_{jb}=1} \mathbf{c}^E(e_{ij}, e'_{ab}) + \sum_{\mathbf{x}_{ia}=1} \mathbf{c}^V(v_i, v'_a),$$
$$= \mathbf{x}^\top \mathbf{A} \mathbf{x}.$$

Thus, the graph matching problem is find the optimal assignment $\mathbf{x}$ that minimizing cost $f(\mathbf{x})$, which is written in formal:

$$\mathbf{x}^* = \arg\min_{\mathbf{x}} \mathbf{x}^\top \mathbf{A} \mathbf{x}, \tag{19}$$
$$s.t. \sum_{i=1}^{n} \mathbf{x}_{ia} \leq 1, \sum_{a=1}^{n'} x_{ia} \leq 1,$$

where the inequalities become equalities if $n = n'$. This formulation corresponds to the well-known quadratic assignment problem (QAP), which is NP-hard.

The NLL $-\log p(\mathbf{G}'|\mathbf{G})$ can be decomposed by the sum of the NLLs of nodes and edges:

$$-\log p(\mathbf{G}'|\mathbf{G}) = \sum_{\delta_{ia}=1, \delta_{jb}=1} -\log P_{0:\tau}^E(e_{ij}, e'_{ab}) + \sum_{\delta_{ia}=1} -\log P_{0:\tau}^V(v_i, v'_a).$$

By interpreting the NLLs of nodes and edges as cost function $\mathbf{c}^V$ and $\mathbf{c}^E$, respectively, the problem of finding the optimal permutation $\sigma^*$ can be formulated as a QAP, as in Equation (19).

### D.4 SOLUTION METHOD

Exact solution methods for the QAP, such as mixed integer programming (MIP), requires combinatorial optimization, which incurs prohibitive computational costs (Sahni & Gonzalez, 1976). Many accelerated algorithms adopt branch-and-bound strategies that utilize bounds of objective functions (Anstreicher, 2003; Gilmore, 1962; Loiola et al., 2007; Xia, 2008), reducing the exploration space. However, the combinatorial optimization cannot be avoidable in these strategy. Alternatively, continuous relaxation methods (Cho et al., 2014; Leordeanu & Hebert, 2005) solve the QAP in Equation (19) using a continuous vector $\mathbf{x}$, bypassing combinatorial explorations. However, the continuous relaxation yields non-binary vectors, which require a discretization step, potentially introducing errors.

In our work, we compared two continuous relaxation methods: spectral method (SM) and maximum polling method (MPM), followed by the Hungarian algorithm for post-discretization. Both methods iteratively update the continuous version of assignment vector $\mathbf{x}$ as:

$$\mathbf{x}_{k+1} = \frac{\mathbf{x}_k - \mathbf{A} \odot \mathbf{x}_k \varepsilon}{v}, \tag{20}$$

where $\odot$ denotes matrix multiplication for SM and max-pooled matrix multiplication for MPM, and $\varepsilon$ denotes step size, and $v$ denotes the normalizer. Specifically, the $\mathbf{A} \odot \mathbf{x}$ for SM is described as

$$(\mathbf{A} \odot \mathbf{x})_{ia} = \mathbf{x}_{ia}\mathbf{A}_{ia:ia} + \sum_{j \in \mathcal{N}_i} \sum_{b \in \mathcal{N}_a} \mathbf{x}_{jb}\mathbf{A}_{ia;jb}.$$

In the MPM case, the operation is defined as

$$(\mathbf{A} \odot \mathbf{x})_{ia} = \mathbf{x}_{ia}\mathbf{A}_{ia:ia} + \sum_{j \in \mathcal{N}_i} \max_{b \in \mathcal{N}_a} \mathbf{x}_{jb}\mathbf{A}_{ia;jb}.$$

In practice, we modified the cost matrix $\mathbf{A}$ slightly for efficiency. To accelerate the process, we neglect all edges corresponding to dummies in $\mathbf{G}'$, which significantly reduces computational cost, making $|\mathcal{N}_a|$ scale linearly with $n'$. Additionally, a small Gaussian perturbation was applied to $\mathbf{A}$, slightly altering the minima in the continuous vector space. We solved each QAP ten times to compensate the effects of randomness, improving performance with reasonable computational costs (see Appendix D.5).

## D.5 PERFORMANCE OF GRAPH MATCHING ALGORITHM

In this section, we compare the performance of the following algorithms under different conditions: (1) SM algorithm, (2) MPM algorithm, and (3) MPM algorithm with randomness. The hyperparameters $\bar{\alpha}(\tau)/\bar{\alpha}(0) = 0.3$ for the reference process $\mathbb{Q}$.

To evaluate the effectiveness of QAP solvers on molecular graphs, we selected 100 molecules from the ZINC test set. In all experiments, the source molecule was treated as fully connected (with dummy types), while the target molecule retained only its original edges. We performed up to 1,000 iterations of updates according to Equation (20) with a specified tolerance.

To assess the performance of the algorithms, we permuted the molecular graphs and then tested whether the original indices could be recovered, where optimality is achieved with the inverse permutation. For this task, we evaluated the exact matching ratio, which indicates the proportion of cases where the original indices were successfully recovered. Due to the inherent symmetry in molecular graphs, multiple optimal permutations may exist. Therefore, instead of relying solely on exact index recovery, we base the success criterion on the objective function value. If the Negative Log-Likelihood (NLL) error falls below 1e-2, the solution is considered successful.

For different molecule pairs where no optimal permutation is available, which is common in practical scenarios, exact solvers become computationally prohibitive even with relatively small molecular graphs. As such, we evaluated the performance of the QAP solvers by measuring the reduction in NLL between the initial and optimized permutations.

We conducted experiments with various hyperparameters, and the results are illustrated in Table 3. All algorithms successfully found exact matches for the same molecules, but the MPM algorithm performed better than the SM algorithm when applied to different molecule pairs. For all subsequent experiments, we adopted the MPM algorithm with randomness in the cost matrix as the QAP solver.

Table 3: **Comparison of different algorithms based on configuration parameters.** NLL drop represents the improvement in likelihood between randomly paired molecules, while exact match reflects the percentage of permutations successfully recovered after applying a random permutation to the molecules. $\uparrow$ and $\downarrow$ denote higher and lower values are better, respectively.

| Algorithm | Configuration | | | | | | NLL drop($\uparrow$) | Exact matching($\uparrow$) |
| | Pooling | Tolerance | Precision | Max # iterations | Noise coefficient | # trials | | |
|---|---|---|---|---|---|---|---|---|
| (1) SM | Sum | 1e-9 | FP64 | 1000 | 0 | 1 | 2.251 | 100 % |
| (2) MPM | Max | 1e-5 | FP32 | 1000 | 0 | 1 | 13.256 | 100 % |
| (3) MPM + Randomness | Max | 1e-5 | FP32 | 1000 | 1e-6 | 10 | 13.256 | 100 % |
| (4) Our setting | Max | 1e-4 | FP32 | 2500 | 1e-6 | 10 | 14.324 | 100 % |

## D.6 RELATION TO GRAPH EDIT DISTANCE

The graph edit distance (GED) is a widely used and flexible metric for measuring the dissimilarity between two graphs. It is defined as the minimum cost required to transform one graph into another

through a sequence of unit operations. Each unit operation can be a removal, substitution, or insertion, and can be applied to either node or edges. The total cost of the transformation is the sum of the costs assigned to these components.

According to Bougleux et al. (2015), finding the minimal-cost transformation between two graphs is equivalent to the QAP problem with an associated cost function. Though both problems are NP-hard, the equivalence is meaningful in that there are numerous approximation algorithms for the QAP problem that operate within polynomial time.

The basic idea for re-formulation into QAP problem involves the introduction of dummy nodes and dummy edges, where removal (insertion) operations could be replaced by substitution into (from) dummies. The cost function of unit operation is defined as $\mathbf{c}^V(v_i, v'_a)$ and $\mathbf{e}^E(e_{ij}, e'_{ab})$ for node replacement and edge replacement, respectively. Let $\alpha = \bar{\alpha}(\tau)/\bar{\alpha}(0)$, the cost replacement is defined as:

$$
\mathbf{c}^V(v_i, v'_a) = \begin{cases} -\log \frac{(d^V-1)\alpha+1}{d^V} & \text{if } v_i = v'_a \\ -\log \frac{-\alpha+1}{d^V} & \text{otherwise,} \end{cases}
$$

$$
\mathbf{c}^E(e_{ij}, e'_{ab}) = \begin{cases} -\log \frac{(d^E-1)\alpha+1}{d^E} & \text{if } e_{ij} = e'_{ab} \\ -\log \frac{-\alpha+1}{d^E} & \text{otherwise,} \end{cases}
$$

where $d^V$ and $d^E$ denote the cardinality of $\mathcal{X}^V$ and $\mathcal{X}^E$, respectively. Similar to Equation (19), we can formulate it to a quadratic problem with the objective function $f$ as:

$$
\begin{aligned}
f(\mathbf{x}) &= \sum_{\mathbf{x}_{ia}=1, \mathbf{x}_{jb}=1} \mathbf{c}^E(e_{ij}, e'_{ab}) + \sum_{\mathbf{x}_{ia}=1} \mathbf{c}^V(v_i, v'_a), \\
&= \sum_{\mathbf{x}_{ia}=1, \mathbf{x}_{jb}=1} -\log P^E_{0:\tau}(e_{ij}, e'_{ab}) + \sum_{\mathbf{x}_{ia}=1} -\log P^V_{0:\tau}(v_i, v'_a), \\
&= p(\sigma \mathbf{G}'|\mathbf{G}),
\end{aligned}
$$

where, the $\sigma$ is the graph permutation associated to the assignment vector $\mathbf{x}$.

However, the GED is not same to the NLL. Note that the edit cost functions penalize every operations with same cost $-\log \frac{-\alpha+1}{d}$ except for the identity operation. Though the cost of identity operation is lesser than the others, the optimal transformation would not contains any identity operation. In real, the cost of identity operation does not affect the optimal edit path, implying that GED is not equal but proportion to the NLL, where the difference proportional to the number of the nodes and edges that are equal under the optimal graph matching $\sigma^*$. Though the GED is not exactly same to the NLL, the problem is equivalent to the graph matching problem.

This observation provides a clear interpretation of the underlying dynamics of the SB problem, revealing that the associated OT cost is effectively the GED. Therefore, solving the SB problem can be understood as finding the OT plan between graph distributions, where the transport cost is defined by the GED.

# E   IMPLEMENTATION DETAILS

## E.1   PARAMETERIZATION

In this section, we briefly describe the neural network parameterization and the practical training loss. According to Equations (5) and (6), the neural network approximates the generator $A_t^{\Lambda \cdot |0}(x, y)$ and $\tilde{A}_t^{\Lambda \cdot |\tau}(y, x)$, which are formulated as conditional expectation of $A(x, y; z)$ and $\tilde{A}(y, x; z)$, respectively (see Lemma B.5).

The target generator takes the form:

$$
A_s(x, y; z) = \begin{cases} A_s(x, y) \frac{P_{s:\tau}(y, z)}{P_{s:\tau}(x, z)}, & \text{if } x \neq y \\ -\sum_{u \neq x} A_s(x, u) \frac{P_{s:\tau}(u, z)}{P_{s:\tau}(x, z)}, & \text{if } x = y \end{cases} \tag{21}
$$

, where $A_s$ and $P_{s:\tau}$ are tractable attributes of $\mathbb{Q}$. Let the the transition probability matrix be $P(s,\tau) : [0,\tau]^2 \rightarrow \mathbb{R}^{|\mathcal{X}| \times |\mathcal{X}|}$, with $P_{s:\tau}(x,y) = \mathbf{e}_x^\top P(s,\tau) \mathbf{e}_y$. Thus, the neural network predicts the distribution of $z$ given $X_s = x$, denoted as $z_\theta(s,x) \in \mathbb{R}^{|\mathcal{X}|}$, and our parameterization choice is as follows:

$$A_s^{M^\theta}(x,y;\theta) = A_s(x,y) \frac{\mathbf{e}_y^\top P(s,\tau) z_\theta(s,x)}{\mathbf{e}_x^\top P(s,\tau) z_\theta(s,x)},$$

for $y \neq x$. The time reverse generator is also defined similarly.

Though the loss formulation was defined as continuous manner, we approximate it with the discretizations. Firstly, we will replace $A_t(X_t, X_t; z) - A_t^{M^\theta}(X_t, X_t)$ with as follows:

$$A_t(X_t, X_t; z) - A_t^{M^\theta}(X_t, X_t) \approx \frac{1}{\Delta t}(1 + A_t(X_t, X_t; z)\Delta t) \log \frac{1 + A_t(X_t, X_t; z)\Delta t}{1 + A_t^{M^\theta}(X_t, X_t)\Delta t},$$

$$\approx \frac{1}{\Delta t} P_{t:t+\Delta t}^{\mathbb{Q} \cdot | \tau = z}(X_t, X_t) \log \frac{P_{t:t+\Delta t}^{\mathbb{Q} \cdot | \tau = z}(X_t, X_t)}{P_{t:t+\Delta t}^{M^\theta}(X_t, X_t)}.$$

Similarly,

$$A_t(x,y;z) \log \frac{A_t(x,y;z)}{A_t^{M^\theta}(x,y)} \approx \frac{1}{\Delta t} P_{t:t+\Delta t}^{\mathbb{Q} \cdot | \tau = z}(x,y) \log \frac{P_{t:t+\Delta t}^{\mathbb{Q} \cdot | \tau = z}(x,y)}{P_{t:t+\Delta t}^{M^\theta}(x,y)}.$$

Thus the discretized loss function corresponding to Equation (5) is:

$$L(\theta) = \sum_{t_i} \frac{1}{\Delta t} \mathbb{E}_{\Lambda_{t_i}, \tau} \left[ \sum_y P_{t:t+\Delta t}^{\mathbb{Q} \cdot | \tau = z}(x,y) \log \frac{P_{t:t+\Delta t}^{\mathbb{Q} \cdot | \tau = z}(x,y)}{P_{t:t+\Delta t}^{M^\theta}(x,y)} \right].$$

The time reversed loss can be discretized as same way.

## E.2 DATA PROCESSING FOR ZINC250K DATASET EXPERIMENT

In Section 5.2, we experimented with the standard ZINC250K dataset. In our molecular graph representation, $\mathbf{G} = (\mathbf{V}, \mathbf{E})$, the node vectors $\mathbf{V} = (v^{(i)})_i$ represent the atomic types, and the edge vectors $\mathbf{E} = (e^{(ij)})_{ij}$ represent bond orders. Due to this implementation choice, node features other than the atomic type cannot be represented, causing occasional failures in decoding into molecules. Still, to focus on the SB problem itself, we filtered out molecules whose graph representations are not directly converted into molecules by the RDKit package. Among the node features, formal charge and explicit hydrogen information is crucial in decoding process since, without them, each atom's valency cannot be inferred, so that corresponding molecule cannot be uniquely determined. Thus, the following criteria were applied to filter molecules from the ZINC250K dataset: (1) all atoms do not possess a formal charge and (2) all aromatic atoms do not have an explicit hydrogen. Our final training and test dataset contain 23,936 and 5,984 molecule pairs, respectively.

## E.3 NEURAL NETWORK PARAMETERIZATION AND HYPERPARAMETER SETTINGS

**DDSBM and DBM.** Our neural network parameterization is based on Vignac et al. (2022), which uses a graph transformer network (Dwivedi & Bresson, 2020). In short, it takes as input a noisy graph $\mathbf{G}_t = (\mathbf{V}_t, \mathbf{E}_t)$ and predicts a distribution over the target graphs. Structural and spectral features are also used as inputs to improve the expressivity of neural networks. We refer the reader to Vignac et al. (2022) for more details.

For noise scheduling, we employ a slightly different strategy than Vignac et al. (2022). While Vignac et al. (2022) uses a cosine schedule for $\bar{\alpha}_t$, we implement a symmetric scheduling of $\alpha_t$ by incorporating $\alpha_{\min}$, as defined below:

$$\alpha(t) = \frac{\partial_t \bar{\alpha}(t)}{\bar{\alpha}(t)} = \cos \left( \frac{t/\tau + s}{1 + s} \cdot \frac{\pi}{2} \right)^2 \cdot (1 - \alpha_{\min}) + \alpha_{\min}, \tag{22}$$

for $0 \leq t \leq \tau/2$, with $\alpha(t) = \alpha(\tau - t)$ for the remaining half of the schedule. For instance, with 100 diffusion steps, $\bar{\alpha}(\tau) \approx 0.95$ when $\alpha_{\min} = 0.999$, and $\bar{\alpha}(\tau) \approx 0.90$ when $\alpha_{\min} = 0.99795$.

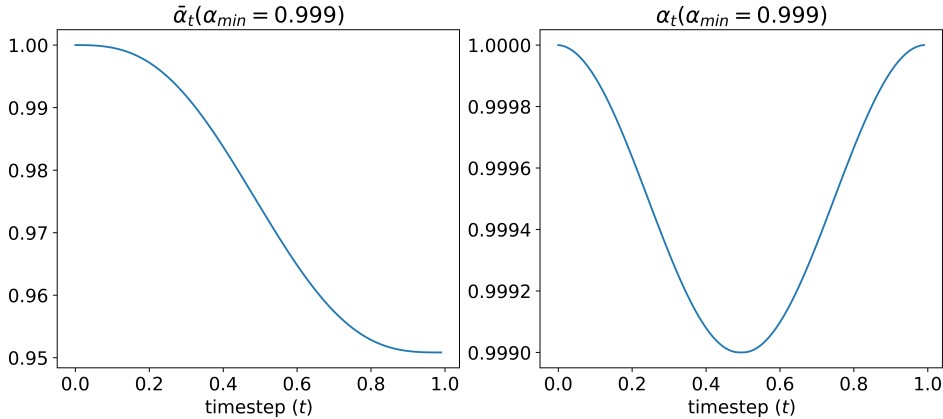

Figure 9: **Plot of $\alpha(t)$ and $\bar{\alpha}(t)$ as functions of timestep $t$ where $\alpha_{\mathbf{min}} = 0.999$**

We trained DDSBM models with IMF iterations, and DBM models, an one-directional variant of DDSBM with fixed joint molecular pairs, were trained with the same number of gradient updates to ensure consistency. Both DBM and DDSBM reported in this work were trained using four RTX A4000 GPUs. The detailed hyperparameters for training are shown in Table 4.

Table 4: **Training hyperparmeters of DBM and DDSBM for graph transformation.**

| Task | Model | diffusion steps | $\alpha_{\min}$ | epoch | SB iterations |
|------|-------|-----------------|-----------------|-------|---------------|
| ZINC250K | DBM | 100 | 0.999 | 1800 | – |
| ZINC250K | DDSBM | 100 | 0.999 | 300 | 6 |
| Polymer | DBM | 100 | 0.999 | 1250 | – |
| Polymer | DDSBM | 100 | 0.999 | 250 | 5 |

**Graph-to-Graph Translation.** For HierG2G and AtomG2G, we used the default settings provided on the official GitHub repository. Both models were trained until maximum epochs by default with a single RTX A4000 GPU.

### E.4 DETAILS ABOUT METRICS

We here provide detailed explanations about each metric used in the results section. They are classified into two groups; the first is to evaluate the marginal distribution of the generated data (data-wise) and the other is to evaluate the joint distribution between initial and generated data (pair-wise). Among the metrics explained below, **Validity**, **Uniqueness**, **Novelty**, **NSPDK**, $W_1$, and **FCD** are metrics that evaluate the marginal distributions, while **NLL** and **MAD** are metrics that evaluate the joint distribution.

#### E.4.1 BASIC METRICS

- **Validity:** Measures the proportion of chemically valid molecules generated by the model. A valid molecule adheres to fundamental chemical rules, such as valency constraints.

- **Uniqueness:** Represents the fraction of unique molecules in the generated set. This metric ensures the diversity of the generated molecules and prevents redundancy.

- **Novelty:** Quantifies the fraction of generated molecules that are not present in the training dataset. This evaluates the ability of the model to explore new regions of the chemical space.

### E.4.2 GRAPH STRUCTURAL METRICS

- **Negative Log-Likelihood (NLL):** Evaluates the quality of the joint distribution by measuring the cost of transforming one graph into another based on the reference process. This metric is closely related to the graph edit distance (GED), which quantifies the minimal number of modifications required to transform one graph into another (see Appendix D.6).

- **Neighborhood Subgraph Pairwise Distance Kernel (NSPDK):** Computes the distance between graph distributions using mean maximum discrepancy (MMD). NSPDK evaluates similarity based on local neighborhood subgraphs, which incorporate node and edge features as well as the underlying graph structure (Costa & Grave, 2010).

### E.4.3 MOLECULAR PROPERTIES METRICS

- **Wasserstein-1 Distance ($W_1$):** Measures the distance between the property distributions of the target and generated molecules for the properties to be modified, focusing on the marginal distribution. This metric reflects how closely the generated molecular properties match the desired target distribution.

- **Mean Absolute Difference (MAD):** Quantifies the average absolute difference in key properties within each molecular pair of initial and generated molecules. Low MAD value ensures that critical molecular properties such as drug-likeness (QED) and synthetic accessibility score (SAscore) remain consistent during graph transformation (Bickerton et al., 2012; Ertl & Schuffenhauer, 2009).

- **Fréchet ChemNet Distance (FCD):** Calculates the similarity between the target molecule distribution and the generated molecules, focusing on the marginal distribution of the molecules (Preuer et al., 2018). It is based on comparing the feature distributions of molecules embedded using a pre-trained neural network.

## F SUPPLEMENTARY RESULTS

### F.1 MORE STATISTICAL RESULTS FOR MAIN EXPERIMENTS

Tables 5 and 6 show average and standard deviation for all the metrics of ZINC and polymer experiments from three different training runs, respectively. For clarity, we plot the raw values of each molecular property—$\log P$, QED, and SAscore—in Figure 10.

Table 5: **Distribution shift performance on ZINC.** ↑ and ↓ denote higher and lower values are better, respectively. Standard deviation values for three independent runs included compared to the original table in the main result.

| Model | Val.(↑) | Uniq.(↑) | Nov.(↑) | NLL(↓) | NSPDK(↓) | LogP $W_1$(↓) | QED MAD(↓) | SAscore MAD(↓) | FCD(↓) |
|---|---|---|---|---|---|---|---|---|---|
| Reference | - | - | - | 360.862 | 1.47e-4 | 2.007 | 0.153 | 0.595 | 4.807 / 0.279 |
| AtomG2G | 99.9±0.1 | 64.4±2.6 | 99.3±0.1 | 355.025±0.484 | 9.70e-3±2.01e-3 | 0.162±0.034 | 0.143±0.001 | 0.697±0.019 | 5.019±0.572 |
| HierG2G | **100.0**±0.0 | 73.7±1.3 | 99.5±0.1 | 344.458±4.454 | 2.10e-2±4.75e-3 | **0.113**±0.045 | 0.146±0.005 | 0.687±0.032 | 5.742±0.378 |
| DBM | 87.6±1.2 | **100.0**±0.0 | **100.0**±0.0 | 288.572±3.327 | 8.04e-4±2.23e-4 | 0.150±0.012 | 0.141±0.002 | 0.608±0.013 | 1.046±0.043 |
| DDSBM | 94.8±1.9 | **100.0**±0.0 | 99.9±0.0 | **160.461**±10.409 | **7.30e-4**±9.29e-5 | 0.139±0.003 | **0.120**±0.009 | **0.402**±0.028 | **0.833**±0.082 |

Table 6: **Distribution shift performance on polymer.** ↑ and ↓ denote higher and lower values are better, respectively. Standard deviation values for three independent runs included compared to the original table in the main result.

| Model | Val.(↑) | Uniq.(↑) | Nov.(↑) | NLL(↓) | NSPDK(↓) | GAP $W_1$(↓) | FCD(↓) |
|---|---|---|---|---|---|---|---|
| Reference | - | - | - | 749.800 | 5.64e-4 | 0.312 | 1.469 / 0.384 |
| DBM | 43.4±0.015 | **99.8**±0.002 | **97.4**±0.003 | 580.415±11.279 | 5.82e-3±3.62e-r | 0.249±0.019 | 2.230±0.175 |
| DDSBM | **97.4**±0.004 | 94.5±0.011 | 71.3±0.038 | **212.047**±18.444 | **4.18e-3**±1.55e-4 | **0.127**±0.013 | **1.074**±0.038 |

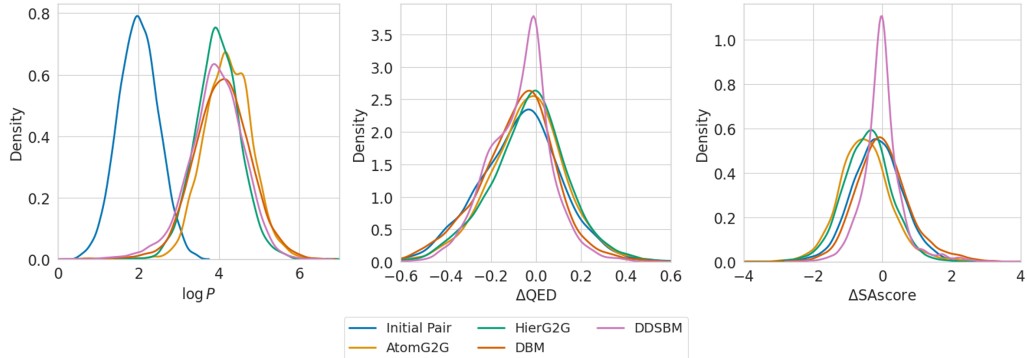

Figure 10: **Density plots comparing the distributions of three molecular properties (log *P*, ΔQED, and ΔSAscore) across four different molecular generation methods: AtomG2G, HierG2G, DBM, and DDSBM.**

## F.2 ABLATION STUDIES

We conducted ablation studies about the effects of graph matching algorithm and initial coupling of data on the distribution shift performance and the convergence of IMF iteration. Note that the former and the latter affect graph vectors and graphs, respectively.

### F.2.1 THE EFFECT OF GRAPH MATCHING ALGORITHMS

We first conducted an experiment about graph matching algorithms' effects on the performance of DDSBM. As Table 7 shows, the performance variations across graph matching algorithms were not significant. We attribute this outcome to the inherent characteristics of the DDSBM framework. As the SB iterations progress, the differences between initial and generated graphs would be decreased, in other words, they become increasingly similar. This is due to our reference process employs non-zero $\bar{\alpha}_\tau$ to assign low probabilities to dissimilar data pairs. Consequently, the influence of the graph matching algorithm on the optimality of graph data for each Markovian projection gradually diminishes, as supported by the Figure 11. This effect causes the model to converge in a similar manner regardless of the specific graph matching algorithm applied at each IMF iteration, resulting in minor differences on the final values as shown in the Table 7.

Table 7: **Ablation study on graph matching.** ↑ and ↓ denote higher and lower values are better, respectively.

| Algorithm | Val.(↑) | Uniq.(↑) | Nov.(↑) | NLL(↓) | NSPDK(↓) | LogP $W_1$(↓) | QED MAD(↓) | SAscore MAD(↓) | FCD(↓) |
|---|---|---|---|---|---|---|---|---|---|
| (1) SM | 94.6 | **100.0** | 99.9 | 163.931 | 1.17e-3 | 0.159 | 0.122 | **0.388** | 0.837 |
| (2) MPM | **95.8** | **100.0** | **100.0** | **155.378** | **6.95e-4** | 0.164 | **0.110** | 0.401 | 0.773 |
| (3) MPM + Randomness | 95.3 | **100.0** | 99.9 | 158.534 | **6.78e-4** | **0.130** | 0.116 | 0.424 | 0.759 |
| (4) Our setting | 94.1 | **100.0** | 99.9 | 164.480 | 7.49e-4 | 0.139 | 0.124 | 0.392 | **0.747** |

Furthermore, we additionally examined two scenarios on our main expeirment with the ZINC250K dataset: 1) training *with* graph matching during the IMF iterations, and 2) training *without* any graph matching algorithm during the whole training process. For the former, the graph matching setup used in our main ZINC experiment in Section 5.2 was used without change. Figure 12 shows that, similar to the comparison of several graph matching algorithm, the use of graph matching merely affected the convergence of DDSBM after a sufficient number of IMF iterations. Still, we observed that, without graph matching, both the training loss and the negative log-likelihood (NLL) between the original and generated data start at higher values and require more iterations to reach levels comparable to those achieved with graph matching. This indicates that graph matching algorithms, while not essential for the eventual convergence of DDSBM, can accelerate the convergence through the IMF iterations, providing an additional benefit.

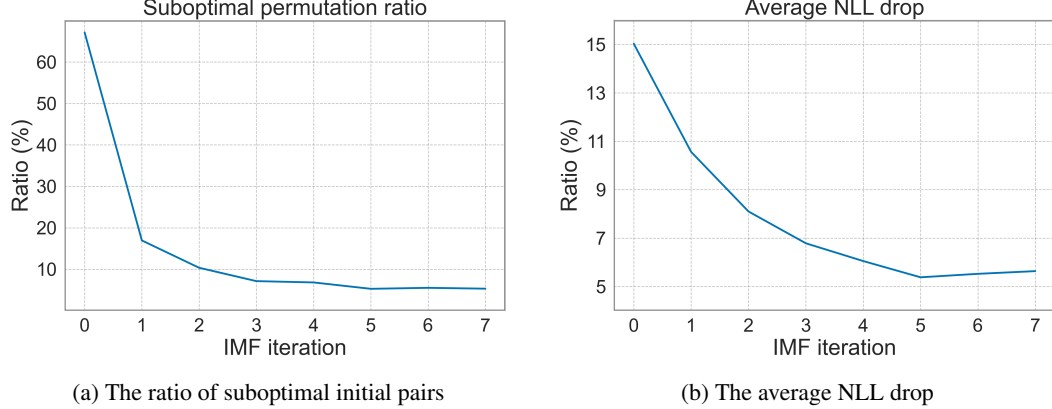

(a) The ratio of suboptimal initial pairs      (b) The average NLL drop

Figure 11: **The effect of graph matching algorithm on graph permutation matching versus the IMF iterations.** The result is computed from training data from a single run of our graph matching algorithm, which updated during an IMF iteration. (a) The ratio of suboptimal initial permutation pairs, found by the graph matching algorithm. For each iteration, this amount of graph data pairs are modified in their permutations. (b) The average NLL drop percentage for the data pairs whose permutations were changed by the graph matching algorithm, i.e. $(\text{NLL}_{\text{init}} - \text{NLL}_{\text{new}})/\text{NLL}_{\text{init}}$.

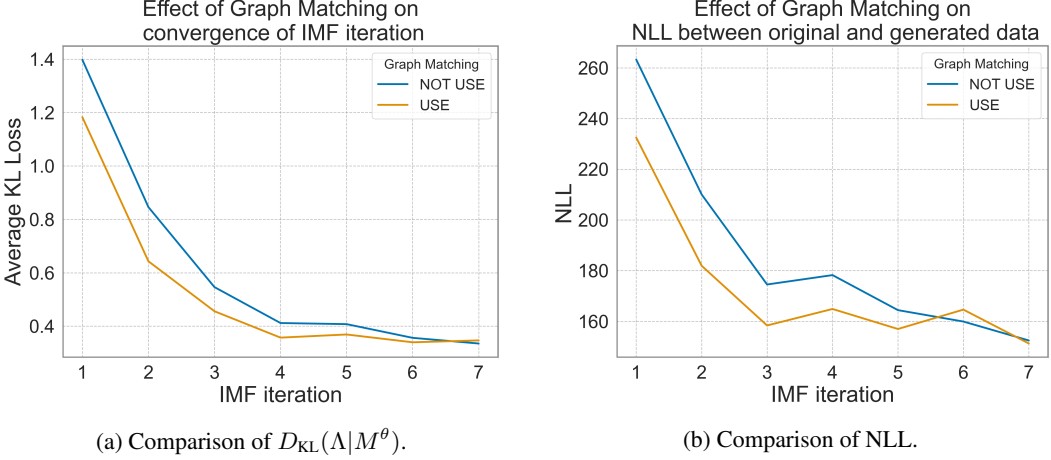

(a) Comparison of $D_{\text{KL}}(\Lambda|M^{\theta})$.      (b) Comparison of NLL.

Figure 12: **Comparison of training loss, $D_{\text{KL}}(\Lambda|M^{\theta})$, and NLL between original and generated data across IMF iterations of experiment setup with and without graph matching algorithm.** Here for graph matching, we used algorithm (4) in Table 7. Also, note that NLL in Figure 12b is calculated directly from original data and generated data without permutation alignment.

### F.2.2 THE EFFECT OF INITIAL COUPLING

We further analyze the molecular optimization task in Section 5.2 with different initial couplings. In that section, we used randomly coupled molecules as the initial coupling for all the models. However, except for DDSBM, all of them assume that suitable molecule pairs have already been identified to be provided. To meet this, we adopted Tanimoto similarity (Willett et al., 1998) as a pseudo-metric to find similar molecule pairs between the two molecule distributions, denoted as Tanimoto similarity-based coupling. We note that the similarity-based coupling is another optimal transport problem of *maximizing* the sum of pair-wise molecular similarities, where we employed the Hungarian method to obtain a sub-optimal solution.

Using the Tanimoto similarity-based coupling, we retrained all models discussed in Section 5.2 and compared their performance. Here, we denote a newly trained DDSBM model as DDSBM-T to deviate it from the model trained on the randomly coupled data. From Table 8, we observe that all models achieved lower NLL values compared to when they were trained with random coupling. The

HierG2G model exhibits much lower FCD and NLL values compared to those of random coupling, indicating that previous graph transformation methods can be improved if more optimal pairs are provided as training data.

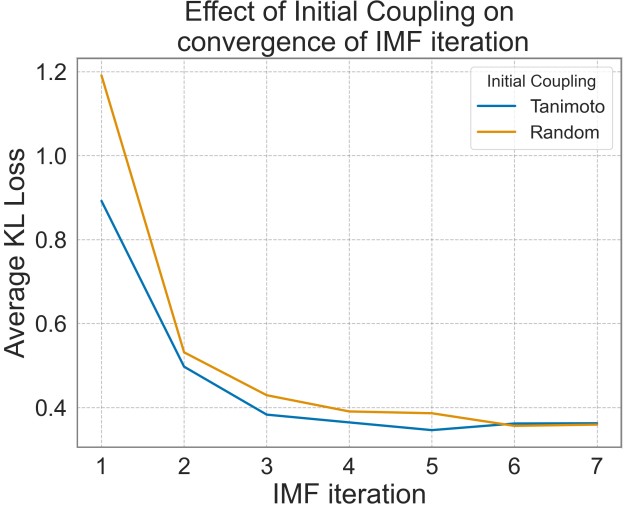

Figure 13: **Comparison of $D_{\mathbf{KL}}(\Lambda|M^\theta)$ across IMF iterations for two types of initial couplings: Random and Tanimoto similarity.**

The distinct feature of DDSBM-T is that all baseline models learn graph transformations between the molecule pairs with high Tanimoto similarity, while DDSBM-T learns graph transformations with minimal cost defined by its reference process $\mathbb{Q}$. It cannot be guaranteed that the distance defined by our reference process is better than the Tanimoto similarity from the perspective of molecular optimization. Essentially, the success of molecular optimization should be measured by how well the target property is adjusted while preserving other key properties. Apparently, Table 8 shows that DDSBM-T outperforms the other baselines in terms of molecular property metrics. This suggests that the graph transformation from DDSBM retains other molecular properties, attaining the goal of the molecule optimization task.

Finally, we analyze the effect of initial coupling on the training process, especially focusing on the approach to convergence. We illustrate the training losses of DDSBM models from two different initial couplings, random and Tanimoto similarity-based, in Figure 13, respectively. DDSBM-T shows consistently lower loss values up to the sixth IMF iteration and reaches convergence at the third IMF iteration.

Table 8: **Distribution shift performance on ZINC with initial coupling based on the Tanimoto similarity.** As in Table 1, reference refers to metrics from the initial coupling, used as a standard to evaluate each model's graph translation. The experimental setting is the same as described Section 5.2, except for the initial coupling. ↑ and ↓ denote higher and lower values are better, respectively.

| Model | Type | Val.(↑) | Uniq.(↑) | Nov.(↑) | NLL(↓) | NSPDK(↓) | LogP $W_1$(↓) | QED MAD(↓) | SAscore MAD(↓) | FCD(↓) |
|---|---|---|---|---|---|---|---|---|---|---|
| Reference[1] | - | - | - | - | 245.765 | 1.63e-4 | 2.011 | 0.126 | 0.367 | 4.811 / 0.315 |
| AtomG2G | Latent | **100.0** | 99.8 | **99.9** | 289.674 | 5.22e-3 | 0.315 | 0.144 | 0.558 | 1.578 |
| HierG2G | Latent | **100.0** | 99.7 | **99.9** | 264.072 | 1.33e-3 | 0.189 | 0.127 | 0.446 | 1.171 |
| DBM | Bridge | 90.2 | **100.0** | **99.9** | 220.594 | **5.91e-4** | 0.141 | 0.127 | 0.508 | **0.749** |
| DDSBM-T | Schrödinger Bridge | 95.6 | **100.0** | **99.9** | **152.856** | 6.23e-4 | **0.103** | **0.110** | **0.393** | 0.911 |

[1] NLL, $W_1$, and MADs were calculated using random pairs from the test set. Two FCD values are provided: the first compares the initial molecules in the test set with the terminal molecules in the training set, and the second compares the terminal molecules in both sets. Also, the reference NSPDK is computed with the terminal molecules from training and test sets.

## F.3 EXAMPLES FOR GENERATED MOLECULES ON MOLECULE OPTIMIZATION TASKS

To show the difference between molecules generated from DDSBM and others, we visualized some selected examples for the ZINC and polymer datasets, respectively (see Figures 14 and 15). As

expected from the superior performance of DDSBM compared to other methods in terms of joint distribution metrics (NLL and properties MAD), the generation result from DDSBM gives more similar graph structure compared to source molecules in both ZINC and polymer datasets. We also present generation trajectories for different molecules in ZINC and polymer datasets in Figures 16 and 17.

Figure 14: **Visualization of molecules generated by DDSBM, DBM, and HierG2G compared to the source molecule.**

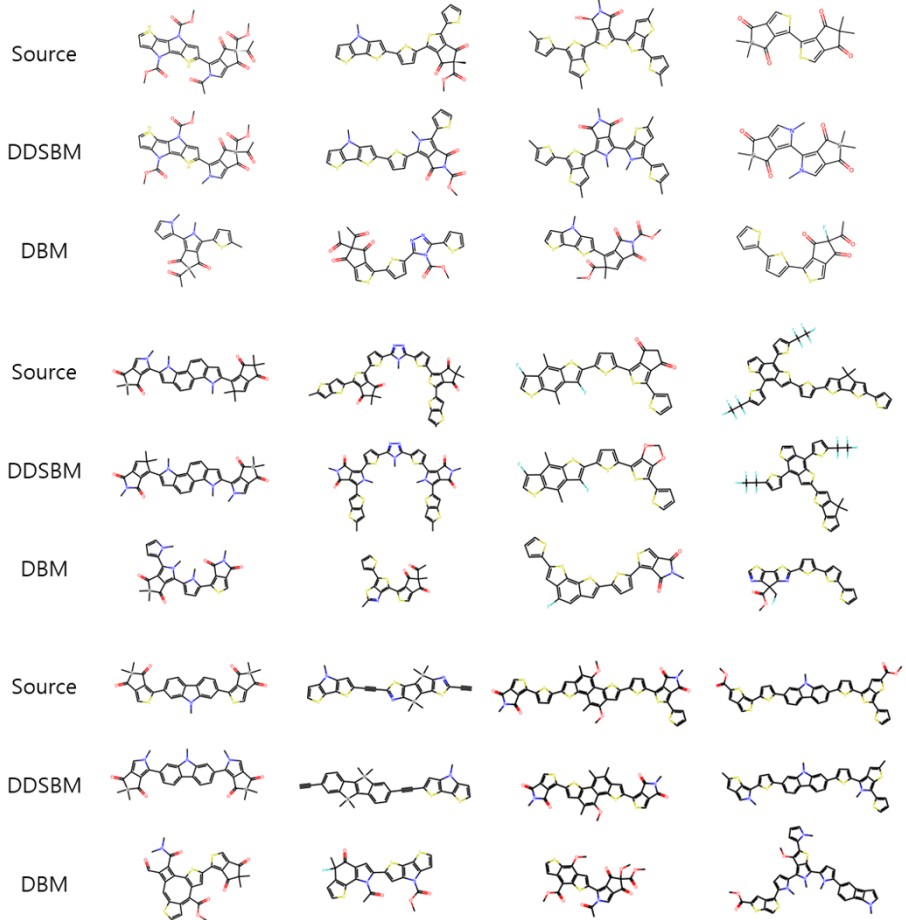

Figure 15: **Visualization of molecules generated by DDSBM and DBM with the source molecule.** The samples generated by DBM and DDSBM were selected from the molecules predicted to have a blue color with GAP values in the range of 2.56–2.75 eV.

## G UNCONDITIONAL GRAPH GENERATION

### G.1 EXPERIMENTAL SETUP AND METRICS

We conducted an additional evaluation of DDSBM on an unconditional graph generation task. To ensure consistency, we adopted the same hyperparameter settings and training configurations as in DiGress (Vignac et al., 2022), except for the noise scheduling as detailed in Table 4. Training, validation, and test splits were same as DiGress, but early stopping based on the validation set was not employed, similar to the main tasks. The training hyperparameters are detailed in Table 9.

For each task, both the DDSBM and DBM were trained for the same number of epochs, with sampling from the prior distribution repeated five times. The results were averaged across the five for the Maximum Mean Discrepancy (MMD) of graph features (degree distributions, clustering coefficients, and orbit counts). Additionally, for the planar and QM9 datasets, further metrics such as Validity/Uniqueness/Novelty (V.U.N.) were evaluated.

### G.2 SYNTHETIC GRAPH GENERATION

In the Community-20 task, both DBM and DDSBM demonstrated superior performance compared to DiGress in terms of degree, clustering, and orbit metrics. This result highlights the strengths of

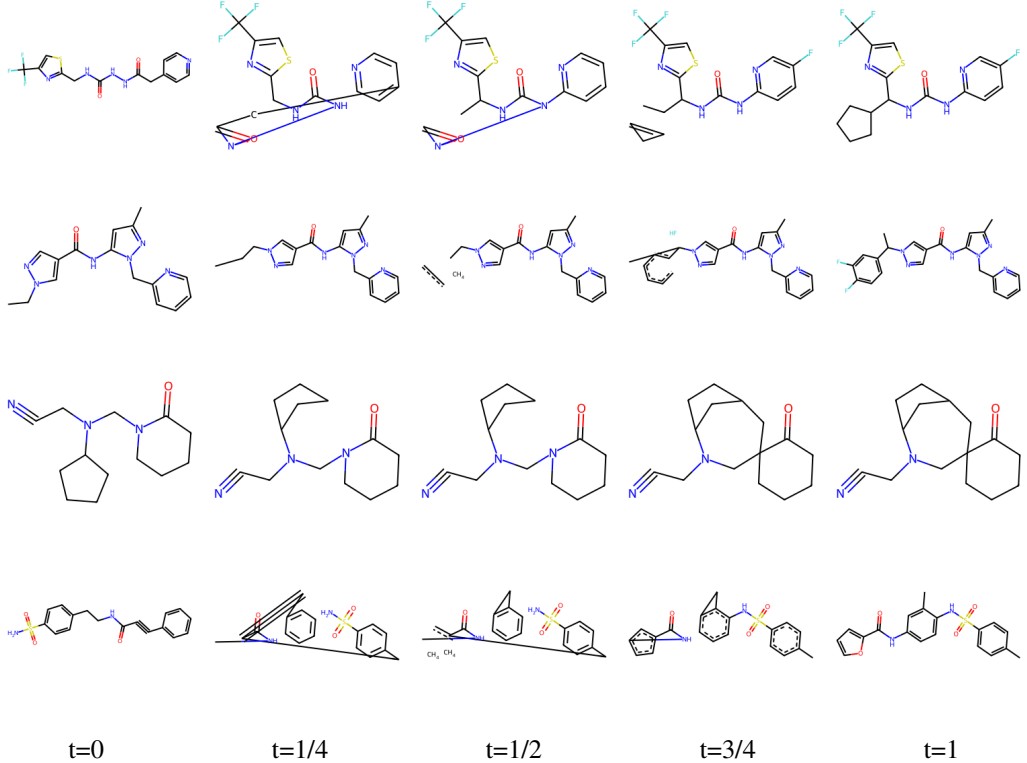

t=0                    t=1/4                    t=1/2                    t=3/4                    t=1

Figure 16: **Trajectory visualization of ZINC molecule optimization task generated by DDSBM.** Note that **t=0** and **t=1** are the original and generated molecules, respectively.

Table 9: **Training hyperparameters of DBM and DDSBM for unconditional graph generation**

| Task | Model | diffusion steps | $\alpha_{min}$ | epoch | SB iterations |
|------|-------|-----------------|----------------|-------|---------------|
| QM9 | DBM | 100 | 0.999 | 200 | – |
| QM9 | DDSBM | 100 | 0.999 | 100 | 2 |
| Community-20 | DBM | 500 | 0.9998 | 1000k | – |
| Community-20 | DDSBM | 500 | 0.9998 | 200k | 5 |
| Planar | DBM | 1000 | 0.9999 | 100k | – |
| Planar | DDSBM | 1000 | 0.9999 | 10k | 10 |

DBM and DDSBM in unconditional graph generation tasks as well. Notably, DDSBM achieved the best performance in clustering and competitive results in orbit while also significantly reducing the NLL value compared to DBM. These findings underscore the capacity of the DDSBM to generate high-quality graph structures at a minimal cost, even when bridging noisy distribution to data distribution.

### G.3 SMALL MOLECULAR GRAPH GENERATION

For the QM9 task, both DBM and DDSBM demonstrated significant improvements in FCD compared to DiGress, as reported in Nguyen et al. (2024). Additionally, enhancements in validity and novelty were also observed. While DBM showed slightly better performance in FCD and validity, DDSBM exhibited strengths in novelty compared to DBM. For other metrics, DDSBM and DBM displayed comparable performance.

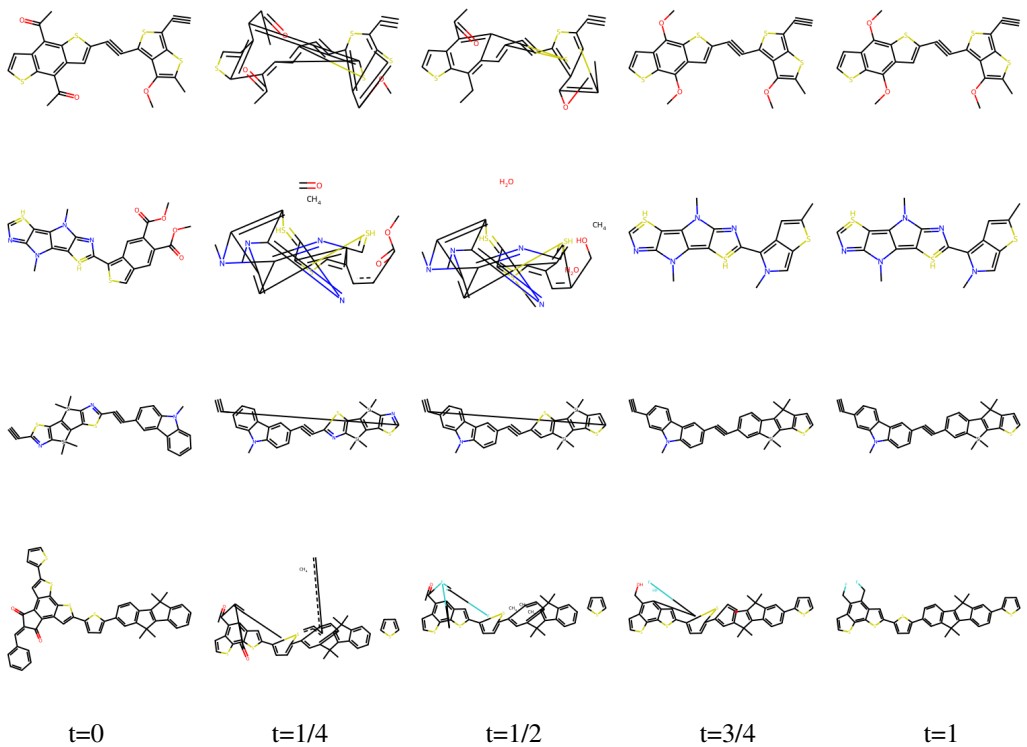

t=0            t=1/4            t=1/2            t=3/4            t=1

Figure 17: **Trajectory visualization of Polymer optimization task generated by DDSBM.** Note that **t=0** and **t=1** are the original and generated molecules, respectively.

Table 10: **Unconditional graph generation performance on Community-20.** ↑ and ↓ denote higher and lower values are better, respectively. The best performance is highlighted in bold, and the second-best performance is underlined.

| Model | Degree ↓ | Clustering ↓ | Orbit ↓ | NLL ↓ |
|---|---|---|---|---|
| GraphRNN[1] | 8.00e-2 | 1.19e-1 | 4.00e-2 | - |
| GRAN[1] | 6.00e-2 | 1.12e-1 | 1.00e-2 | - |
| GG-GAN[1] | 8.00e-2 | 2.17e-1 | 8.00e-2 | - |
| SPECTRE[1] | **1.00e-2** | 1.89e-1 | 2.00e-2 | - |
| DiGress[1] | 2.00e-2 | 6.30e-2 | 1.00e-2 | - |
| DBM | 1.80e-2 | 3.80e-2 | **5.14e-3** | 3.26e+2 |
| DDSBM | 1.75e-2 | **2.78e-2** | 5.81e -3 | **2.87e+2** |

[1] These results are taken from Vignac et al. (2022).

### G.4 EXAMPLES FOR GENERATED GRAPH ON UNCONDITIONAL GRAPH GENERATION

To illustrate how graph structure is preserved in DDSBM, we visualized a selection of examples from the Planar, Community-20, and QM9 unconditional generation tasks. For all three tasks, the generation trajectories were visualized in Figures 18 to 20. In the case of planar graphs, we plotted the initial and generated graphs using the trained DDSBM and the public checkpoints of Digress (see Figure 21).

Table 11: **Unconditional graph generation performance on Planar.** ↑ and ↓ denote higher and lower values are better, respectively. The best performance is highlighted in bold, and the second-best performance is underlined.

| Model | Degree ↓ | Clustering ↓ | Orbit ↓ | V.U.N. ↑ | NLL ↓ |
|---|---|---|---|---|---|
| GraphRNN[1] | 4.90e-3 | 2.79e-1 | 1.25e+0 | 0.0 | - |
| GRAN[1] | 7.00e-4 | 4.34e-2 | 9.00e-4 | 0.0 | - |
| SPECTRE[1] | 5.00e-4 | 7.75e-2 | 1.20e-3 | 25.0 | - |
| ConGress[1] | 4.76e-3 | 2.73e-1 | 1.30e+0 | 0.0 | - |
| DiGress[1] | **2.80e-4** | **3.72e-2** | **8.50e-4** | 75.0 | - |
| DBM | 9.35e-4 | 8.95e-2 | 8.67e-3 | **81.5** | 1.96e+3 |
| DDSBM | 7.39e-4 | 5.81e-2 | 9.60e-4 | 76.0 | **1.51e+3** |

[1] These results are taken from Vignac et al. (2022).

Table 12: **Unconditional graph generation performance on QM9.** ↑ and ↓ denote higher and lower values are better, respectively. The best performance is highlighted in bold, and the second-best performance is underlined.

| Model | Val. ↑ | Uniq. ↑ | Nov. ↑ | FCD ↓ | NLL ↓ |
|---|---|---|---|---|---|
| GraphAF[1] | 74.43 | 88.64 | 86.59 | 5.27e+0 | - |
| MoFlow[1] | 91.36 | **98.65** | 94.72 | 4.47e+0 | - |
| GraphDF[1] | 93.88 | 98.58 | **98.54** | 1.09e+1 | - |
| GDSS[1] | 95.72 | 98.46 | 86.27 | 2.90e+0 | - |
| GraphArm[1] | 90.25 | 95.62 | 70.39 | 1.22e+0 | - |
| GLAD[1] | 97.12 | 97.52 | 38.75 | 2.01e-1 | - |
| Digress[1] | 99.00 | 96.66 | 33.40 | 3.60e-1 | - |
| DBM | **99.87** | 96.59 | 36.54 | **9.90e-2** | 7.55e+1 |
| DDSBM | 99.67 | 96.83 | 38.06 | 1.05e-1 | **5.38e+1** |

[1] These results are taken from Nguyen et al. (2024).

### G.5 COMPARISON OF GENERATION PERFORMANCE ACROSS VARYING NFES

We compared the generation performance, specifically validity and FCD, of DDSBM and DBM on the QM9 task across different the number of function evaluations (NFEs), set at 5, 10, 20, 25, 50, and 100. The results show that DDSBM maintains performance reasonably well even at low NFEs, whereas DBM shows a more pronounced performance degradation under the same conditions (Figure 22). We attribute this to DDSBM finding optimal generation paths that can be predicted with fewer NFEs.

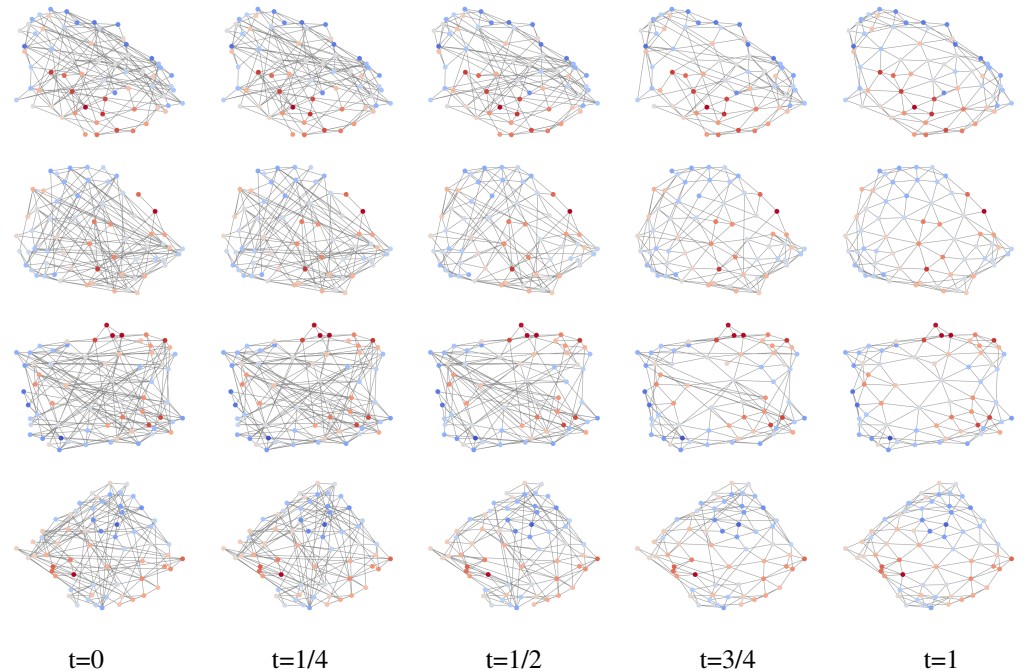

t=0       t=1/4       t=1/2       t=3/4       t=1

Figure 18: **Trajectory visualization of planar graph generated by DDSBM.** Note that **t=0** and **t=1** are the prior and generated graphs, respectively.

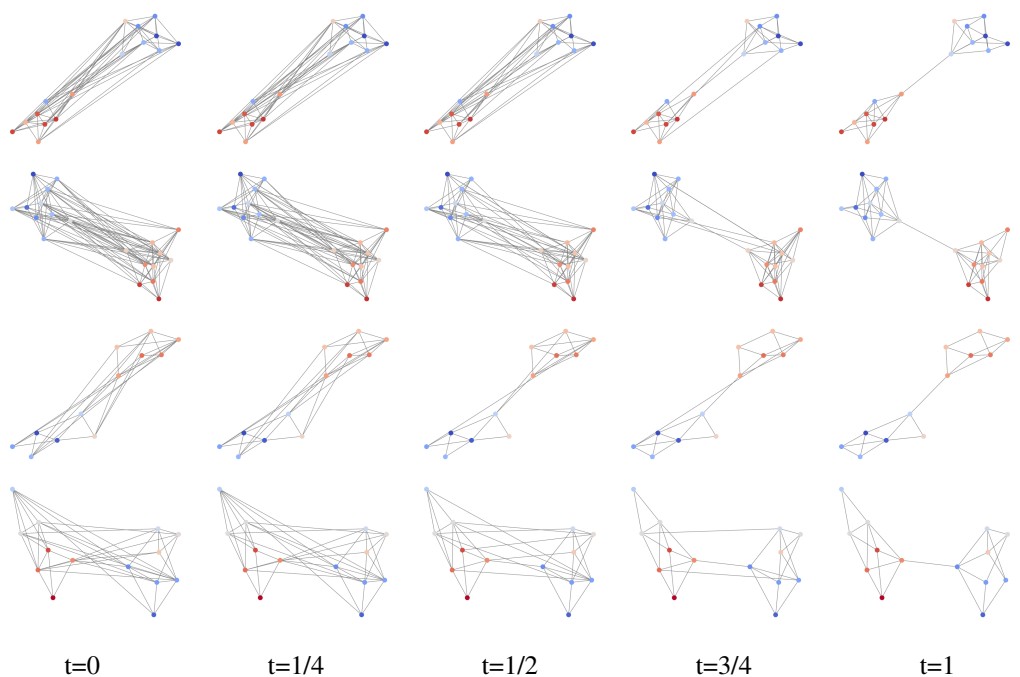

t=0       t=1/4       t=1/2       t=3/4       t=1

Figure 19: **Trajectory visualization of community-20 graph generated by DDSBM.** Note that **t=0** and **t=1** are the prior and generated graphs, respectively.

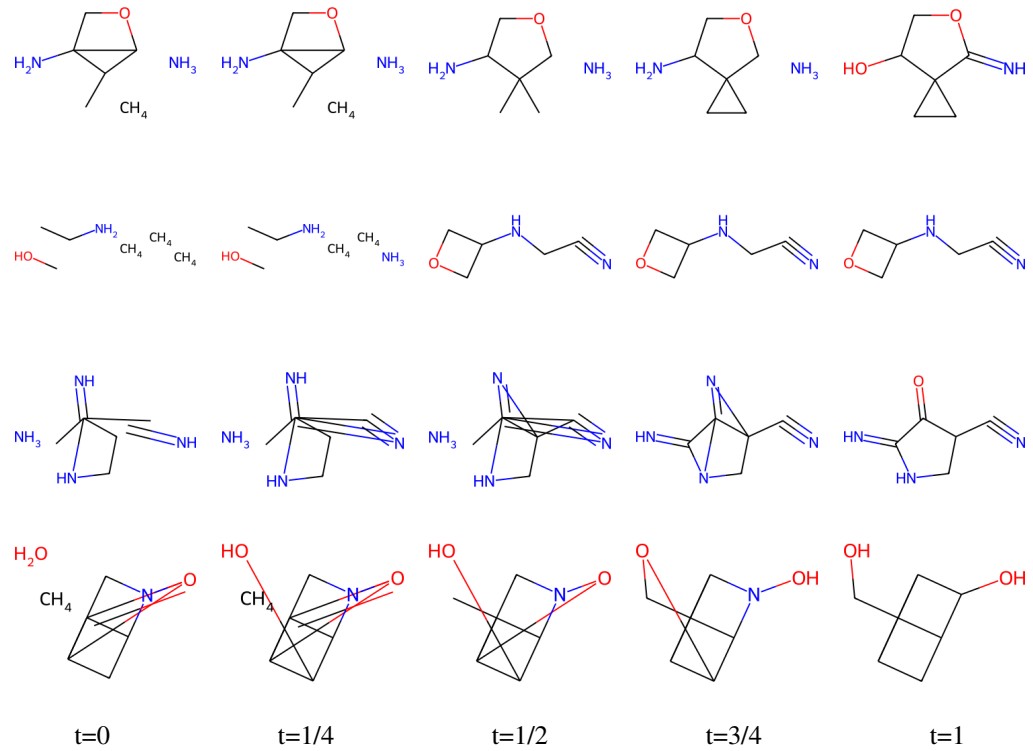

Figure 20: **Trajectory visualization of QM9 graph generated by DDSBM.** Note that **t=0** and **t=1** are the prior and generated graphs, respectively.

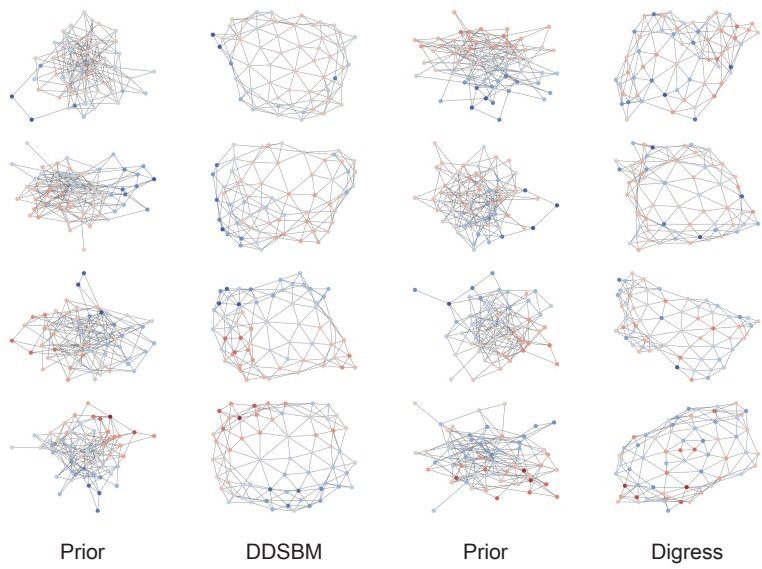

Figure 21: **Visualization of planar graph generated by DDSBM and Digress with the source prior graph.** The color of each node is assigned based on the spectral features of the prior graph. Graphs generated by DDSBM tend to better preserve the graph structures of the source prior graph compared to those generated by Digress.

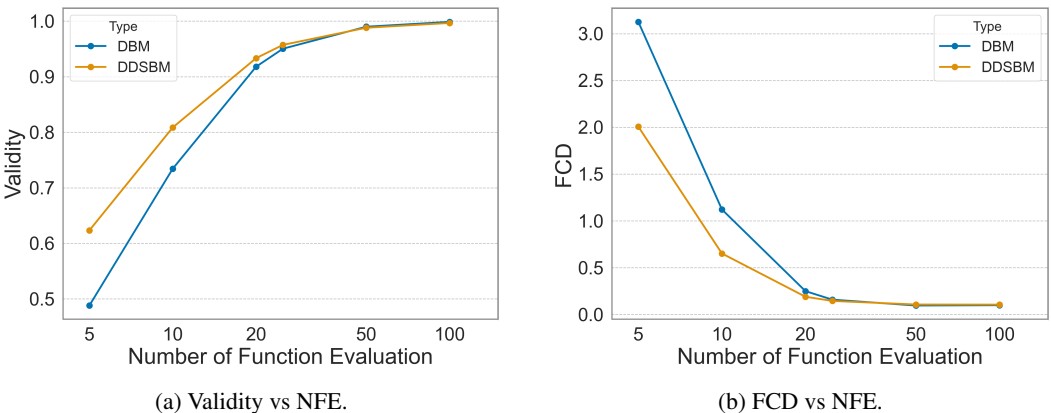

(a) Validity vs NFE.

(b) FCD vs NFE.

Figure 22: **Comparison of DBM and DDSBM in terms of generation performance with respect to number of diffusion steps (NFE) in QM9 experiment.** Here, we used NFE of 5, 10, 15, 20, 25, and 100 timesteps.

