# OpenReview forum: "Discrete Diffusion Schrödinger Bridge Matching for Graph Transformation"
_ICLR.cc/2025/Conference — ICLR 2025 Poster_

### Official Review · Reviewer_b65H · 2024-11-02

**Soundness:** 3
**Presentation:** 2
**Contribution:** 3
**Rating:** 6
**Confidence:** 4

**Summary:**

Adapt diffusion Schrodinger bridge matching to the discrete domain and instantiate it through molecular optimization

**Strengths:**

1. This method adapts diffusion Schrodinger bridge matching which opens a new avenue for this task.
2. The experimental setting for molecular optimization is a very relevant downstream task for the discrete domain, and it builds up some baselines for that task.
3. The work provides corresponding theoretical guarantees.

**Weaknesses:**

1. Globally, the structure is complete but the writing is not easy to follow. The writing can be enhanced at least in the following way:
* The square brackets around the citation should not appear twice.
* The name of section 2.2 'SOLUTION METHOD' needs to be optimized.
* The supplementary material also contains the body part of the paper.
* The structure section 2/3 is a bit confusing:
  * There should be an introduction to the continuous methods (IPF, IMF) as prior knowledge since the method highly dependent on them, then a transition to the method for the discrete domain. Here, section 2 starts by discussing the discrete domain and introducing the definitions together, which makes it a bit confusing which parts come from previous work and which part comes from this work.
  *  Consequently, the insights about the transition from continuous domain to discrete domain are missing.

2. Under the claim of proposing a discrete model for DSBM, the results only contain 2 molecule datasets with few supportive ablation experiments being given. It will support better the contribution of this work to give toy results for examples except for molecule optimization such as toy datasets with discrete features, or results on other types of graphs, or more ablations, or pure generation results (as in continuous SB papers), or more visualizations of the resulting optimization chain. No need to have them all, but similar supportive evidence would definitely make the experiments more complete.

**Questions:**

How significantly does the choice of graph-matching algorithm impact the overall performance of the method?
If it has a substantial influence, there is another reason that experiments for some simpler modalities without such complex matching may help to clarify the contribution.

---

> ### Author Response · Authors · 2024-11-13
>
> Thank you for your constructive feedback. We appreciate your detailed comments, as they provide valuable insights that will certainly help improve the quality of our work.
>
> W1: We acknowledge the need for clearer structure and presentation. We will revise the manuscript to clearly highlight our strengths. As soon as the revisions are complete, we will reply with an official comment.
>
> W2: We understand the importance of providing additional experimental evidence as proposing a discrete model for DSBM. To address this, we plan to extend our experiments on additional datasets proposed in SPECTRE [1] for graph generation task. The datasets include synthetic graph dataset (**Community-small**) and widely-used molecular dataset (**QM9**). Additionally, we will conduct graph generation on larger synthetic graph dataset (**Planar**), to further validate our approach. We will also include the generation trajectories for our previously conducted experiments to provide deeper insights. Specifically, we will compare DBM and DDSBM with DiGress in terms of degree, cluster and orbit for all datasets, while additionally evaluating VUN (valid, unique & novel graphs) for the planar dataset. We will share experimental results as they become available. Please let us know if there are any additional results or analyses that the reviewer would find helpful for a more comprehensive evaluation of DDSBM.
>
> W3: We will conduct experiments to compare the effects of graph matching algorithms displayed in Appendix table 4.
>
> We will provide an official comment once these revisions are finalized. Thank you again for your valuable feedback.
>
> Reference
>
> [1] Martinkus, Karolis, et al. "Spectre: Spectral conditioning helps to overcome the expressivity limits of one-shot graph generators." *International Conference on Machine Learning*. PMLR, 2022.

---

> ### Author Response · Authors · 2024-11-22
>
> Thank you for your thoughtful feedback. We deeply appreciate your detailed comments, as they provide valuable insights that will surely elevate the quality of our work.
>
> **[Weakness 1]** **Globally, the structure is complete but the writing is not easy to follow.**
>
> As the reviewer pointed out, there are indeed parts of the manuscript where readability should be improved. We greatly appreciate the suggestions for improving these aspects.
>
> After a careful review of the draft, we realized that the transition between the continuous setting and the extension to the discrete setting was not been clearly explained. Specifically, the beginning of Section 2 introduces notations primarily related to discrete càdlàg paths, while Section 2.1 follows with a description of the Schrödinger Bridge (SB) in the continuous setting. This lack of clarity hinders to adequately convey our main contribution, which is an “extension of the SB problem from the continuous to the discrete setting”.
>
> To address this, we are revising the manuscript to first provide a thorough explanation of previous work and solution methods for SB in the continuous setting, before extending to the discrete setting. In addition, we plan to move the related work section after the introduction to allow for a clearer comparison between previous work and our contribution. **We are actively working on improving the manuscript’s readability. Once these revisions are complete and the updated results are incorporated, we will upload the revised version and provide further comments**.
>
> **[Weakness 2]**
>
> As the reviewer mentioned, DDSBM is a framework designed to solve the Schrödinger Bridge problem specifically for discrete data, especially graph data. However, DDSBM's performance has not been evaluated on graph datasets outside of the chemical domain.
>
> Based on the reviewer’s valuable suggestion, we conducted an additional evaluation of DDSBM on a pure generation task. This evaluation was performed on the community-20, planar, and QM9 datasets, and the performance comparisons are summarized in the table below. For these additional experiments, we used the **DiGress [1]** graph transformer as the backbone network for DDSBM. To ensure consistency, we adopted the same hyperparameter settings and training configurations as in DiGress, except for the noise scheduling as detailed in Appendix C.2. Training, validation, and test splits were kept consistent across all tasks, but early stopping based on the validation set was not employed, similar to the main tasks (ZINC, Polymer).
>
> We also clarify the comparison between DDSBM and other baseline models, such as DBM and DiGress:
>
> - All models share the same backbone neural network.
> - Both DDSBM and DBM share the same reference diffusion process.
> - The key difference between DDSBM and DBM lies in the application of the IMF algorithm, while both involve the same Markov projection.
>
> For each task, both the DDSBM and DBM were trained for the same number of epochs, with sampling from the prior distribution repeated five times. The results were averaged across the five for the Maximum Mean Discrepancy (MMD) of graph features. Additionally, for the planar and QM9 datasets, further metrics such as Validity/Uniqueness/Novelty (V.U.N.) were evaluated.
>
> **[Results of Community-20]**
>
> \begin{array}{l|cccc}
> \text{Model} & \text{Degree} \downarrow & \text{Clustering} \downarrow & \text{Orbit} \downarrow & \text{NLL} \downarrow \newline
> \hline
> \text{GraphRNN} & 8.00\text{e-}02 & 1.19\text{e-}01 & 4.00\text{e-}02 & - \newline
> \text{GRAN} & 6.00\text{e-}02 & 1.12\text{e-}01 & 1.00\text{e-}02 & - \newline
> \text{GG-GAN} & 8.00\text{e-}02 & 2.17\text{e-}01 & 8.00\text{e-}02 & - \newline
> \text{SPECTRE} & \textbf{1.00e-02} & 1.89\text{e-}01 & 2.00\text{e-}02 & - \newline
> \text{DiGress} & 2.00\text{e-}02 & 6.30\text{e-}02 & 1.00\text{e-}02 & - \newline
> \hline
> \text{DBM} & 1.80\text{e-}02 & \underline{3.80\text{e-}02} & \textbf{5.14e-03} & \underline{3.26\text{e+}02} \newline
> \text{DDSBM} & \underline{1.75\text{e-}02} & \textbf{2.78e-02} & \underline{5.81\text{e-}03} & \textbf{2.87e+02} \newline
> \end{array}
>
> In the Community-20 task, both DBM and DDSBM demonstrated superior performance compared to DiGress in terms of degree, clustering, and orbit metrics. This result highlights the strengths of DBM and DDSBM in pure generation tasks as well. Notably, DDSBM achieved the best performance in clustering (2.78.E-02) and competitive results in orbit (5.81.E-03) while also significantly reducing the NLL value (2.87.E+02) compared to DBM. These findings underscore the capacity of the DDSBM to generate high-quality graph structures at a minimal cost, even when bridging noisy distribution to data distribution.

---

> ### Author Response · Authors · 2024-11-22
>
> **[Results of QM9]**
> \begin{array}{l|ccccc}
> \text{Model} & \text{Validity} \uparrow & \text{Uniqueness} \uparrow & \text{Novelty} \uparrow & \text{FCD} \downarrow & \text{NLL} \downarrow \newline
> \hline
> \text{GraphAF} & 74.43 & 88.64 & 86.59 & 5.27\text{e+}00 & - \newline
> \text{MoFlow} & 91.36 & \textbf{98.65} & \underline{94.72} & 4.47\text{e+}00 & - \newline
> \text{GraphDF} & 93.88 & \underline{98.58} & \textbf{98.54} & 1.09\text{e+}01 & - \newline
> \text{GDSS} & 95.72 & 98.46 & 86.27 & 2.90\text{e+}00 & - \newline
> \text{GraphArm} & 90.25 & 95.62 & 70.39 & 1.22\text{e+}00 & - \newline
> \text{GLAD} & 97.12 & 97.52 & 38.75 & 2.01\text{e-}01 & - \newline
> \text{DiGress} & 99.00 & 96.66 & 33.40 & 3.60\text{e-}01 & - \newline
> \hline
> \text{DBM} & \textbf{99.87} & 96.59 & 36.54 & \textbf{9.90e-02} & \underline{7.55\text{e+}01} \newline
> \text{DDSBM} & \underline{99.67} & 96.83 & 38.06 & \underline{1.05\text{e-}01} & \textbf{5.38e+01} \newline
> \end{array}
>
> For the QM9 task, both DBM and DDSBM demonstrated significant improvements in FCD compared to DiGress, as reported in reference [2]. Additionally, enhancements in validity and novelty were also observed. While DBM showed slightly better performance in FCD (9.90.E-02) and validity (99.87%), DDSBM exhibited strengths in novelty (38.06%) compared to DBM. For other metrics, DDSBM and DBM displayed comparable performance.
>
> **[Results of Planar]**
> \begin{array}{l|ccccc}
> \text{Model} & \text{Degree} \downarrow & \text{Clustering} \downarrow & \text{Orbit} \downarrow & \text{V.U.N} \uparrow & \text{NLL} \downarrow \newline
> \hline
> \text{GraphRNN} & 4.90\text{e-}03 & 2.79\text{e-}01 & 1.25\text{e+}00 & 0 & - \newline
> \text{GRAN} & 7.00\text{e-}04 & \underline{4.34\text{e-}02} & \underline{9.00\text{e-}04} & 0 & - \newline
> \text{SPECTRE} & \underline{5.00\text{e-}04} & 7.75\text{e-}02 & 1.20\text{e-}03 & 25 & - \newline
> \text{ConGress} & 4.76\text{e-}03 & 2.73\text{e-}01 & 1.30\text{e+}00 & 0 & - \newline
> \text{DiGress} & \textbf{2.80e-04} & \textbf{3.72e-02} & \textbf{8.50e-04} & 75 & - \newline
> \hline
> \text{DBM} & 9.35\text{e-}04 & 8.95\text{e-}02 & 8.67\text{e-}03 & \textbf{81.5} & \underline{1.96\text{e+}03} \newline
> \text{DDSBM} & 7.39\text{e-}04 & 5.81\text{e-}02 & 9.60\text{e-}04 & \underline{76} & \textbf{1.51e+03} \newline
> \end{array}
>
> In contrast to other tasks, we observed that DBM/DDSBM demonstrated  lower performance than DiGress on the Degree, Cluster, and Orbit metrics. We regard this to two primary reasons.
>
> Firstly, for DDSBM, the Planar task requires relatively more epochs to converge compared to other tasks. The values reported in the table correspond to the results from 100K epochs, which have not yet reached convergence. Second, we observed a slight trade-off between validity and the graph feature MMD metrics on each IMF iteration during training. We are currently continuing the training process for this task and will include the fully converged results in the manuscript at a later stage.
>
> As evidenced by the NLL metric, DDSBM involves minimal changes compared to DBM, which we also confirmed with the chain visualization. Following the reviewer's suggestion, we plan to include this in the paper and will incorporate it while revising the manuscript.
>
> We appreciate the reviewer’s understanding and patience as we refine our analysis for this specific task.
>
> **[Reference]**
>
> [1] Vignac, C., Krawczuk, I., Siraudin, A., Wang, B., Cevher, V., & Frossard, P. (2022). Digress: Discrete denoising diffusion for graph generation. arXiv preprint arXiv:2209.14734.
>
> [2] Nguyen, V. K., Boget, Y., Lavda, F., & Kalousis, A. (2024). GLAD: Improving Latent Graph Generative Modeling with Simple Quantization. arXiv preprint arXiv:2403.16883.

---

> ### Author Response · Authors · 2024-11-22
>
> **[Question 1 Graph matching ablation]**
>
> As suggested by the reviewer, we have analyzed the effect of graph matching on the performance of DDSBM by comparing the results of using four different graph matching algorithms. This experiment was conducted on the ZINC dataset, and the evaluation was performed after six IMF iterations (See Table 5 in Appendix).
>
> The performance differences across graph matching algorithms were not significant, as shown in the following table. We attribute this to DDSBM's ability to preserve the permutation of the original graph based on our reference process. This property enables the generation of pairs with reduced NLL by Markovian projection during IMF iterations. Due to this effect, the model converges in a similar manner regardless of the specific graph matching algorithm applied at each IMF iteration, resulting in minor differences as shown in the table.
>
> \begin{array}{l|ccc|ccc|ccc}
> \text{Algorithm} & \text{Val.} \uparrow & \text{Uniq.} \uparrow & \text{Nov.} \uparrow & \text{FCD} \downarrow & \text{NLL} \downarrow & \text{NSPDK} \downarrow & \text{LogP } W_1 \downarrow & \text{QED MAD} \downarrow & \text{SAScore MAD} \downarrow \newline
> \hline
> \text{(1) SM} & 94.6 & \mathbf{100.0} & 99.9 & 0.837 & 163.931 & 1.17\text{e-}3 & 0.159 & 0.122 & \mathbf{0.388} \newline
> \text{(2) MPM} & \mathbf{95.8} & \mathbf{100.0} & \mathbf{100.0} & 0.773 & \mathbf{155.378} & 6.95\text{e-}4 & 0.164 & \mathbf{0.110} & 0.401 \newline
> \text{(3) MPM + Randomness} & 95.3 & \mathbf{100.0} & 99.9 & 0.759 & 158.534 & \textbf{6.78e-4} & \mathbf{0.130} & 0.116 & 0.424 \newline
> \text{(4) Our setting} & 94.1 & \mathbf{100.0} & 99.9 & \mathbf{0.747} & 164.480 & 7.49\text{e-}4 & 0.139 & 0.124 & 0.392 \newline
> \end{array}
>
> Based on the observation that the performance of DDSBM is not strongly related to the algorithm of graph matching, one can infer that DDSBM can generate pairs that reduce NLL, minimizing the effect of graph matching. With this perspective, we also compared the results of **two scenarios**: **1)** one without using any graph matching throughout the whole training process, and **2)** another with  graph matching both on the initial data and during the IMF iterations. Note that the latter is the same experimental setup as in the current manuscript.
>
> Interestingly, similar to the ablation study of the graph matching algorithm, we found that the use of graph matching merely affected the result after a sufficient number of IMF iterations. However, we found that both the training loss and the NLL between the original and generated data values were initially higher in the absence of graph matching than in the presence of graph matching, and they gradually converged to similar levels as iterations progressed. This indicates that graph matching algorithms, while not essential for the eventual convergence of DDSBM, can accelerate convergence during the IMF iterations, providing an additional benefit beyond the Tanimoto coupling discussed in the manuscript.
>
> We sincerely appreciate the reviewer’s constructive feedback, which allowed us to identify the effect of graph matching on the performance of DDSBM. This investigation has provided us with a clearer understanding of DDSBM’s capabilities, and we are pleased to have been able to confirm and clarify its performance more thoroughly. We will include the contents of this discussion in appendix of our revised manuscript.

---

> > ### Comment · Reviewer_b65H · 2024-11-24
> >
> > Thank you for providing the new experiments, which have addressed most of my questions regarding the experiments. However, for the writing part, I would need a revised manuscript to better understand the changes being made.
> >
> > I have a few specific questions regarding the generation tasks you presented:
> > * How many epochs are required for the experiments?
> > * How many steps are used to generate the graphs? Since your method should require significantly fewer steps compared to traditional diffusion methods, it would be helpful to have clarity on this.

---

> > > ### Author Response · Authors · 2024-11-25
> > >
> > > Dear Reviewer b65H,
> > >
> > > Thank you for your thoughtful feedback and for acknowledging the additional experiments we conducted. We appreciate your interest in the writing improvements, and we have thoroughly revised the manuscript to incorporate all the changes. We outlined the major changes made in the revised manuscript from **General Response** and **Summary of Additional Experiments and Results**. We also tried to fix the "square brackets" issue. If the problem persists, feel free to inform about them.
> > >
> > > Regarding your specific questions:
> > > 1. **Number of Training Epochs**:
> > >
> > >     The number of epochs during training is shown in **Table 4** and **Table 9** in the Appendix of the revised manuscript. For convenience, we have reconstructed the table here. To ensure a fair comparison, we used the same total number of epochs for DBM and DDSBM on each task.
> > >
> > > \begin{array}{ll|cccc}
> > > \text{Task} & \text{Model} & \text{Epoch} & \text{SB Iterations} & \text{Total Epoch} & \text{Diffusion Steps} \newline
> > > \hline
> > > \text{ZINC250K} & \text{DBM} & 1800 & - & 1800 & 100 \newline
> > > \text{ZINC250K} & \text{DDSBM} & 300 & 6 & 1800 & 100 \newline
> > > \text{Polymer} & \text{DBM} & 1250 & - & 1250 & 100 \newline
> > > \text{Polymer} & \text{DDSBM} & 250 & 5 & 1250 & 100 \newline
> > > \text{QM9} & \text{DBM} & 200 & - & 200 & 100 \newline
> > > \text{QM9} & \text{DDSBM} & 100 & 2 & 200 & 100 \newline
> > > \text{Community-20} & \text{DBM} & 1000\text{k} & - & 1000\text{k} & 500 \newline
> > > \text{Community-20} & \text{DDSBM} & 200\text{k} & 5 & 1000\text{k} & 500 \newline
> > > \text{Planar} & \text{DBM} & 100\text{k} & - & 100\text{k} & 1000  \newline
> > > \text{Planar} & \text{DDSBM} & 10\text{k} & 10 & 100\text{k} & 1000 \newline
> > > \hline
> > > \end{array}
> > >
> > > We note that the total epoch is calculated by multiplying the number of epochs per iteration by the number of Schrödinger (SB) iterations.
> > >
> > > 2. **Number of Steps Used to Generate the Graphs**:
> > >
> > >     The number of diffusion steps used for each task is provided in the table above. We conducted additional experiments comparing the generation performance across different Numbers of Function Evaluations (NFEs) on the QM9 task. The results demonstrate that DDSBM can achieve satisfactory performance with a small number of diffusion steps below one hundred, while DiGress, a traditional diffusion method, used 500 diffusion steps for the same task. Please see Figure 22 in Appendix of the revised manuscript.
> > >
> > > Your constructive review helped us provide a thorough evaluation of DDSBM's performance.
> > >
> > > We look forward to your reply. Thank you again.

---

> > > > ### Comment · Reviewer_b65H · 2024-11-25
> > > >
> > > > Thank you for addressing my concerns and the revised version is significantly more readable. For future improvement, I would still suggest considering another round of revision.
> > > > But I raise my score to 6, as it makes a contribution as the first attempt at graph interpolation, even if the initial submission wasn’t entirely complete.

---

> > > > > ### Author Response · Authors · 2024-11-26
> > > > >
> > > > > Dear Reviewer b65H,
> > > > >
> > > > > We’re happy to hear that our rebuttal effectively addressed your concerns, and we sincerely appreciate your support for our work.
> > > > >
> > > > > If you have any additional questions or suggestions, please don’t hesitate to let us know.
> > > > >
> > > > > Best regards,
> > > > >
> > > > > The Authors

---

### Official Review · Reviewer_7pXr · 2024-11-03

**Soundness:** 3
**Presentation:** 2
**Contribution:** 2
**Rating:** 3
**Confidence:** 3

**Summary:**

This paper introduces discrete diffusion Schrödinger bridge model using continuous-time Markov chains. The proposed model was validated on molecular optimization problem.

**Strengths:**

Strong performance compared to diffusion bridge model.

**Weaknesses:**

Overall, this paper failed to distinguish its own contributions from existing work and to compared itself with relevant works. First of all, the proposed discrete diffusion Schrödinger bridge matching (DDSBM) highly relies on continuous-time Markov chains (CMTC). However, this paper failed to refer flow matching models [1, 2] which introduce CMTC for discrete data domains.

[1] Campbell, Andrew, et al. "Generative Flows on Discrete State-Spaces: Enabling Multimodal Flows with Applications to Protein Co-Design." Forty-first International Conference on Machine Learning.

[2] Gat, Itai, et al. "Discrete flow matching." arXiv preprint arXiv:2407.15595 (2024).

And, this is not the first paper that solving Schrödinger bridge (SB) problem in discrete state spaces. There are works for discrete SB [3, 4]. However, the relevance and difference between the proposed method were not discussed.

[3] Harchaoui, Zaid, Lang Liu, and Soumik Pal. "Asymptotics of discrete Schrödinger bridges via chaos decomposition." Bernoulli 30.3 (2024): 1945-1970.

[4] Chow, Shui-Nee, et al. "A discrete Schrodinger bridge problem via optimal transport on graphs." calculus of variations 20.33 (2021): 34.

**Questions:**

1. What is the relevance and difference between the proposed method and existing discrete SB models in Weakness?

2. How does the performance of molecular optimization compare with discrete flow matching models?

---

> ### Author Response · Authors · 2024-11-13
>
> Thank you for your thoughtful review and suggestions on related works. Here, we would like to briefly explain the primary task of this work. Our focus lies in transformations between graphs, specifically including molecular graphs. Rather than addressing the transportation between distributions defined over topological spaces of graphs, we tackle the problem of transporting the source graph to the target graph through structural edits (graph transformations).
>
> **[Regarding Weakness 1 & Question 2]**
>
> As you suggested, comparing flow matching models (FM) and diffusion bridge models (DBM) would indeed be an interesting study. In essence, FM and DBM approaches are similar in that they share the objective of training a transportation between two distributions.
>
> The proposing method defines a reference process as a continuous-time Markov chains (CTMC) in a discrete metric state space, then solves the associated “Schrödinger bridge problem” (SBP). The SBP can be interpreted as an entropy-regularized optimal transport problem (EOT), making it closely related to optimal transport-flow matching (OT-FM). However, it is important to emphasize that, while both DBM and SBP aim to transport between distributions, SBP aims to find the optimal joint distribution (paired data distribution) in contrast to DBM, whose objective is to learn transportation map based on a given joint distribution. This distinction also parallels the difference between FM and OT-FM.
>
> Due to these differences, it would be challenging to conduct a controlled comparison between FM and our method, as several variables in the methodology—such as deep learning model architecture, SDE vs. ODE formulations, sampling algorithms, and training of joint distributions—would be difficult to control. Additionally, the two FM papers [1, 2] you suggested focus on transporting node feature distributions within a fixed graph structure rather than addressing the transformation of the graph structure itself. Reflecting image and text data inherently form sequences or grid structures, the suggested references do not alter these underlying structures. For these reasons, we believe the suggested references may not align closely with the main focus of our manuscript.
>
> To the best of our knowledge, the only work on OT-FM for graphs is [5], which is currently under peer review at ICLR 2025, and code for this work has not yet been released.
>
> **[Regarding Weakness 2 & Question 1]**
>
> As the reviewer mentioned, our proposed DDSBM is not the first to address the Schrödinger bridge problem in discrete states. Reference [3] addresses the SBP between two distributions based on empirical datasets realized from each distribution, and shows that as the number of observations grows, the solution converges to the SB between the true distributions. Since DDSBM is also framed as an SBP between empirical distributions, this work could indeed be a useful reference. However, our work focuses on the transport problem between graph structures (”*discrete*-states”) rather than the fact that the problem is addressed with sampled (”*discretized*”) distributions. Also, we propose a solution method for the SBP in discrete setting, called DDSBM, while the reference [3] does not. Due to these aspects, the reference [3] differs from the central objective of DDSBM.
>
> Reference [4] defines the Schrödinger bridge problem *on* graphs and provides a numerical solution, establishing SBP on a graph space where nodes have defined probabilities and edges have defined weights. Within the context of this reference, our work could be considered an SBP for graph transformations: each node represents a graph dataset, such as a molecular graph, and edge weights are defined through a reference stochastic process (CTMC). Unlike the reference [4], which cannot generate new data, our work aims to produce transformed data for novel cases through generative modeling, addressing the transportation of data distributions. Nevertheless, the reference [4] indeed provides a valuable perspective on defining and solving the Schrödinger bridge problem in discrete spaces, and we will reference it in our manuscript.

---

> ### Author Response · Authors · 2024-11-13
>
> **[Additional Experiment Plan]**
>
> Our proposed method is not limited to mappings between molecular graphs alone with its theoretical support. To further substantiate this claim, we plan to conduct additional experiments applying our approach to a more diverse set of general graphs beyond molecular structures.
>
> Specifically, we plan to extend our experiments on additional datasets proposed in SPECTRE [6] for graph generation task. The datasets include synthetic graph dataset (**Community-small**) and widely-used molecular dataset (**QM9**). Additionally, we will conduct graph generation on larger synthetic graph dataset (**Planar**), to further validate our approach.
>
> We believe this review process will constructively enhance the quality of the manuscript. If there are any additional aspects you would like us to discuss, please feel free to let us know.
>
> **[References]**
>
> [1] Campbell, Andrew, et al. "Generative Flows on Discrete State-Spaces: Enabling Multimodal Flows with Applications to Protein Co-Design." Forty-first International Conference on Machine Learning.
>
> [2] Gat, Itai, et al. "Discrete flow matching." arXiv preprint arXiv:2407.15595 (2024).
>
> [3] Harchaoui, Zaid, Lang Liu, and Soumik Pal. "Asymptotics of discrete Schrödinger bridges via chaos decomposition." Bernoulli 30.3 (2024): 1945-1970.
>
> [4] Chow, Shui-Nee, et al. "A discrete Schrodinger bridge problem via optimal transport on graphs." calculus of variations 20.33 (2021): 34.
>
> [5] https://openreview.net/forum?id=rMyfMS5nMt
>
> [6] Martinkus, Karolis, et al. "Spectre: Spectral conditioning helps to overcome the expressivity limits of one-shot graph generators." *International Conference on Machine Learning*. PMLR, 2022.

---

> ### Author Response · Authors · 2024-11-22
>
> Thank you for thoughtful review and recommendations regarding related works.
>
> We have revised the Methods section to address reference [1], as suggested:
> "The SB problem in discrete setting has recently been studied by Chow et al. (2021) [1], which formulates an optimal transport problem on graphs as the SB problem. Graph transformation can be seen as a analogous problem in terms of the data states are discrete, but the fact that each data is also an individual graph requires distinct formulation for transition between one another."
>
> In response to reviewers’ suggestions, we have performed additional experiments to further validate our approach and enhance the robustness of our results.
> Specifically, we included error bars calculated over multiple training runs and introduced three new evaluation metrics: NSPDK, Uniqueness, and Novelty.
> These additions offer a more comprehensive and reliable assessment of our method.
>
> Furthermore, we expanded the scope of our experiments to include unconditional generations in several widely-used datasets: QM9 (a molecular dataset), as well as Comm20 and Planar (general graph datasets).
> These additional evaluations provide a more comprehensive validation of our approach.
> The detailed results of these additional experiments have been provided in our official comments to other reviewers.
>
> We are also actively revising the manuscript to improve its clarity and readability.
> We welcome further discussion or suggestions regarding the additional results and are committed to incorporating feedback to strengthen the manuscript.
>
> **[Reference]**
>
> [1] Chow, Shui-Nee, et al. "A discrete Schrodinger bridge problem via optimal transport on graphs." calculus of variations 20.33 (2021): 34.

---

> ### Author Response · Authors · 2024-11-25
> **Gentle reminder**
>
> Dear Reviewer 7pXr,
>
> We have thoroughly revised the manuscript to address all the suggested changes.
>
> We have summarized the major revisions made in the revised manuscript, from **”General Response”** and “**Summary of Additional Experiments and Results”**.
>
> This is a gentle reminder, as we approach the final stages of the discussion period.
>
> We sincerely hope for a valuable and constructive conclusion to the discussions regarding our paper.

---

> ### Comment · Area_Chair_N4YZ · 2024-11-25
>
> Could please acknowledge and respond to the rebuttal.

---

> > ### Author Response · Authors · 2024-11-30
> >
> > Dear Reviewer 7pXr,
> >
> > We wanted to kindly follow up on our rebuttal to ensure that you’ve had the opportunity to review our responses to your valuable feedback. If you find our clarifications and the enhancements we’ve made satisfactory, we would be grateful if you could consider re-evaluating our paper. If you have any additional questions or require further discussion, we would be more than happy to discuss about it.
> >
> > Thank you again for your time and thoughtful consideration.
> >
> > Best regards,
> >
> > Authors

---

> > > ### Comment · Reviewer_7pXr · 2024-12-01
> > >
> > > It says 'transportation between distributions defined over topological spaces of graphs, we tackle the problem of transporting the source graph to the target graph through structural edits', but I don't see any difference in terms of learning transportation from the source distribution to the target distribution. Also, no comparison with the latest discrete flow matching was made, and additional experiments did not show a clear advantage in comparison with the discrete diffusion model and DBM. Thus, I will keep the current score.

---

> > > > ### Author Response · Authors · 2024-12-02
> > > > **Reply to Reviewer 7pXr (1/2)**
> > > >
> > > > First of all, we start with clarification of the distinction between our work and a prior research about discrete Schrödinger Bridge Problem (SBP), particularly [4], which the reviewer mentioned. Specifically, the key difference lies in the main purpose of the works. While [4] primarily addresses the discrete SB problem by demonstrating the existence of unique solutions and mathematical characterizing them, our work extends this line of research to high-dimensional applications through generative modeling with training neural network. This clear distinction was also treated in the previous works about continuous SB problems (e.g., DSB, DSBM, and IDBM), where one of the novel contributions was approaching to the established solution methods for the SBP with a generative modeling perspective to enable their application in high-dimensional datasets.
> > > >
> > > > We provide responses to your comment.
> > > > 1. **About “I don't see any difference in terms of learning transportation from the source distribution to the target distribution.”**
> > > >
> > > >     As it mentions, our work aims to address the optimal transportation problem between source and target distributions defined in a discrete space. However, while we formulate the graph transformation problem as a SBP between distributions, with the goal of generating new corresponding graphs given an input graph, the prior work [4] primarily explore the discrete SB problem itself, investigating conditions for unique solutions and various theoretical properties, without providing practical methods to solve it. The prior works formulate the discrete SB problem as solving a couple of ordinary differential equations (ODEs) given boundary values, i.e. a boundary value problem (BVP), rather than learning transportation. However, this approach raises challenges such as solution convergence and computational feasibility, especially for high-dimensional data. In high-dimensional spaces, exploring all possible graph configurations, assigning costs to every edge, and solving the BVP can become computationally intractable. In contrast, our proposed DDSBM framework overcomes these limitations by utilizing generative modeling techniques. Instead of exhaustive exploration and assignment to solve a BVP, DDSBM focuses solely on the graph data discovered during the learning and generation process of the diffusion bridge. This makes DDSBM significantly more efficient. Additionally, as a generative modeling framework, DDSBM is capable of generating corresponding graphs for previously unseen graph data during inference. In comparison, prior works are restricted to performing transportation only between the two predefined distributions, making it impossible to discover mappings for novel graph data.
> > > >
> > > >     We discuss the difference with a more specific example from our manuscript, the experiment with the ZINC250K dataset. Our DDSBM framework leverages approximately 60,000 graphs from the combined source and target distributions. During the Iterative Markovian Fitting (IMF) process, the inherent stochasticity of the diffusion bridge allows DDSBM to explore new graph data dynamically. Importantly, due to the Markovian nature of the process, any graph generated in this intermediate step can be discarded immediately after use without affecting the overall process. In contrast, prior approaches require exhaustive exploration of all possible graph configurations. For example, [GDB-17] reported that the number of chemically stable molecules with up to 17 heavy atoms is approximately 166 billion. As the maximum number of nodes increases, the number of possible graphs grows exponentially. Solving the SB problem over such an immense graph space using prior methods is computationally infeasible, and we explicitly state this limitation in our introduction: **“To bridge the gap, we propose a novel framework called Discrete Diffusion Schrödinger Bridge Matching (DDSBM), utilizing the continuous-time Markov chains (CTMCs) to solve the SB problem in a high-dimensional discrete setting.”** Additionally, existing methods fix the terminal distribution entirely to the given empirical distribution. This constraint means they cannot generate corresponding target graphs for unseen source graphs. In contrast, DDSBM is designed to handle such scenarios, offering the ability to generate new graph data for novel source graph inputs. This fundamental difference arises because prior works focus on defining the SB problem and studying its mathematical properties. Our work, on the other hand, formulates graph transformation as an SB problem and provides a practical framework for solving the SBP in high-dimensional discrete spaces. Furthermore, our work introduces theoretical advancements by adapting the IMF used in continuous SB problems to the discrete setting, providing a solid theoretical foundation for its application.

---

> > > > ### Author Response · Authors · 2024-12-02
> > > > **Reply to Reviewer 7pXr (2/2)**
> > > >
> > > > 2. **About “no comparison with the latest discrete flow matching was made,”**
> > > >
> > > >     We have carefully reviewed your suggested references and would like to clarify the distinction between these works and our approach. While the referenced works do indeed address discrete state data and provide relevant formulations, their primary focus is on data types such as language and images rather than graphs. These data types, while discrete, inherently lack the permutation invariance property that is fundamental to graphs due to their specific node ordering. Furthermore, in terms of graphs, these approaches consider settings where node connectivity (edge) is fixed and only node attributes change. This is fundamentally different from the graph symmetry and structural dynamics addressed in our work, as the mathematical objects governing the transition kernels are different. Our work, as emphasized in the title, focuses on solving the SB problem specifically for graph transformations. Graphs inherently possess unique symmetries, and designing a framework to work effectively in this domain requires dedicated methods tailored to these properties. Recent work in discrete flow matching does not take graph-specific properties or symmetries into account, so we have not conducted direct comparisons with these methods, as in our previous response. Additionally, it is important to emphasize a key conceptual difference: our work focuses on identifying the **optimal** transportation for given boundary condition by solving SBP. This distinction highlights a significant divergence in both purpose and methodology between our approach and the referenced works. While the referenced works [1,2] are designed to find transportation plan based on a predefined coupling (either unconditional or conditional), our framework tackles the challenge of discovering the optimal transportation plan itself.
> > > >
> > > > 3. **About “Additional experiments did not show a clear advantage in comparison with the discrete diffusion model and DBM.”**
> > > >
> > > >     The primary purpose of the additional experiments was to demonstrate the capability of DDSBM across general graph generation tasks. However, please note that improving unconditional generation performance was not the main objective of DDSBM. Again, our main goal is transforming an initial graph distribution to terminal graph distribution via discrete SBP. Thus, to evaluate about the advantage of DDSBM, we believe the reviewer should focus on our main tasks, as well as evaluated in the main manuscript on the ZINC250K and Polymer datasets. For these tasks, DDSBM significantly outperformed DBM across all metrics used to evaluate both marginal and joint distributions, showing clear advantage in graph transformation (differ from graph generation) task. Regarding the additional experiments provided during the rebuttal period, DDSBM achieved performance that was comparable to or better than existing methods on the community-20 and QM9 datasets, with the exception of the planar dataset. While the benchmark metrics show only minor differences between DBM and DDSBM, we note that DDSBM exhibited notably better robustness in terms of the number of function evaluations (NFEs), as highlighted in Appendix G.5. Specifically, even with reduced NFEs, DDSBM demonstrated less performance degradation compared to DBM. This indicates that DDSBM allows for faster generation with minimal performance loss by finding an optimal joint distribution. In summary, DDSBM not only performs tasks that existing discrete diffusion models cannot but also achieves superior results compared to DBM in its primary transportation tasks. We believe these points highlight the unique advantages and contributions of DDSBM.
> > > >
> > > > **[References]**
> > > >
> > > > [1] Campbell, Andrew, et al. "Generative Flows on Discrete State-Spaces: Enabling Multimodal Flows with Applications to Protein Co-Design." Forty-first International Conference on Machine Learning
> > > >
> > > > [2] Gat, Itai, et al. "Discrete flow matching." arXiv preprint arXiv:2407.15595 (2024)
> > > >
> > > > [4] Chow, Shui-Nee, et al. "A discrete Schrodinger bridge problem via optimal transport on graphs." calculus of variations 20.33 (2021): 34.
> > > >
> > > > [DSB] De Bortoli, V., Thornton, J., Heng, J., & Doucet, A. (2021). Diffusion schrödinger bridge with applications to score-based generative modeling. *Advances in Neural Information Processing Systems*, *34*, 17695-17709.
> > > >
> > > > [DSBM] Shi, Y., De Bortoli, V., Campbell, A., & Doucet, A. (2024). Diffusion Schrödinger bridge matching. *Advances in Neural Information Processing Systems*, *36*.
> > > >
> > > > [IDBM] Peluchetti, S. (2023). Diffusion bridge mixture transports, Schrödinger bridge problems and generative modeling. *Journal of Machine Learning Research*, *24*(374), 1-51.
> > > >
> > > > [GDB-17] Ruddigkeit, Lars, et al. "Enumeration of 166 billion organic small molecules in the chemical universe database GDB-17." *Journal of chemical information and modeling* 52.11 (2012): 2864-2875.

---

> > > > > ### Comment · Reviewer_7pXr · 2024-12-02
> > > > >
> > > > > Thank you for the response.
> > > > >
> > > > > The manuscript claims to be based on CTMC. Thus, it seems necessary to compare it with CTMC-based pre-defined transportation models. Comparing the proposed approach with studies that pre-defined transportation, as in the referenced paper, could help determine whether the learned transportation is indeed optimal.
> > > > >
> > > > > If the paper's focus is not on the effectiveness of the model in unconditional generation but rather on the transportation between two discrete distributions for graphs, then it would be more appropriate to compare it with related works like [1] (Igashov et al., "RetroBridge: Modeling Retrosynthesis with Markov Bridges").
> > > > >
> > > > > Overall, the manuscript seems to avoid direct comparisons with closely related works, which makes it difficult to assess the effectiveness of the proposed method.

---

> > > > > > ### Author Response · Authors · 2024-12-03
> > > > > > **Reply to Reviewer 7pXr (1/2) : About CTMC-based pre-defined transportation model**
> > > > > >
> > > > > > Thank you for your comments and the opportunity to clarify and strengthen our manuscript.
> > > > > >
> > > > > > The primary contribution of our work lies in learning the (entropy-regularized) optimal transportation plan between two discrete graph distributions, particularly in cases where **pre-defined joint distribution is unavailable**. As the reviewer noted, we compared DDSBM with DBM to compare the optimality and sub-optimality of the transportation plan learned, where the DBM is **CTMC-based pre-defined (randomly initialized) transportation model**.
> > > > > >
> > > > > > The notion of **optimality** in our study is defined relative to the reference process. This is quantitatively evaluated using the Negative Log-Likelihood (NLL) metric, which is presented in Tables 1 and 2. Furthermore, we describe in Appendix D how NLL can be interpreted in the context of graph edit distance, providing an additional perspective on the optimality.
> > > > > >
> > > > > > In contrast, "CTMC-based pre-defined transportation models," as mentioned by the reviewer, address tasks where a **pre-defined** **joint coupling dataset is available**. This is fundamentally different from the problem our manuscript addresses. Notably, such tasks cannot be effectively modeled using CTMC-based methods, as detailed in Theorem B.1 of our paper. To achieve joint pair training in those settings, **reciprocal modeling** is required (as shown in works like [1]), which is out of the scope.
> > > > > >
> > > > > > [1] Zhou, L., Lou, A., Khanna, S., & Ermon, S. (2023). Denoising diffusion bridge models. *arXiv preprint arXiv:2309.16948*.

---

> > > > > > ### Author Response · Authors · 2024-12-03
> > > > > > **Reply to Reviewer 7pXr (2/2) : About Requested Experiments**
> > > > > >
> > > > > > **[Discrimination from RetroBridge]**
> > > > > >
> > > > > > As mentioned in the related works section of the main manuscript, **RetroBridge** is a framework designed to solve the retrosynthesis planning task where a **well-defined data pair** exists. This task has the following two characteristics:
> > > > > >
> > > > > > 1. There is a clear data pair of reactant and product from chemical reactions.
> > > > > > 2. Node-level alignment information between reactant and product is provided.
> > > > > >
> > > > > > RetroBridge effectively leverages these task characteristics to define and solve the transportation problem.
> > > > > >
> > > > > > In contrast, our study addresses the optimal transport problem between two data distributions **where well-defined data pairs do not exist**, and therefore, node-level alignment information is also absent. Due to this difference, the **evaluation metrics** and **research goals(objective)** of the two approaches are fundamentally different.
> > > > > >
> > > > > > Nevertheless, as Reviewer 7pXr suggested, we can evaluate how well DDSBM identifies optimal joint pairs compared to other works that address transportation between distributions without focusing on its optimality. For this purpose, the baseline model we presented is the DBM mentioned in the manuscript, which is conceptually very similar to RetroBridge.
> > > > > >
> > > > > > In summary, because the objective of the RetroBridge model is very different from our work, we have selected and presented DBM as a control model to evaluate the optimality of DDSBM in this manuscript.
> > > > > >
> > > > > > ---
> > > > > >
> > > > > > **[Additional Experiment Results]**
> > > > > >
> > > > > > In response to your request, we conducted additional experiments to evaluate RetroBridge’s performance on the **ZINC task with random initial coupling,** presented in our main manuscript.
> > > > > >
> > > > > > Using the official RetroBridge implementation (available at [RetroBridge GitHub repository](https://github.com/igashov/RetroBridge/tree/main)), we trained and evaluated the model on our dataset. The results show that **DDSBM outperforms RetroBridge across all metrics except for novelty**, demonstrating the superior performance of DDSBM on **graph transformation tasks**. Also, it is worth noting that RetroBridge exhibited a very low validity score of 29.6%, which highlights its challenges in this task.
> > > > > >
> > > > > > \begin{array}{l|ccc|ccc|ccc}
> > > > > > \text{Model} & \text{Val.} \uparrow & \text{Uniq.} \uparrow & \text{Nov.} \uparrow & \text{FCD} \downarrow & \text{NLL} \downarrow & \text{NSPDK} \downarrow & \text{LogP } W_1 \downarrow & \text{QED MAD} \downarrow & \text{SAScore MAD} \downarrow \newline
> > > > > > \hline
> > > > > > \text{RetroBridge} & 29.6 & \mathbf{100.0} & \mathbf{100.0} & 15.178 & 354.290 & 3.95\text{e-}2 & 0.737 & 0.131 & 2.173 \newline
> > > > > > \text{DBM} & 87.6 & \mathbf{100.0} & \mathbf{100.0} & 1.046 & 288.572 & 8.04\text{e-}4 & 0.150 & 0.141 & 0.608 \newline
> > > > > > \text{DDSBM} & \mathbf{94.8} & \mathbf{100.0} & 99.9 & \mathbf{0.833} & \mathbf{160.461} & \textbf{7.30e-4} & \mathbf{0.139} & \mathbf{0.120} & \mathbf{0.402} \newline
> > > > > > \end{array}
> > > > > >
> > > > > > The poor performance of RetroBridge across multiple evaluation metrics can likely be attributed to the fundamental differences in the objectives of our work and RetroBridge as discussed in the previous section.
> > > > > >
> > > > > > We hope that these response adequately address the reviewers' concerns. If you find our response satisfactory, we hope you could re-evaluate our paper.

---

### Official Review · Reviewer_AZ9a · 2024-11-03

**Soundness:** 3
**Presentation:** 3
**Contribution:** 3
**Rating:** 8
**Confidence:** 2

**Summary:**

This work presents Discrete Diffusion Schrödinger Bridge Matching (DDSBM), a framework that adapts continuous-time Markov chains (CTMCs) to address the Schrödinger Bridge (SB) problem within high-dimensional discrete spaces. Using the Iterative Markovian Fitting (IMF) technique, DDSBM enables optimal graph modifications by minimizing structural changes, which is particularly valuable for molecular optimization tasks in drug and material discovery. In this context, the framework aligns with graph edit distance (GED), allowing efficient property-driven modifications while preserving the molecule's structure. Experiments show DDSBM achieves minimal structural shifts and successfully maintains desirable molecular properties, outperforming traditional graph translation methods.

**Strengths:**

- The Schrödinger bridge problem is well studied in the continuous-state diffusion literature, but has, to the best of my knowledge, not yet been applied to the discrete setting. The paper nicely bridges this gap, hence proposing a valuable contribution to the discrete diffusion literature.
- The paper looks sound and technically strong.
- Empirical results seem to indicate that this approach outperforms previous baselines by a large margin.

**Weaknesses:**

- The choice of the FCD metric the evaluate the graph structure distributions is surprising, since it is typically used to assess distribution learning from a chemical perspective. Why not use NSPDK for example ?
- I’d like to see some error bars on your results.
- The method description is quite technical, and might be hard to understand for the average graph machine learning practioner. It could be worth it to include a short tutorial on the Schrödinder Bridge problem to ease the reader understanding.

**Questions:**

- What is the Hungarian algorithm ? You mention it in section 3.3. I'd appreciate if you could drop a reference or give some details in appendix.
- The baselines outperform your approach in terms of Validity. Even though the results on FCD and NLL indicate that your method is stronger, I'd appreciate if you could comment on this metric.

---

> ### Author Response · Authors · 2024-11-13
>
> We thank you for your thoughtful feedback and valuable suggestions, which have significantly contributed to the clarity and rigor of our work. Below, we outline our planned revisions and responses to each point raised.
>
> W1: We will incorporate the NSPDK metric as suggested.
>
> W2: We agree that presenting error bars would enhance the robustness of our results. To address this, we will conduct additional experiments and include error bars across different training runs.
>
> W3: We acknowledge that the current version may sound overly technical to the typical graph machine learning practitioner. In response, we will revise this section to improve clarity and accessibility, reflecting your suggestion. Once the revisions are complete, we will provide an official comment.
>
> Q1: We apologize for omitting a reference for the Hungarian algorithm. The Hungarian algorithm, also known as the Kuhn-Munkres algorithm [1], is a combinatorial optimization technique for solving the assignment problem. We used the Hungarian algorithm implemented in the Pygmtools package [2]. In the revised manuscript, we will add the references.
>
> Q2: We acknowledge that the baseline models show higher validity compared to our method. However, we observed that these models also demonstrate relatively low uniqueness, which suggests that their validity scores may be overestimated due to potential redundancy in generated structures. In this regard, we will conduct a thorough re-evaluation to verify these findings, and we will add a detailed discussion on this point in the revised manuscript.
>
> Once these revisions are complete, we will provide an official comment. Thank you once again for your constructive comments.
>
> If there are any additional requests, please feel free to let us know.
>
> References
>
> [1] Kuhn, H. W. (1955). The Hungarian method for the assignment problem. *Naval research logistics quarterly*, *2*(1‐2), 83-97.
>
> [2] Wang, R., Guo, Z., Pan, W., Ma, J., Zhang, Y., Yang, N., ... & Yan, J. (2024). Pygmtools: A python graph matching toolkit. *Journal of Machine Learning Research*, *25*, 1-7.

---

> ### Author Response · Authors · 2024-11-22
>
> We would like to thank the reviewer for valuable feedback.
>
> **[Weakness 1, 2]**
>
> We conducted additional experiments and included error bars across different training runs.
>
> Table 1 (ZINC) and Table 2 (Polymer) will be updated based on three independent training runs. We confirmed that the trends remain consistent across multiple experiments, further demonstrating the robustness of DDSBM. See the revised ZINC table and revised Polymer table.
>
> Additionally, we included NSPDK as a metric. Similar to other metrics, DDSBM demonstrated superior performance. We are grateful to the reviewer for guiding us to further validate our approach.
>
> **[Weakness 3]**
>
> We are currently reorganizing Sections 2, 3, and 4 to better convey our insights.
>
> In addition, based on the reviewer's suggestion, we plan to include an introduction to the Schrödinger Bridge problem and CTMC in the appendix to help readers better understand the context.
>
> We are also preparing a graphical item—chain visualizations—to provide an intuitive explanation of how DDSBM operates on graphs.
>
> We aim to ensure that this addition makes the concepts more accessible and comprehensible to potential readers.
>
> **[Question 2]**
>
> To clearly highlight the performance differences in the validity metric between the baseline and DDSBM, we conducted evaluations on novelty and uniqueness. While the baselines achieved almost 100% validity, their novelty and uniqueness scores were significantly lower compared to DDSBM. Thanks to the reviewer’s suggestions, we were able to clarify the distinctions between the baseline and our method and further reaffirm the strengths of DDSBM.
>
> Thank you once again for your valuable feedback, which has helped us clarify the key arguments of our paper.
>
> If there are any question, please feel free to let us know.

---

> > ### Author Response · Authors · 2024-11-22
> > **Revised Table (ZINC and Polymer)**
> >
> > ## 1. Revised ZINC Table
> > ---
> >
> > \begin{array}{lc|ccc|ccc|ccc}
> > \text{Model} & \text{Type} & \text{Val.} \uparrow & \text{Uniq.} \uparrow & \text{Nov.} \uparrow & \text{FCD} \downarrow & \text{NLL} \downarrow & \text{NSPDK} \downarrow & \text{LogP } W_1 \downarrow & \text{QED MAD} \downarrow & \text{SAScore MAD} \downarrow \newline
> > \hline
> > \text{AtomG2G} & \text{Latent} & 99.9 (\pm 0.1) & 64.4 (\pm 2.6) & 99.3 (\pm 0.1)  & 5.019 (\pm 0.572) & 355.025 (\pm 0.484) & 9.70\text{e-}3 (\pm 2.01\text{e-}3)  & 0.162 (\pm 0.034) & 0.143 (\pm 0.001) & 0.697 (\pm 0.019) \newline
> > \text{HierG2G} & \text{Latent} & \textbf{100.0} (\pm 0.0) & 73.7 (\pm 1.3) & 99.5 (\pm 0.1)  & 5.742 (\pm 0.378) & 344.458 (\pm 4.454) & 2.06\text{e-}2(\pm 4.75\text{e-}3)  & \textbf{0.113} (\pm 0.045) & 0.146 (\pm 0.005) & 0.687 (\pm 0.032) \newline
> > \hline
> > \text{DBM} & \text{Bridge} & 87.6 (\pm 1.2) & \textbf{100.0} (\pm 0.0) & \textbf{100.0} (\pm 0.0) & 1.046 (\pm 0.043) & 288.572 (\pm 3.327) & 8.04\text{e-}4 (\pm 2.23\text{e-}4) & 0.150 (\pm 0.012) & 0.141 (\pm 0.002) & 0.608 (\pm 0.013) \newline
> > \text{DDSBM} & \text{Schrödinger Bridge} & 94.8 (\pm1.9) & \textbf{100.0} (\pm 0.0) & 99.9 (\pm 0.0) & \textbf{0.833} (\pm 0.082) & \textbf{160.461} (\pm 10.409) & \textbf{7.30e-4} (\pm 9.29\text{e-}5) & 0.139 (\pm 0.003) & \textbf{0.120} (\pm 0.009) & \textbf{0.402} (\pm 0.028)  \newline
> > \end{array}
> >
> > ## 2. Revised Polymer Table
> > ---
> > \begin{array}{lc|ccc|ccc|c}
> > \text{Model} & \text{Type} & \text{Val.} \uparrow & \text{Uniq.} \uparrow & \text{Nov.} \uparrow & \text{FCD} \downarrow & \text{NLL} \downarrow & \text{NSPDK} \downarrow & \text{GAP } W_1 \downarrow \newline
> > \hline
> > \text{DBM} & \text{Bridge} & 43.4 (\pm 0.015) & \textbf{99.8} (\pm 0.002) & \textbf{97.4} (\pm 0.003)  & 2.230 (\pm 0.175) & 580.415 (\pm 11.279) & 5.82\text{e-}3 (\pm 3.62\text{e-}4) & 0.249 (\pm 0.019) \newline
> > \text{DDSBM} & \text{Schrödinger Bridge} & \textbf{97.4} (\pm 0.004) & 94.5 (\pm 0.011) & 71.3 (\pm 0.038) & \textbf{1.074} (\pm 0.038) & \textbf{212.047} (\pm 18.444) & \textbf{4.18e-3} (\pm 1.55\text{e-}4) & \textbf{0.127} (\pm 0.013) \newline
> > \end{array}

---

> ### Author Response · Authors · 2024-11-25
>
> Dear Reviewer AZ9a,
>
> We have added a tutorial on the Schrödinger Bridge problem and Iterative Markovian Fitting in **Appendix A**.
>
> This tutorial includes detailed explanations of the theoretical background and links each part directly to the relevant sections of the manuscript, making it more accessible to readers.
>
> In addition, we have provided insights showing that SB in discrete spaces can be intuitively understood as finding a coupling that best preserves the initial states, which aligns with the core motivation behind the development of DDSBM.
>
> We hope that this tutorial will help general readers to understand the Schrödinger Bridge problem and our work better.
>
> We look forward to your feedback and sincerely thank you once again.

---

> > ### Comment · Reviewer_AZ9a · 2024-11-25
> > **Answer to Rebuttal**
> >
> > Thanks for writting a detailed rebuttal and for trying to make your work accessible to a broader audience. I appreciated the effort to write such a detailed tutorial in appendix.
> >
> > Your approach seems to yield consistent results and your work seems technically strong. Even though I have quite a limited experience with the SB problem, I think your work deserves acceptance. I have increased my score accordingly.

---

> > > ### Author Response · Authors · 2024-11-26
> > >
> > > Dear Reviewer AZ9a,
> > >
> > > We’re glad to hear that our rebuttal effectively addressed your concerns, and we sincerely appreciate your support for our work.
> > >
> > > Feel free to give us any further questions or suggestions, to improve our paper.
> > >
> > > Best regards,
> > >
> > > The Authors

---

> ### Comment · Area_Chair_N4YZ · 2024-11-25
>
> Could please acknowledge and respond to the rebuttal.

---

### Author Response · Authors · 2024-11-25
**General Response**

First of all, we thank all reviewers for their thoughtful and constructive reviews.

We are pleased to share the revised version of our manuscript, which has been updated based on the valuable feedback provided. The changes are highlighted in **blue**. We combined the appendix with to the main manuscript. Below, we briefly summarize the key changes:



**[About Readability]**

1. **Revised the flow from Section 2 to Section 4:**

    We reconstructed our manuscript to clearly distinguish previous works and our contributions. The manuscript now first explains the related works (**Section 2**) and theoretical backgrounds about Schrödinger Bridge (SB) problem in the continuous space (**Section 3.1**). Following that, we outline about how one can extend SBP to discrete state space (**Section 3.2**). Building on these, we finally present our main contributions, which are proving the convergence of IMF iteration in discrete setting (**Section 3.3**), and its practical application methods in discrete space (**Section 4**).

2. **Added a tutorial on the Schrödinger Bridge problem and Iterative Markovian Fitting (IMF):**

    For general readers, we added the tutorial in **Appendix A** with explanations of the theoretical backgrounds and linked each part of the tutorial directly to the relevant sections in the manuscript. We also included some insights that SB in discrete space can be intuitively thought of as finding coupling that best preserves the initial states, which aligns with the main motivation behind developing DDSBM.

3. **Clarified our contribution:**

    Our major contribution lies in introducing DDSBM to solve SBP in discrete state spaces and applying it to the graph transformation problem. However, the previous manuscript lack of sufficient explanation. To address this, we have additional explanation for our contribution in the beginning of **Section 3.3** and **Section 4**. We also explicitly highlighted that our key contribution lies in introducing a diffusion model-based generative framework capable of tackling high-dimensional graph transformation tasks, which has not been addressed in prior works. (**Section 3.3, line 193-197**)

We hope that these revisions adequately address the reviewers' concerns and enhance the quality and clarity of our paper. We are grateful for the time and effort the reviewers have invested in providing their insightful feedback.

---

### Author Response · Authors · 2024-11-25
**Summary of Additional Experiments and Results**

For the experiments, we would like to express our sincere gratitude to the reviewers for their valuable feedback and constructive suggestions, which have significantly improved our work on DDSBM. We have carefully addressed all the comments and updated the manuscript accordingly. Below, we outline the major changes made in the revised version:

1. **Error Bars on Experiments:**

    We have reran our main experiments on the ZINC and Polymer datasets three times to calculate the average performance and standard deviations. This enhancement provides statistically more reliable results. Please see **Table 1**, **Table 2**, **Appendix Table 5**, and **Appendix Table 6** for the updated results.

2. **Additional Metrics in Experiments:**

    We have included the NSPDK (Neighborhood Subgraph Pairwise Distance Kernel) metric in all experiments, as suggested. Previously, we described FCD (Fréchet ChemNet Distance) as a graph structural metric. With the addition of NSPDK, we have reclassified FCD as a chemical property metric for a clarity. We appreciate the suggestion of a more appropriate metric.
3. **Ablation Studies:**

    We have added extensive ablation study results and analyses in **Appendix F.2**. This includes an ablation study on different types of graph matching algorithms. From this experiment, we empirically demonstrated that graph matching can accelerate the convergence of IMF iterations in practice. Additionally, we moved influence of initial data pairing methods on IMF iteration convergence, which was previously discussed in **Section 5.4**, as an another type of ablation for practical acceleration of DDSBM training.

4. **Unconditional Generation Experiments:**

    We conducted experiments on the QM9 dataset, commonly used as a benchmark for molecular generative models, and the synthetic Community-20 and Planar datasets frequently used for general graph benchmarks. DDSBM achieved the best or second-best performance in most metrics on both QM9 and Community-20. Furthermore, we consistently showed that DDSBM maintains the overall graph structure effectively, as evidenced by the low NLL (Negative Log-Likelihood) for both initial and generated data. Please see **Appendix G** for detailed results.

5. **Training Hyperparameters:**

    We have added detailed information on the hyperparameters used during training, including the number of training epochs. **Table 4** provides this information for the main experiments on the ZINC250K and Polymer datasets, and the new **Appendix Table 9** includes the hyperparameters for the unconditional generation tasks.

6. **Trajectory Images:**

    We have included trajectory images of the generation process for both the main results and the new experiments conducted during the revision. These can be found in **Appendix Figures 16-20**. We believe that visualizing the generation results that were analyzed only with NLL and other metrics would provide an intuitive understanding of the generation process.

7. **Impact of the Number of Function Evaluations (NFEs):**

    We compared the generation performance (Validity, FCD) of DDSBM and DBM on the QM9 task across different NFEs (5, 10, 20, 25, 50, 100). The results show that DDSBM maintains performance reasonably well even at low NFEs, whereas DBM shows a more pronounced performance degradation under the same conditions. We attribute this to that DDSBM finds optimal generation paths that can be predicted with less NFEs.

We believe that these additional results and discussions have significantly improved the quality of our paper. Once again, we thank to all the reviewers for their time and effort in reviewing our paper.

---

### Meta-Review · Area_Chair_N4YZ · 2024-12-20

**Metareview:**

The authors introduce Discrete Diffusion Schrödinger Bridge Matching (DDSBM), a new method for transforming data in discrete spaces like graphs using continuous-time Markov chains. DDSBM improves key properties of molecules with minimal changes to their structure, preserving other important features.

Strengths: see comments by reviewers.

Weaknesses: The main major weakness after rebuttal was raised by reviewer 7pXr, who requested comparisons with the latest discrete flow matching methods.

Overall the paper was well received the rest of the reviewers.

**Additional Comments On Reviewer Discussion:**

See my comments above.

---

### Decision · Program_Chairs · 2025-01-22

Accept (Poster)